# Dense water formation in the coastal northeastern Adriatic Sea: the NAdEx 2015 experiment

Ivica Vilibić[1], Hrvoje Mihanović[1], Ivica Janeković[2,3], Clea Denamiel[1], Pierre-Marie Poulain[4], Mirko Orlić[5], Natalija Dunić[1], Vlado Dadić[1], Mira Pasarić[5], Stipe Muslim[1], Riccardo Gerin[4], Frano Matić[1], Jadranka Šepić[1], Elena Mauri[4], Zoi Kokkini[4], Martina Tudor[6], Žarko Kovač[1], Tomislav Džoić[1]

[1]Institute of Oceanography and Fisheries, Split, Croatia
[2]Ruđer Bošković Institute, Zagreb, Croatia
[3]The University of Western Australia, School of Civil, Environmental and Mining Engineering & UWA Oceans Institute, Crawley WA, Australia
[4]Istituto Nazionale di Oceanografia e di Geofisica Sperimentale – OGS, Trieste, Italy
[5]University of Zagreb, Faculty of Science, Andrija Mohorovičić Geophysical Institute, Zagreb, Croatia
[6]Meteorological and Hydrological Service of Croatia, Zagreb, Croatia

*Correspondence to*: Ivica Vilibić (vilibic@izor.hr)

**Abstract.** The paper investigates wintertime dynamics of the coastal northeastern Adriatic Sea, and is based on numerical modelling and in situ data collected through field campaigns executed during the winter and spring of 2015. The data have been collected by a variety of instruments and platforms (ADCPs, CTDs, glider, profiling float), and have been accompanied by atmospheric-ocean ALADIN/ROMS modelling system. Research focus has been put on dense water formation, thermal changes and circulation, and water exchange between the coastal and open Adriatic. According to both observations and modelling results, dense waters are formed in the northeastern coastal Adriatic during cold bora outbreaks. However, dense water formed in this coastal region has, due to lower salinities, lower densities than dense water formed at the open Adriatic. Since the sea is deeper in the coastal area than at the open Adriatic, dense waters from the open Adriatic occasionally enter the coastal area near the bottom of the connecting passages, while the surface flow is mostly outward from the coastal area. Median residence time of the coastal area is estimated to be about 1-2 months, indicating that the coastal area may be relatively quickly renewed by the open Adriatic waters. The model significantly underestimates currents and transports in connecting channels, which may be a result of a too coarse resolution of atmospheric forcing, misrepresentation of bathymetry or absence of the air-sea feedback in the model. Obtained data represents a comprehensive marine dataset, pointing to a number of interesting phenomena to be investigated in the future.

## 1 Introduction

Due to its geographical position and surrounding orography, the Adriatic Sea - a semi-enclosed 800 x 200 km basin located at the north of the Mediterranean (Fig. 1) - can be considered as a unique testbed where a number of processes important

for driving circulation of the Eastern Mediterranean Sea happen (Malanotte-Rizzoli et al., 2014). Dense water formation (DWF) is one of these processes. In the Adriatic Sea, the DWF occurs through both water column cooling and mixing on the shallow and wide northern Adriatic shelf (Vested et al., 1998) and through deep-convection in the 1200-m deep circular South Adriatic Pit (Gačić et al., 2002). The cooling at both locations is a result of strong bora wind (Grubišić, 2004; Grisogono and Belušić, 2009) which may cause widespread heat losses up to 1000 W/m$^2$ (Supić and Orlić, 1999) and localised heat losses up to 2000 W/m$^2$ (Janeković et al., 2014). Although of secondary importance, bora-driven evaporation is also contributing to large densities in the northern Adriatic (Mihanović et al., 2013). Adriatic dense waters are important for: (i) replenishment of deep waters in the Eastern Mediterranean (Roether and Schlitzer, 1991; Bensi et al., 2013), (ii) changing or maintaining internal vorticity of the northern Ionian (Gačić et al., 2010) and (iii) driving decadal oscillations of thermohaline and biogeochemical properties in the Adriatic (Buljan, 1953; Gačić et al., 2010; Civitarese et al., 2010; Batistić et al., 2014).

Focusing on the northern Adriatic, up to 2012 it has been thought that the DWF occurs over open shelf areas only (Vilibić and Supić, 2005). Therein, a pool of very dense waters is created by a double-gyre circulation driven by spatial inhomogeneity of the bora wind (Zore-Armanda and Gačić, 1987; Kuzmić et al., 2006). The generated dense waters are gravitationally transported towards middle Adriatic depressions though a bottom density current (Nof, 1983), mostly along the western Adriatic slope due to the Coriolis force (Artegiani and Salusti, 1987; Vilibić and Mihanović, 2013). A portion of waters is travelling across the southern Palagruža Sill and, when reaching South Adriatic Pit slope and the Bari Canyon, they are being transported downslope to the near-bottom layers (Querin et al., 2013; Langone et al., 2016). This concept has been supported by a number of numerical modelling studies (e.g., Beg-Paklar et al., 2001; Chiggiato and Oddo, 2008). However, this classical northern Adriatic DWF picture has been substantially changed following the exceptional DWF winter of 2012, when formation of dense waters has been observed in the northeastern coastal area as well (Fig. 1) (Mihanović et al., 2013). Subsequent modelling studies implied that up to 40% of the overall dense water generated in the northern Adriatic during winter of 2012 originated from the eastern coastal areas (Janeković et al., 2014), with a significant transport between the coastal and open Adriatic through a number of channels (Vilibić et al., 2016a). It should be emphasized that these two modelling studies were the first ones that used realistic fresh water discharges. Most of previous modelling studies used an old river climatology (Raicich, 1994), which overestimates real river discharges in the eastern Adriatic by an order of magnitude (Janeković et al., 2014), thus preventing numerical reproduction of the DWF in the northeastern coastal areas and also significantly impacting the rates of the DWF over the northern Adriatic shelf areas (Vilibić et al., 2016a).

Interestingly, atmospheric processes over the northeastern coastal Adriatic area have been thoroughly researched, as the maximum of the cold and dry bora wind and its spatial and temporal variability have been reported there, occasionally reaching hurricane strength (Grubišić, 2004; Grisogono and Belušić, 2009; Kuzmić et al., 2015). As opposed to meteorology, less has been known of oceanography of the area. For a long time, the coastal northeastern Adriatic has been considered an area in which significant freshwater fluxes strongly affect thermohaline properties (e.g. Orlić et al., 2000). These freshwater discharges normally come through occasional floods accumulated over the 150-km long and 1600-m high mountain ridge of Velebit (Perica and Orešić, 1997) and a large number of submarine karstic springs (Sekulić and Vertačnik, 1996; Bonacci,

2001; Benac et al., 2003; Surić et al., 2015). Further on, thanks to occasional oceanographic campaigns, the inner area of Velebit Channel has been classified as a two-layer system, with surface salinity exhibiting much lower values (~1.0) than at the open Adriatic (Viličić et al., 2009). Also, it has been known that a strong northern Adriatic thermohaline front (Lee et al., 2005; Poulain et al., 2011) has its starting point in the northeastern coastal Adriatic, specifically in the Kvarner Bay, with

coastal waters advected towards the open sea, particularly during strong bora events (Pullen et al., 2003; Lee et al., 2005; Beg Paklar et al., 2008). As it is topographically separated from the open Adriatic by a number of islands (Fig. 1), the northeastern coastal area has not been considered as eligible for wintertime processes such as dense water formation up to winter of 2012, at least not at rates that may impact the overall northern Adriatic dynamics.

Up till now, winter of 2012 remains the only winter for which dense water formation in the northeastern coastal

Adriatic has been observed and modelled. Question is whether this is due: (i) to exceptionality of the 2012 winter - implying that this was an extraordinary event, or (ii) to a lack of observational campaigns and poor model performance in the area - pointing to both a possibility of regular dense water formation in the area and an omission in our previous research efforts. To answer this question we have envisioned and carried out the North Adriatic Dense Water Experiment 2015 (NAdEx 2015). A number of different platforms and instrumentations for data collection were engaged (Fig. 1), along-side with a state-of-the-

art nested atmosphere-ocean modelling system, all during winter/spring of 2015, i.e. during and post a common Adriatic DWF period. Obtained experimental data and modelling results allow us to: (i) estimate whether the DWF is a common process in the northeastern Adriatic; and (ii) if so, to quantify both DWF and thermohaline changes leading to it; (iii) estimate rate of exchange between coastal and open Adriatic waters through a number of connecting passages, and (iv) get an insight into a mesoscale and frontal processes driven by such a complex area.

Section 2 gives details of the field experiment and data used in this paper, together with a description of the atmosphere-ocean modelling system, Section 3 documents the atmospheric conditions during the winter/spring of 2015, Section 4 describes representative ocean observations, followed by model verification in Section 5. Section 6 displays thermohaline, stratification and buoyancy changes as reproduced by the model, followed by estimates of temperature, salinity and volume transports at the boundaries of the region, including residence times. A throughout discussion and major

conclusions are presented in Section 7.

## 2 Data and methods

### 2.1 The studied area

The northeastern Adriatic is a coastal region consisting of a number of elongated channels and bays (Fig. 1). It communicates with the open Adriatic through a number of narrow (from a kilometre to few kilometres) channels; the only

exception is a wide opening connecting Kvarner Bay to the open Adriatic - Kvarner Bay may thus be considered as a crossing region between coastal and open Adriatic waters. The depth of this inner coastal region is larger (80-100 m) than of the open Adriatic (50-70 m). Only a few small river mouths are located in the area, plus the freshwater input by the hydropower plant

near Senj, altogether with average freshwater input rates of about 80 m³/s (Vilibić et al., 2016a). Yet, there are also tens of submarine springs, which are quite active during and after the prolonged precipitation events, and which may double the freshwater load to the coastal area (Sekulić and Vertačnik, 1996). Furthermore, climate of the region, particularly of Rijeka Bay, is characterized by significant precipitation driven by orography (Gajić-Čapka et al., 2015).

## 2.2. The field experiment

The NAdEx 2015 experiment was carried out between late autumn 2014 and summer 2015. Primary goal of the experiment was to study the DWF in in the coastal northeastern Adriatic, commonly occurring between January and March (Janeković et al., 2014). Temperature, salinity and current data were collected by a number of instruments and observing
platforms deployed in the area (Fig. 1). The whole experiment has been realized through voluntary contributions and collaborative work of several research institutions: Institute of Oceanography and Fisheries, Ruđer Bošković Institute, Geophysical Department of the Faculty of Science of the University of Zagreb, Meteorological and Hydrological Service, all from Croatia, and National Institute of Oceanography and Experimental Geophysics, Italy, thus representing an unique effort for the Adriatic which may serve as a good practice for future research activities in the region.
Currents over the water column were measured at stations A1 to A9 using RDI Acoustic Doppler Current Profilers (ADCPs) between late November 2014 and early August 2015 (A7, A8, A9)/early July (A4) and using Nortek ADCPs between early December 2014 and mid-August 2015 (A1, A5, A6)/late May (A2); ADCP at station A3 malfunctioned after only one week of operation and did not measure any data after that. A Seabird 911 CTD probe accompanied ADCPs at stations A3, A4, A7, A8 and A9, providing the bottom temperature and salinity series between late November 2014 and early August 2015.
Vertical profiles of temperature and salinity data have been acquired by Seabird SBE 25 probe at 19 CTD stations during two cruise legs. Leg 1 was carried out between 3 and 6 December 2014 and Leg 2 between 26 and 29 May 2015. A Teledyne–Webb Research Slocum glider was operated along transect off the Kvarner Bay in a campaign lasting from 24 to 26 February 2015, while Arvor-C profiling float was deployed on 19 February in the northern part of Kvarner Bay and recovered on 15 March 2015 on the Istria coast near the entrance of the bay, profiling regularly the whole water column every 3 hours (Gerin
et al., 2015). Potential density anomaly (PDA, reference pressure equalling zero) was computed from temperature and practical salinity data, following TEOS-10 algorithms (described at http://www.teos-10.org). Complete setting of the experiment is illustrated in Fig. 1.

## 2.3. The modelling system and its setup

The atmospheric-ocean modelling system covering the entire Adriatic Sea has been used as the NAdEx 2015 parent numerical model. Atmospheric part of the system is based on a hydrostatic version of ALADIN numerical weather prediction (NWP) model used by the Meteorological and Hydrological Service of the Republic of Croatia (Tudor et al., 2013, 2015). The

model is operationally integrated four times per day, having 37 sigma levels in vertical and 8 km resolution in horizontal, except for winds which are dynamically downscalled to 2 km (Ivatek-Šahdan and Tudor, 2004). All variables have been provided with the time step of 3 h. Although bora wind may have a substantial variability on periods from several minutes to a few hours, previous modelling studies that used 3-h ALADIN/HR forcing provided reliable results (e.g. Janeković et al.,

2014). The model is initialized with 3Dvar run, at 8 km resolution, using all data through Global Telecommunication System (GTS) and local data exchange (Stanešić, 2011). The model uses SST fields coming from the IFS (Integrated Forecast System) operational forecast run in ECMWF (European Centre for Medium Range Forecast). These SSTs have a positive bias towards in situ measurements during the winter, much lower at the open Adriatic when compared to the SST satellite observations, affecting precipitation maxima but not significantly the wind speed, which is controlled by surrounding topography (Tudor et

al., 2017). Wind gusts have been computed following Brožkova et al. (2006) formulation, which has been tuned for oceanographic simulations in the Mediterranean. ALADIN/HR simulations have been verified in the coastal northeastern Adriatic during severe bora events (Tudor et al., 2012, 2013).

For the ocean part, Regional Ocean Modelling System (ROMS) has been used. ROMS is a 3-D hydrostatic, nonlinear, free surface, s-coordinate, time splitting finite difference primitive equation model (Shchepetkin and McWilliams, 2005, 2009).

Horizontal resolution of the Adriatic model is 2 km, with 20 sigma layers in vertical, following the studies by Janeković et al. (2014) and Benetazzo et al. (2014) which satisfactorily reproduced the DWF in the northern Adriatic. The open boundary conditions at the Otranto Strait (free surface, temperature, salinity, and velocity) are taken from the Adriatic REGional model (AREG, Oddo et al., 2006), with sponge layer at the boundary. The Flather scheme was used for barotropic velocities and a combination of Orlanski-type radiation boundary conditions with nudging (Marchesiello et al., 2001) for baroclinic velocities

and tracers (temperature and salinity). Long-term stability of the model run has been ensured by smoothing bathymetry using a linear programming technique (Dutour Sikirić et al., 2009) that suppresses horizontal pressure gradient errors occurring over complex bathymetries with steep slopes, such as in the Adriatic Sea, and during multiyear integrations (Haidvogel et al., 2000). The ALADIN/HR surface variables were introduced to the ROMS via bulk parameterisation (Fairall et al., 1996). The most recent river discharge climatology has been imposed at the freshwater point sources following Vilibić et al. (2016a) data,

without changing the ambient temperature. More details on the modelling system can be found in Janeković et al. (2014) and Vilibić et al. (2016a).

In addition to the Adriatic model, a nested ocean model (ROMS as well) was imposed to the NAdEx 2015 region to properly reproduce its complex bathymetry (Fig. 1). The nested domain has been tilted by 45$^\circ$ to follow the orientation of the area. The nesting has been done using the 1:4 ratio in horizontal - thus nested model had a horizontal resolution of 500 m - and

keeping 20 sigma levels in vertical. The nested ocean model was forced with the same ALADIN/HR operational fields as the parent model.

The parent modelling system has been operationally integrated since 1 January 2008, while the nested simulation was run between 1 October 2014 and 30 September 2015, covering the experimental NAdEx 2015 period. Verification of the parent model was performed for the winter of 2012 (Janeković et al., 2014; Vilibić et al., 2016a). Basin-wide negative salinity bias

has been found to exist, presumed to come from the AREG model lateral boundaries (Janeković et al., 2014). In the AREG model these boundaries exhibit basin-wide overfreshening coming from old river climatology by Raicich (1994), thus influencing also our parent simulations. Yet, the model was found appropriate for reproduction of thermohaline properties in the area (Vilibić et al., 2016) and quantification of the DWF in both open northern and coastal northeastern Adriatic (DWF sites 1 and 2 in Fig. 1).

## 3 Atmospheric conditions and air-sea interaction

The winter of 2015 (December through March) was characterized by warmer-than-average temperature conditions over the NAdEx area compared to the baseline climatological period of 1961-1990 (MHS, 2015, 2016). According to these reports, the highest positive anomalies (91-98 percentile) were recorded in December and January, followed by average temperatures in February and warm conditions (75-91 percentile) in March. DJFM precipitation values (measured only above land) were close to climatological values, with highest positive anomalies measured in February. Regarding average January-February net heat fluxes over the NAdEx area (as delimited by nested domain boundaries in Fig. 1) - these two months are chosen as the DWF is dominantly occurring at that time (Beg Paklar et al., 2001; Vilibić and Supić, 2005) - the winter of 2015 may be classified as a normal with the respect to other winters between 2008 and 2015. Precisely, cumulative January-February net heat losses equalled to 0.80 MJ/m$^2$, about 50% less than in the winter of 2012 (1.20 MJ/m$^2$) and almost two times larger than in the winter of 2014 (0.49 MJ/m$^2$).

Several cooling events occurred during the winter of 2015, of which three bora episodes – preceding New Year, in early February and early March – were particularly severe. The first severe bora episode lasted for several days (between 28 December 2014 and 1 January 2015), with gusts stronger than 50 m/s in the Velebit Channel and air temperature falling below 0$^\circ$C The bora event between 4 and 7 February 2015 was particularly strong over the NAdEx area, peaking during the night of 5/6 February with measured wind gusts of about 60 m/s in the northern Velebit Channel. A month later, another strong bora event occurred along the eastern Adriatic coast, the latter was however particularly pronounced over the middle Adriatic and southern part of the Velebit Channel, where the wind gusts peaked on 5 March with values of about 55 m/s.

To better understand impact of bora wind on the northeastern coastal Adriatic, we have compared wind stress, net heat flux and water flux variables (all originating from ALADIN model) averaged over all bora events with those averaged between 15 December 2014 and 15 March 2015 (Fig. 2). Herein bora event is defined as a period during which wind blows from ENE and exceeds 15 m/s at the ALADIN grid point off Senj (location G1 in Fig. 1) - ENE represents predominant direction of bora in that area (Zaninović et al., 2008). Maximum wind stress is modelled within the Senj Jet (marked by an arrow in Fig. 2), with area of wind stress stretching from the Kvarner Bay towards the western shore, exactly at the location where major wind jet and ocean frontal zone are commonly found (Pullen et al., 2007; Beg Paklar et al., 2008; Kuzmić et al., 2015). The largest bora-driven heat losses was documented in the Velebit Channel and within the offshore jets, again reaching maximum in the Senj Jet. Heat losses strongly decrease towards the western Adriatic coastline. Such bora driven heat loss

distribution largely follows distribution associated with the extreme bora wind outbreak of winter of 2012 (see Fig. 4 in Janeković et al., 2014). The pattern of bora heat losses also resembles average net heat losses between 15 December 2014 and 15 March 2015, indicating that cooling of the northern Adriatic waters dominantly happens during bora episodes.

As it is strongly dependent on wind speed and humidity, the evaporation pattern (not shown) driven by bora wind follows the net heat loss patterns, with maximum rates exceeding 10 mm/day off Senj. Yet, an interesting pattern is found for the water flux (E-P) associated to the bora episodes, with highest negative values at the open Adriatic and particularly at the western coastline. Negative values may be also found in the NAdEx 2015 area. This implies that the bora wind – as defined by using single station at the core of the strongest jet - is associated with precipitation that appears at the back side of a cyclone, decreasing its rates toward the northwest - Gulf of Trieste - where maximum water uptake has been modelled.

We can conclude section on atmospheric conditions by saying that, in spite of three strong bora events, no exceptional cooling events were observed during winter of 2015. This gives us an opportunity to study whether dense water is generated in the northeastern coastal area during average or even milder-than-average winter conditions.

## 4 Ocean observations

Temperature and salinity data measured between 3 and 6 December 2014 (Leg 1 cruise) at the transect stretching over the NAdEx 2015 area (stations 1 to 19) exhibit a predominant two-layer thermohaline structure (Fig. 3a), with warmer (>16°C) and less saline (<37.0) waters in the surface and intermediate layers - to depths of about 50-60 m. These depths were characterized by a sharp thermocline, under which a pool of colder (13-14°C) and more saline (~38.0) waters was residing. The pool had a substantially higher density (PDA>28.5 kg/m$^3$) than waters residing above (<27.8 kg/m$^3$). Thermocline and halocline followed each other in the first third of the transect (up to station 6), after which halocline formed at smaller depths. Maximum in salinity stretching over the whole water column at stations 11 to 13 indicates an inflow of saline open Adriatic waters through connecting channels where A3 and A4 ADCPs have been moored.

Six months later, during Leg 2 cruise executed between 26 and 29 May 2015, two-layer structure was still evident from temperature data (Fig. 3b), but this time was driven by seasonal heating during springtime (Buljan and Zore-Armanda, 1976). Thermocline was positioned at depths between 20 and 30 m. However, salinity was homogenised over the whole transect, with substantially higher values (37.8-38.0) than observed during Leg 1 cruise, peaking again at stations 11 to 13. The salinity changes between Leg 1 and Leg 2 cruises indicate an advection of high salinity waters from the open Adriatic towards the whole NAdEx 2015 area, occurring dominantly through the T4 connecting passage. Unfortunately, the latter cannot be confirmed by currents measured at station A4, as the station has been positioned to the southwest of the connecting channel, and was thus more influenced by the open-Adriatic northeastern current - a dominant circulation feature in the area (Orlić et al., 2006). PDA distribution was following temperature distribution, with much higher values present in deep layers (29.0-29.1 kg/m$^3$) than during the Leg 1 cruise. The latter indicates that, in spite of lack of extreme cooling events, the DWF

did occur during wintertime, although observed temperatures were much higher and salinities lower than during the extreme winter of 2012 (Mihanović et al., 2013).

The thermohaline properties measured at the bottom of connecting channels, over which the transport between the coastal and open Adriatic area is happening, reveal a rate of the wintertime cooling that happened in the northern Adriatic (Fig. 4). A constant decrease in temperature (Fig. 4a) from beginning of the experiment (early December) up to the end of March was recorded at all stations, having a weak step-like structure presumably associated with strong bora events. At station A4, bottom temperatures were higher, not decreasing below 12°C. Lowest temperature has been observed at the northwestern part of the Kvarner Bay entrance, at station A9, where minimum of about 10.5°C was reached in early February and remained through March. By contrast, these temperatures were observed at neighbouring stations A8 and A7 a month later, indicating a presence of a complex circulation and a deep thermohaline front within the bay. In support to the existence of the front, the difference between temperature and salinity series measured at A7, A8 and A9 stations between 1 February and 31 March 2015 (in terms of their averages and variability) is significant at the 99% level. Existence of the wintertime thermohaline front through the water column can be clearly seen on glider measurements performed off the Kvarner Bay on late 26 February (Fig. 5), when strong bora conditions were present in the area. However, the front weakened the day after, when the glider returned back over the almost same track. Kokkini et al. (2017) ascribe the variability of the front to the wind forcing, where strong bora wind is in favour of a sharp front. This front has also been observed and investigated during previous wintertime campaigns (Lee et al., 2005; Poulain et al., 2011).

As for bottom salinity (Fig. 4b), it decreased in mean values when going northwestward, from station A4 to station A7, and again across the Kvarner Bay entrance to the station A9. In addition, strong salinity variability at daily and weekly scale was embedded into the series, varying between 37.5 and 38.5 at A7 to A9 stations. Such a pronounced variability indicates presence of a thermohaline front, changing its position over time. The variability was particularly strong during wintertime (January-March 2015), decreasing during the springtime. Although having monthly variations, an overall increase in salinity was recorded at all stations between mid-December 2014 and early February 2015. Temperature and salinity southeast-northwest differences along the outer NAdEx 2015 area are reflected in PDA values (Fig. 4c), which increased from mid-December ($\sim$28.0 kg/m$^3$) to mid and late February ($\sim$29.2 kg/m$^3$). Maximum PDA values were sustained until late March, presumably associated with near-bottom outflow or inflow of dense waters. PDA values slowly decreased during springtime.

Pronounced spatial and temporal changes in thermohaline properties of the Kvarner Bay may be quantified by analysing the profiling float data (Fig. 6). Arvor-C float temperature and salinity profiles obtained at the inner part of the Kvarner Bay show a weak stratification over the water column, except at the very bottom, where a few metres thin layer with substantially higher temperatures ($\sim$0.8°C) and salinities ($\sim$0.5) was detected. This thin near-bottom layer was not present over the central western Kvarner Bay where the float was transported between 22 and 25 February (for position of the float see Fig. 1). The layer was however present at the outer western part of the bay where float drifted between 3 and 5 of March. PDA values of this bottom layer reached 29.4 kg/m$^3$, being larger than PDA measured at the A stations or CTDs for about 0.3 kg/m$^3$. Similar few metres thin density layer which was found in mid-February 2012 at the western Adriatic shelf was associated with

the initial near-bottom outflow of dense waters from the northern Adriatic (Vilibić and Mihanović, 2013). Analogously, our hypothesis is that a small fraction of dense waters generated on the northern Adriatic shelf, having higher densities than waters of the Kvarner Bay (Janeković et al., 2014), spreads towards the deeper Kvarner Bay as a weak bottom density current.

Question is, do dense waters spread from the coastal area to the open Adriatic as well? And if so, which of these two processes is more important? Assessment of wintertime ADCP data (Fig. 7) reveals a substantial baroclinic component atop the barotropic circulation at all stations during and after two strong bora episodes (1 February – 1 April 2015), when the DWF and dense water flow were expected to occur. A weaker currents at station A9 and strong outflow at A7 and A8 stations in the surface layer indicate presence of an anticyclonic curl at the entrance of the Kvarner Bay. The pattern in currents also resembles patterns of local wind stress and wind curl, which are pronounced off the southern tip of the Istria Peninsula (Pullen et al., 2003; Grubišić, 2004). An inflow to the Kvarner Bay may be seen in the bottom layer of the station A9 - part of this inflow might be ascribed to dense waters coming from the northern Adriatic shelf, thus confirming our hypothesis of the origin of the bottom density current seen in Arvor-C data. Near the bottom of stations A7 and A8, the mean flow is weak, yet changeable in time, suggesting an interplay between dense waters coming from the coastal area with those coming from the northern Adriatic shelf. Going to the southeast, at station A4, measured currents were parallel to the coastline. However, being deployed too far from the connecting channel, this station was indeed not measuring interchange between coastal and open Adriatic waters but the Eastern Adriatic Current, which may be strong in that region (Orlić et al., 2006). Currents measured at the station A2, located in the Sedmovraće Channel, exhibit a strong baroclinic pattern, pointing to an exchange of waters between open and coastal Adriatic: outflowing current is present in the surface layer and inflowing current near the bottom. Yet, these currents are strongly affected by local bathymetry, probably resembling the effects of both very narrow connecting channel (about 600 m in width) and the Eastern Adriatic Current modulated by the cape of Veli Rat (about 2 km south of station A2). Finally, current data measured at the A1 station document predominant inflow of the open Adriatic waters, mostly in the surface layer. The inflow is likely driven by orientation of the channel and the incoming Eastern Adriatic Current. Interestingly, wintertime baroclinic circulation, with a predominant outflow from Velebit Channel in surface layer and inflow in bottom layer, is also maintained in the inner channels (stations A5 and A6). This particularly refers to the currents measured at the station A6 located near the Senj bora jet, implying that currents at this stations are likely largely wind-driven.

**5 Model validation**

The modelling system was validated against available observations. Verification on the CTD data collected over the Leg 1 cruise (Fig. 8a,c) reveals an underestimation of temperature in surface layer and overestimation of temperature in deep layers, where a pool of cold and dense waters was observed in reality (Fig. 3a). Oppositely, salinity has been overestimated in surface layers and underestimated in near-bottom layers. An underestimation in salinity in the most shallow southeastern part of the transect (around station 18) is presumably due to submarine springs which are discharging freshwater from neighbouring Vrana Lake to the sea and which are not introduced to the model. Altogether, the model did not properly reproduce the observed

two-layer structure, but rather much more homogenized water column, without dense water pool in the deepest parts of the NAdEx area. Thermohaline properties during the Leg 2 cruise (Fig. 8b,d) show better agreement with the observations, particularly salinity, where the bias is about 3 times lower than for the Leg 1 cruise. This particularly refers to the surface layer (Fig. 9). The temperature bias has also been smaller during Leg 2 cruise. However, a drawback was present in reproduction of

the thermocline, as the largest root-mean-square error is present exactly at these depths (15-35 m). Nevertheless, model successfully recreated presence of cold and saline bottom layer, as well as a two-layer structure observed during the Leg 2 cruise, keeping both bias and root-mean-square-error pretty low near the bottom. Modelled bottom PDA values were higher (around 29.0 kg/m$^3$) at dates corresponding to the Leg 2 cruise, than at dates corresponding to the Leg 1 cruise (around 27.8 kg/m$^3$), indicating that model was able to reproduce the DWF in the NAdEx area.

Model verification performed on the float and glider data is shown in Fig. 10. The float data has been verified on the nested model simulation, while parent model simulation has been used for verification of glider measurements. An inspection of results indicates that the model is able to reproduce observed thermohaline properties inside the Kvarner Bay. There are however several omissions there: (i) both temperature and salinity show an increase in negative bias from inner to outer part of the bay; (ii) the model is not able to reproduce narrow bottom density current. The latter is because the model does not have

sufficient resolution in vertical to reproduce such a thin bottom layer. An overestimation of both temperature and salinity of the open Adriatic waters (negative bias) has been visible on the model-to-observation differences along the glider pathway, where the parent model does produce warmer and saltier water northwest from the measured thermohaline front. By contrast, temperature and salinity biases were much lower in absolute values southwest from the front. These results imply two conclusions: (i) the position of the thermohaline front was not properly modelled, and (ii) the strength of the front in the model

is much weaker than captured by glider observations.

          Observation-to-model Q-Q plots of temperature and salinity constructed by comparing float, glider and CTD data (Fig. 11a, b) indicate that temperature data have been reproduced fairly well over the inner NAdEx area (float and CTD), but not at the open Adriatic, i.e. in the area off Kvarner Bay (glider). As for salinity, model overestimated values below 37.6 compared to the CTD measurements, while higher salinities were successfully reproduced. Salinities measured with the Arvor-

C profiler and glider were generally overestimated by the model over most of percentile distribution, except for salinities around 37.6 and for the upper tail of the distribution (>38.1). This particularly holds for salinities measured by the glider, i.e. for salinities modelled with a lower resolution parent model.

          Box-Whiskers plots (Fig. 11c, d) confirms that model reproduced well the temperatures in the Kvarnerić and Kvarner Bays (CTD and float measurements), with a minor portion of relative differences exceeding 10%. However, temperatures

measured by the glider off Kvarner Bay were significantly overestimated by the parent model. Summarily, the model best reproduced salinity in the interior of the basin, a bit worse in the Kvarner Bay and much worse off the bay.

          Comparison of bottom temperature and salinity, and bottom current speed data, as measured at stations A1 to A9 (Fig. 12), indicates that temperature and salinity have been fairly reproduced by the model at all stations, differing by less than 1$^o$C (10%) and 0.3 (1%), except for some sparse data. That also refers to the reproduction of thermohaline properties and

changes in time, which exhibit low biases - particularly for temperature - at all stations over the whole measuring interval (not shown). Consequently, a significant decrease in temperature and salinity properties in the outer Kvarner Bay (stations A7 to A9) has been also reproduced by the model. The biases are slightly larger at A7, A8 and A9 stations, particularly in temperature (0.3-0.6ºC), as the model do not reproduce a thin near-bottom inflow of warmer and saltier waters in the outer part of the Kvarner Bay (Fig. 5). Yet, these biases, and also overall temperature, salinity and current biases at most of A stations are smaller than root-mean-square values (not shown), in line with a fair reproduction of thermohaline properties in the NAdEx area by the model. However, current speeds have been strongly underestimated by the model at all stations, between 50% and 80% in average. There may be several reasons for that: (i) too coarse horizontal resolution of ocean model, (ii) insufficient resolution of atmospheric model and inappropriate reproduction of bora-driven mesoscale variability, and (iii) inappropriate boundary conditions. Reasons for this underestimation will be discussed in more detail in Section 7.

Aside underestimation, comparison of mean currents and the associated standard deviation ellipses (Fig. 7) exhibit a number of differences between modelled and observed currents, being largely the result of the complex bathymetry. Yet, vertical structure of currents - i.e. a rate of change of current speed over the vertical – has been fairly modelled at all stations, except A2. The latter is the results of the complex bathymetry in the region underrepresented by the model, as connecting channel Sedmovraće, off which the station A2 has been positioned toward the open sea, is very narrow, about 600 m in its deep section. Also, the station is located a bit off the channel, presumably exhibiting a strong interaction between the Eastern Adriatic Current and the channel current. At all other stations the model reproduce either the surface maximum in currents and a decrease towards the bottom (A1, A4, A7, A8, A9) or maximum currents in the bottom layer (A5) or two-layer circulation (A6). The current direction is fairly reproduced at A1, A4, A5, A6 and A8, but much worse at A7 and A9 (plus A2). Still, the overall transport from the inner coastal area towards the open sea has been properly captured by the model for all stations, except A2.

In conclusions, the model is reproducing thermohaline properties and the DWF in the coastal northeastern Adriatic (inner domain) fairly well and may thus be used for quantification of related processes and dynamics there. It should however be taken into account that a restricted (weaker than observed) water mass communication has been reproduced between coastal and open Adriatic through connecting passages.

## 6 Model results

### 6.1 Thermohaline, buoyancy and stratification changes

Modelled temporal changes of thermohaline properties at location G1, positioned in the Velebit Channel at the core of the Senj bora jet, and at location G2, positioned in the outer Kvarnerić Channel, are displayed in Fig. 13. Water column was vertically homogenized over most of the basin from December until early April, except in areas influenced by the freshwater load. Later on, surface thermocline developed, deepening to about 30-40 m in early July. Salinity series at the G1 location exhibit pronounced daily and weekly changes in the upper layer, as being influenced by the nearby freshwater discharge of the

Senj hydropower plant. Yet, vertical homogeneity have been reached there during cooling events around 30 December 2014, 5 February and 5 March 2015 - this is due to its position on the track of the Senj bora jet. Between bora events, the ocean started to relax through horizontal advection. The cooling was also modelled at the G2 location, but with a milder stepwise structure, as this location is in the wake of the bora wind - thus the cooled waters were largely advected to this area in the days following the cooling events.

A simple box model (e.g. Gill, 1982) of energy balance has been applied to these locations, relating the decrease of the ocean temperature $\Delta T$ in a box to surface heat losses:

$$\Delta T = \frac{1}{Hc\rho_0} \int_{t_1}^{t_2} Q \, dt$$

where $Q$ is the surface heat flux in the time interval between $t_1$ and $t_2$, while $H$ is the ocean depth. The specific heat of sea water c and sea water density $\rho_0$ was approximated by constant values of 3990 J/kg K and 1027.5 kg/m$^3$. This model assumes no lateral exchange of energy between the box and the adjacent sea, what is a fair approximation for short and transient events like the bora. At station G1 this simplified formula gives $\Delta T$ = -0.96°C, -0.48°C and -0.03°C for three bora events (the third bora was very weak at G1), respectively, while the respective cooling rate as provided by the model is -1.04 °C, -0.51 °C and -0.07°C. Given the assumptions of this simple box model, one can conclude that the cooling in the area is dominantly driven by the bora wind.

Density persistently increased at both locations until mid-March, when maximum PDA values were modelled at both G1 and G2 locations. This maximum is a results of the severe bora wind episode that peaked in some parts of the area on 5/6 March (see Section 3). As a consequence, thermohaline circulation strengthened and the open-Adriatic saline waters were advected to the coastal area, particularly to its outer parts (G2 location). An increase in salinity in the coastal area has been additionally intensified in May, presumably driven by the lagged thermohaline circulation of the Adriatic-Ionian basin (Orlić et al., 2006).

Buoyancy changes, estimated following Marshall and Schott (1999) over the nested model domain, document buoyancy loss that was dominantly occurring during bora outbreaks (Fig. 14). The most pronounced buoyancy loss occurred around 30 December 2014, being largest in the Velebit Channel inner area, a bit lower along the bora jets (Senj Jet, Pašman Jet, Janeković et al., 2014), and lowest at a wake of the bora wind (e.g. G2 location). Buoyancy loss was predominantly driven by the heat loss, while haline-driven buoyancy changes were of minor importance. Buoyancy losses during bora events decreased stratification of the area (Fig. 15) – stratification index has been computed following Turner (1973) for the whole water column - which was present in some parts of the basin prior to bora events. This particularly applies to the inner Velebit Channel, where maximum buoyancy losses with rates high enough to homogenize the whole water column were modelled. We can conclude by saying that model successfully reproduced DWF during severe bora outbreaks. Knowing this, we can study dense water dynamics in the area and interchange between inner coastal and open Adriatic waters in more detail.

**6.2 Lateral boundary fluxes, residence and flushing times**

Heat and salinity fluxes normal to transects T1 to T789 (notation is following numeration of stations A1 to A9) and averaged between 15 December 2014 and 15 May 2015 are shown in Fig. 15. Positive fluxes are considered if directed towards northeast at T2, T3, T4 and T789, and towards northwest at T1, T5 and T6. Heat fluxes indicate that the coastal northeastern Adriatic gained energy mostly over transects T1 and T3. This applies particularly to T1 surface layers and T3 bottom layers, while near-surface heat fluxes were of opposite direction at the T3 transect. Fluxes at T2 transect also show an inward-outward structure, with the strongest outward values modelled near the bottom. Fluxes were predominantly weakly negative along the T4 transect and strongly negative along the T789 transect, indicating that the coastal area was losing energy and salt through the northern connecting passages, in particular through the Kvarner Bay. The predominance of negative fluxes, occurring mostly through the Kvarner Bay, may be perceived through averaging transports over transects and the outer lateral boundary (Table 1) ($T_{outer}$=T1+T2+T3+T4+T789). In summary, the coastal northeastern Adriatic lost energy through lateral boundaries, and in particular through Kvarner Bay boundary, during the winter and spring of 2015 (15 December 2014 – 15 May 2015), a period which encompasses preconditioning, generation and spreading phase of the DWF in the area. That is presumably due to wind-driven dynamics that dominates over the thermohaline circulation and are driven by strong bora events.

Normal modelled fluxes at inner transects (T5 and T6, Fig. 16) were mostly directed nothwestward in the very surface layer, presumably due to bora-driven currents. Fluxes were of opposite direction in the bottom layer, following two-layer circulation in these constrictions. The total inner transports ($T_{inner}$=T5+T6) show transport of energy, salt and mass towards the inner coastal area of the Velebit Channel (Table 1). However, this transport is much weaker than the transport modelled at the outer boundaries of coastal waters.

Modelled fluxes and transports are highly variable in time (Fig. 17), peaking with values 5-10 higher than average values, particularly over the outer lateral boundaries (Table 2). As they occur over the by far largest transect, transports at the transect T789 dominate $T_{outer}$ transports. It is interesting that during some peak events, such as 28 December 2014 and 22 February 2015, the T1 inward transport peaks simultaneously as the outward transport at transects T2, T4 and T789. Therefore, the peak inflow at the southern boundary of the NAdEx 2015 area, which is under the influence of the Eastern Adriatic Current (occasionally being directed toward the inner waters southeast of the transect T1), is balanced by the strong outflow at the northwestern lateral boundaries. By contrast, there are situations in which peak transports are localized over smaller parts of the domain. For example, peak outward transports at transects T2 and T3 are balanced by the inward transport at T1 on 5 March 2015. These transports were presumably driven by strong bora which blew severely in the southeastern part of the NAdEx 2015 domain, increasing its speed towards the southeast (central Adriatic). Transports at the lateral boundaries became much weaker after the early March bora, having lower values in April and May.

Dense water average mass flux (Fig. 18), defined as a flow of water with PDA >29.2 $kg/m^3$ normal to transects (Janeković et al., 2014), was largest at the outer lateral boundaries at transects T3 and T789. As expected, mass flux is directed towards the open Adriatic, but with higher values in the surface layer than near the bottom, as wind-driven transport is stronger

there. Also, the near-bottom minima can be a sign of a sporadic penetration of dense waters coming from the open sea, i.e. northern Adriatic shelf, towards the deep coastal area, supported also by the profiling float data (Fig. 5). That dense water flow may reverse, particularly at northern outer transects T4 and T789, is clearly seen in Fig. 19. The strongest dense water outflow event was modelled to occur around 22 February, about two weeks after the early February severe bora events, through all but T3 outer lateral boundary transects. During the peak dense water outflow, the dense water volume transport over outer boundaries reached the maximum value of 0.4 Sv, being comparable with the peak dense water outflow modelled during the extreme winter of 2012 (Janeković et al., 2014). The dense water transport across inner transects is also interesting, as on average it is directed towards the inner coastal area. Yet, the inflow is stronger at transect T5 in late February and late March, while the maximum inflow is occurring at transect T6 during late March and April.

Residence times are computed for both outer and inner coastal areas and for a period between 15 December 2014 and 15 May 2015. Residence time (RT) has been computed by applying the formula:

$$RT = \frac{\iiint_{x,y,z} \rho(x,y,z) \, dx \, dy \, dz}{\oiint_{C,z} \rho(x,y,z) \, \vec{u}(x,y,z).\vec{n} \, dC \, dz}$$

where $\rho(x,y,z)$ is the density of the water at each point $(x,y)$ and for each depth z of the domain, while $\vec{u}(x,y,z).\vec{n}$ is the normal velocity along the contour $C$ of the domain. Such an approach assumes that (i) only the velocities at the border of the domain are used to calculate the residence time, (ii) only the outflow of water at the border are taking into account; the goal here is to only look at how long it would take for the water masses to leave the domain assuming that there is no income of water within the studied domain, and (iii) the residence time is calculating with a time step of 3 h over the period of the studied event, assuming a steady state of the dynamical conditions at each time step. Box-whiskers statistics has been computed for total (RT) and dense water (DWRT) residence times (Fig. 20). The median RT for the whole coastal basin (bordered by outer transects T1, T2, T3, T4 and T789) has been estimated to 19 days, although it appeared much shorter during strong wind outbreak episodes (down to 7 days) and much longer during low mass exchange between coastal and open Adriatic waters (up to 40 days). Dense water residence times (DWRT; waters with PDA >29.2 kg/m$^3$) are much lower, with median value of 11 days and normally not longer than 35 days. DWRT is smaller than RT as the volume of the generated dense waters is much smaller than the volume of the whole coastal area, while the dense water outflow was substantial ad occurring over a large portion of the transects, not just at the bottom.

**7 Discussion and conclusions**

The coastal northeastern Adriatic is one of the Adriatic regions so far least investigated. There have been several reasons for this: (i) it has been considered to have no substantial influence to the overall Adriatic dynamics, as it is separated from the open Adriatic by several long islands, physically restricting water exchange, (ii) the area is positioned far away from research institutes, whose monitoring activities have thus not encompassed the area (see e.g. published data catalogues by

Buljan and Zore-Armanda (1966, 1979) and Zore-Armanda et al. (1991)), and (iii) satellite remote sensing does not perform well in the region, as numerous channels and complex topography impair quality of data (Klemas, 2011). In addition, this area has wrongly been treated as a basin with strong freshwater fluxes, particularly of riverine origin (Raicich, 1994). Recently, it has been found that this old climatology of river discharge overestimates real river discharges by almost an order of a magnitude (Janeković et al., 2014; Vilibić et al., 2016a). Only limited parts of the coastal northeastern Adriatic were investigated previously (e.g. Rijeka Bay in early 1980s, Gačić et al., 1983, outer Kvarner Bay during the winter of 2002/2003, Lee et al., 2005; Poulain et al., 2011). The NAdEx 2015 experiment is the first and up-to-now the only experiment that systematically approached monitoring of the northeastern coastal Adriatic, including communication between the open Adriatic and the coastal area through connecting channels. The NAdEx 2015 was realized through a strong collaboration and engagement of different institutions, resulting in a multiplatform marine dataset which can serve as a baseline for future investigations in the area. The choice of the experimental area – the coastal northeastern Adriatic - emerged from its role in the winter of 2012, when it exhibited an unprecedented heat loss (up to 2000 W/m$^2$) and DWF rates that strongly contributed to the overall Adriatic dense water dynamics (Mihanović et al., 2013; Janeković et al., 2014).

We have overviewed observations and estimated rates of selected basic processes, using both an extensive set of measured data and a coupled atmosphere-ocean model. Our main intention was to answer two questions: (i) whether or not the DWF is common in the northeastern coastal Adriatic, and (ii) if so, what is DWF-related dynamics behaviour, in particular regarding water exchange with the open Adriatic. To answer these questions: (i) the DWF does occur in the coastal northeastern Adriatic during normal winters in terms of wintertime heat losses, as was the case of 2015; and (ii) exchange of waters between coastal and open Adriatic waters through a number of connecting passages has been quantified, with residence time varying from a week to a few weeks and being shorter during strong wind conditions.

There are several observational and modelling findings that are in favour of the DWF occurring in the coastal northern Adriatic, in contrast to eventual advection of dense waters from the "classical" site in the northern Adriatic shelf. First, the DWF did occur in the area through vertical cooling and mixing of the whole water column, as (i) a pool of dense water residing at the bottom of the deepest coastal trenches in early December 2015 – before wintertime cooling events – completely changed the thermohaline structure in late May, and (ii) density of these bottom waters were higher than of the waters residing before the cooling events. Indeed it is known that the coastal northeastern Adriatic is the area where bora wind has its maximum (Grisogono and Belušić, 2009), having rather substantial effects to the ocean through mixing and cooling of the whole water column (Janeković et al., 2014). Next, some of connecting passages between coastal and open Adriatic waters – like these associated with transects T3 and T4 – have a sill, which physically restricts the inflow of open Adriatic dense waters toward the coastal area. Sedmovraće channel in which A2 station has been moored is an exemption, as being even deeper that open and inner waters; yet, there is a physical barrier between inner basin from Sedmovraće to station A1 and Kvarnerić channel, which again hamper the entrance of open Adriatic dense waters towards the deepest parts of the NAdEx region. Finally, the inflow of dense waters towards the Rijeka Bay may occur through Kvarner Bay, which has a very gently slope from its entrance towards the Rijeka Bay. However, the float observations sporadically captured only a few metres thick near-bottom fragments

of the inflowing water having substantial larger density, while ADCP measurements do show the predominance of the outflowing bottom currents in the region. So that, the Kvarner Bay is not the place where dense waters have been found to enter the coastal area. Indeed, the modelled transports in the most of the connecting passages, particularly during strong bora episode, show a substantial transport of dense waters from the coastal area towards the open sea. These results are also in line with previous modelling results by Janeković et al. (2014), who also modelled outflowing dense water transports in connecting passages during the winter of 2012.

According to modelling results, dense water transports during winter of 2015 reached those modelled for winter of 2012 (Janeković et al., 2014). However, density of generated water was much lower in 2015 than in 2012. There are several reasons for this: (i) preconditioning did not include a prolonged period of dry conditions (MHS, 2014, 2015), which may increase salinity of inner coastal waters to values equal to those observed in the open Adriatic (Mihanović et al., 2013), and (ii) wintertime cooling and heat losses were not as pronounced as in 2012, although there were several strong bora events. Although weaker, the DWF of 2015 excited thermohaline circulation in the basin, being detected through an increase of salinity in time (Orlić et al., 2006). Still, the question is whether or not intrusion of the open Adriatic saline waters to the coastal northeastern Adriatic during late winter and spring of 2015 was the result of the open Adriatic thermohaline circulation driven by the DWF in the coastal northeastern Adriatic, or a consequence of the broader open-Adriatic thermohaline circulation. Both circulations may be detected in connecting passages, where the dense water outflow has been dominant, but being almost blocked or even reversed (e.g. see wintertime currents at stations A2 and A9, Fig. 6) in near-bottom layers. A thin (2-5 m) dense water current flew in from the open-Adriatic waters towards the Kvarner Bay. This has been detected by the state-of-the-art measuring platform designed to catch near-bottom dynamics, the Arvor-C profiling float - deployed in the Adriatic Sea for the very first time. All of these findings emphasize the importance of use of state-of-the-art techniques in the Adriatic studies, providing both more details of known processes and capturing some new phenomena that cannot be documented by a classical measuring technique (like CTD or ADCP).

Aside from documentation of processes by measurements, these data may be particularly useful for fine tuning of numerical models, which is a foreseen direction of future investigations in the Adriatic, in both coastal and open ocean areas – particularly as model strongly underestimated measured currents, especially in narrowest connecting passages (Fig. 10). The tuning should include also a densification of sigma layers in the near-bottom layers, where bottom density currents may appear (Vilibić and Mihanović, 2013). For sure, 20 sigma layers in our modelling system were not able to satisfactorily reproduce the observations in the near-bottom layers, like these of Arvor-C profiling float, yet such a number of layers were found to satisfactorily reproduce overall dense water dynamics over the northern Adriatic shelf (e.g. Benetazzo et al., 2014). Estimated variables proportional to currents were also underestimated (e.g. heat, salt, mass and volume fluxes at connecting passages), while residence times were likely overestimated. Yet, thermohaline properties of the coastal area are fairly well reproduced, pointing to the fairness of the model results in reproducing DWF processes and associated dynamics.

Underestimation of water exchange between the coastal and the open Adriatic waters may be a result of several factors. The first and the most obvious one is the horizontal resolution of the ocean model. A resolution of 500 m may not be

high enough to reproduce cross-channel processes occurring in a few kilometres wide channels, such as those approximated by the T2 and T3 transects. This hypothesis is supported by the fact that currents were least underestimated in wider channels and areas, e.g. at T4 and T789 transects. However, as they were still underestimated at latter transects (i.e. corresponding ADCP points), part of the misrepresentation must be due to other factors within the modelling system, like parameterisation of wind effects, vertical diffusivity, lack of wave models, etc.. Surface forcing is the next culprit in the list, since an 8-km mesoscale ALADIN model with a 2-km dynamical downscaling of surface wind might not be sufficient for realistic ocean forcing in such a complex area. This applies to both ocean and atmospheric processes, considering that the bora wind in the area is driven by a complex orography of the Velebit Mountain (Grisogono and Belušić, 2009).

Recent investigations of different types of bora by the TerraSAR-X images (Kuzmić et al., 2015) conclude that different bora types embed a number of small spatial structures in its flow at the kilometre and event sub-kilometre spatial scales. The structures are particularly developed during severe bora outbreaks when large differences between relatively high sea surface temperature (SST) and overflowing very cold air are present. These differences can trigger secondary bora jets and extensive orographic breaking waves that propagate over the entire coastal northeastern area and over much of the open Adriatic, as was the case during the winter of 2012. Last but not least, the role of air-sea feedback during bora events is not negligible, as was shown through comparison of two-way and one-way atmosphere-ocean coupled models (Pullen et al., 2006, 2007; Ličer et al., 2016): air-sea feedback influences position and strength of jet-like structures, it decreases heat losses in the area, and it reduces ocean current in a jet for about 10-20% during bora events, therefore suppressing ocean mixing. In summary, an improvement in reproducing wintertime dynamics in a complex area such as the coastal northeastern Adriatic should be based on: (i) an increase in horizontal resolution of an ocean model, (ii) implementation of high-resolution (1 km horizontal resolution at maximum) non-hydrostatic atmosphere model, and (iii) their two-way coupling. This concept does not include the effects of waves, which are known to affect the DWF at the open Adriatic (Benetazzo et al., 2014); yet, no sensitivity modelling study of waves has been performed for the complex coastal regions, which are characterized by a quite limited fetch and strong deformation of waves due to strong gustiness of bora wind.

To emphasize the importance of the NAdEx 2015 experiment, we should add that preliminary analyses of observations (Vilibić et al., 2016b) reveal a number of interesting phenomena and processes other than those presented in this paper. For example, it seems that dynamics of the thermohaline front that stretches from Kvarner Bay towards the open Adriatic (Lee et al., 2005; Kuzmić et al., 2006; Poulain et al., 2011) is highly variable in time, resembling a near-diurnal wave-like oscillations. A question is if these oscillations are driven by diurnal tides (Cushman-Roisin and Naimie, 2002), Adriatic seiches (Cerovečki et al., 1997), inertial oscillations (Orlić, 1987) or some other phenomena that may induce significant oscillatory currents in the region (Kokkini et al., 2017). In addition, as seen from glider measurements, the front may completely change and even vanish over a daily timescale, and may have a strong impact on the thermohaline and dense water dynamics in the region. Last but not least, high-frequency phenomena have been observed in coastal waters, presumably being a result of inertial, tidal (barotropic and baroclinic), topographic (the Adriatic seiche of 21.5 h) and advective processes strongly influenced by the complex coastal topography. These processes may influence the Adriatic-scale phenomena, similarly as the

exchange between the Venice Lagoon and the Adriatic modulates the Adriatic diurnal tides (Ferrarin et al., 2015). Further investigations of all these processes are envisaged through in-depth analyses of the collected NAdEx 2015 dataset and process-oriented atmosphere-ocean modelling at high resolutions.

**Acknowledgements**: We are indebted to crew members of r/vs *BIOS DVA* and *VILA VELEBITA*, and all researchers, engineers and technicians engaged in data collection through different observing platforms, instrumentations and in data processing. The work has been supported by a number of research and technology projects: ADAM-ADRIA (HRZZ Grant IP-2013-11-5928), CARE (HRZZ Grant IP-2013-11-2831), SCOOL (HRZZ Grant IP-2014-09-5747), ADIOS (HRZZ Grant IP-06-2016-1955), MARIPLAN (HRZZ Grant IP-2014-09-3606), EuroFLEETS-II (FP7 Grant 312762) and by the Euro-Argo-Italy programme.

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

**Table 1. Average temperature, salinity, mass and volume transports at transects T1 to T789, as well as sum of transports at the outer (T1+T2+T3+T4+T789) and inner (T5+T6) boundaries of coastal waters.**

| | $T_1$ | $T_2$ | $T_3$ | $T_4$ | $T_5$ | $T_6$ | $T_{789}$ | $T_{outer}$ | $T_{inner}$ |
|---|---|---|---|---|---|---|---|---|---|
| Salinity ($10^8$ kg s$^{-1}$) | 1.7 | -0.6 | 0.5 | -4.6 | -0.6 | -0.2 | -21.8 | -24.8 | -0.8 |
| Temperature ($10^{11}$ J s$^{-1}$) | 2.4 | -0.8 | 0.7 | -6.3 | -0.8 | -0.2 | -28.4 | -32.4 | -1.0 |
| Mass ($10^6$ kg s$^{-1}$) | 4.6 | -1.7 | 1.0 | -11.3 | -1.7 | -0.5 | -58.2 | -65.0 | -2.1 |
| Volume ($10^3$ m$^3$ s$^{-1}$) | 4.4 | -1.6 | 0.9 | -11.0 | -1.6 | -0.4 | -56.6 | -63.2 | -2.1 |

**Table 2. Temperature, salinity mass and volume sum transports at the outer (T1+T2+T3+T4+T789) and inner (T5+T6) boundaries**
10  **for peak outer transports.**

| | $T_{outer}$ Salinity $10^8$ kg s$^{-1}$ | $T_{outer}$ Temp. $10^{11}$ J s$^{-1}$ | $T_{outer}$ Mass $10^6$ kg s$^{-1}$ | $T_{outer}$ Volume $10^3$ m$^3$s$^{-1}$ | $T_{inner}$ Salinity $10^8$ kg s$^{-1}$ | $T_{inner}$ Temp. $10^{11}$ J s$^{-1}$ | $T_{inner}$ Mass $10^6$ kg s$^{-1}$ | $T_{inner}$ Volume $10^3$ m$^3$ s$^{-1}$ |
|---|---|---|---|---|---|---|---|---|
| 17/12/14 09:00:00 | -113.9 | -186.5 | -299.5 | -291.3 | -1.6 | -1.3 | -4.4 | -4.3 |
| 28/12/14 15:00:00 | -137.4 | -211.4 | -362.3 | -352.4 | -3.8 | -0.4 | -10.1 | -9.8 |
| 17/01/15 09:00:00 | -112.9 | -160.9 | -295.6 | -287.3 | 0.7 | -1.9 | 1.7 | 1.6 |
| 06/02/15 12:00:00 | -113.8 | -142.1 | -298.2 | -289.8 | 0.7 | 4.9 | 1.9 | 1.8 |
| 22/02/15 21:00:00 | -279.3 | -347.2 | -728.6 | -707.9 | -1.5 | -1.5 | -3.9 | -3.8 |
| 05/03/15 09:00:00 | -118.6 | -145.0 | -310.1 | -301.3 | -2.1 | -1.1 | -5.6 | -5.4 |

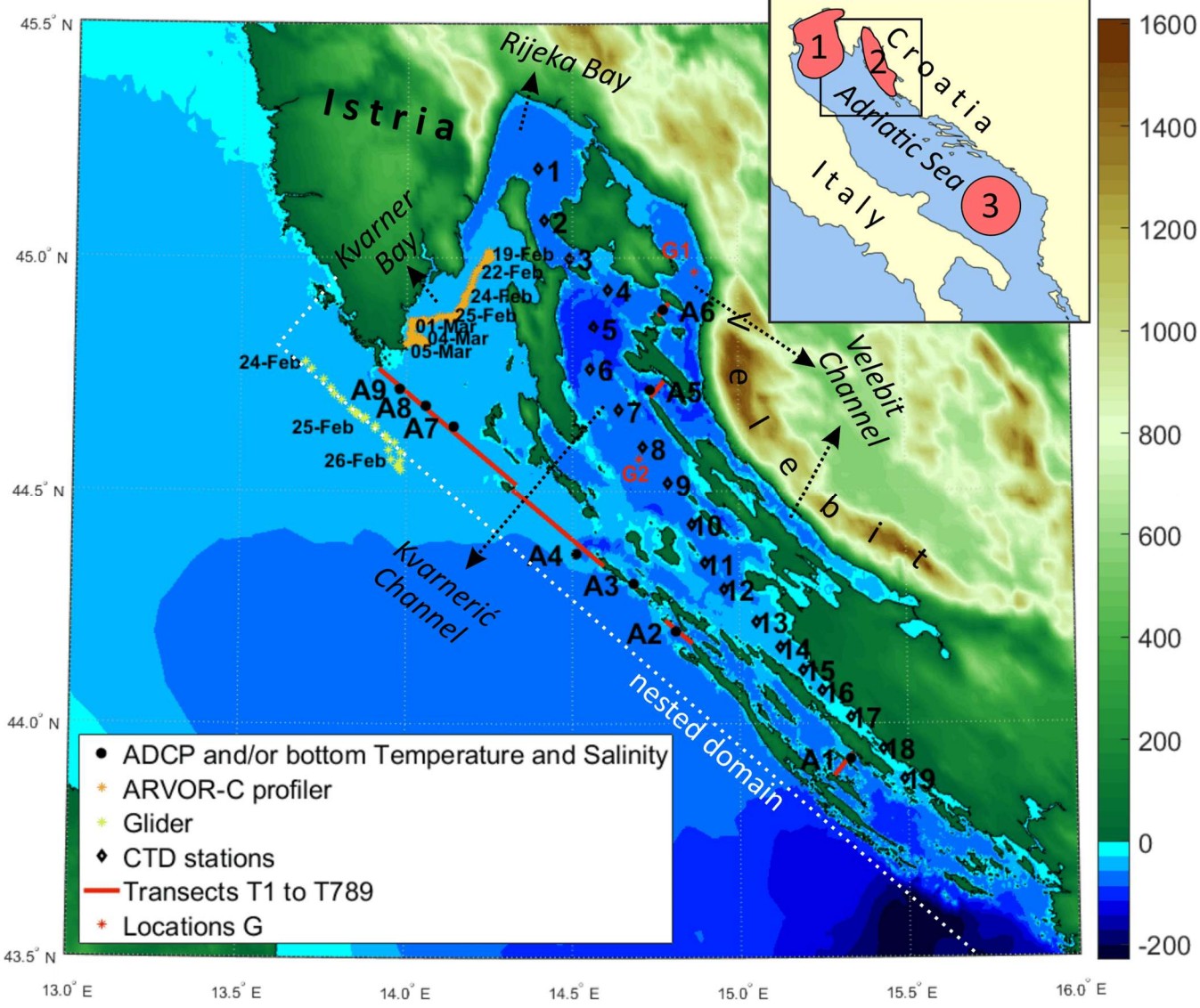

**Figure 1. Geographical position and bathymetry of the coastal northeastern Adriatic, with indicated measurements conducted through the NAdEx 2015 experiment: stations A1 to A9 (black circles) where ADCP/SBE911 were moored at the bottom, stations 1 to 19 (black diamonds) where CTD probe profiling has been executed, Arvor-C profiling (orange stars) and glider profiling (yellow stars). Locations G1 and G2 (red stars) have been used for computation of temporal changes in heat losses, buoyancy changes and thermohaline properties from the modelling system, while the definition of the bora episode has been based on ALADIN/HR wind modelled at G1 location. Transects T1 to T6 and T789 are marked by red lines on which the fluxes and transports have been estimated; the transect labels are associated to the equivalent A station labels. Nested ROMS domain boundary is indicated by dashed white line. Inset numbers 1, 2 and 3 denote the areas where dense water formation is documented in the Adriatic Sea.**

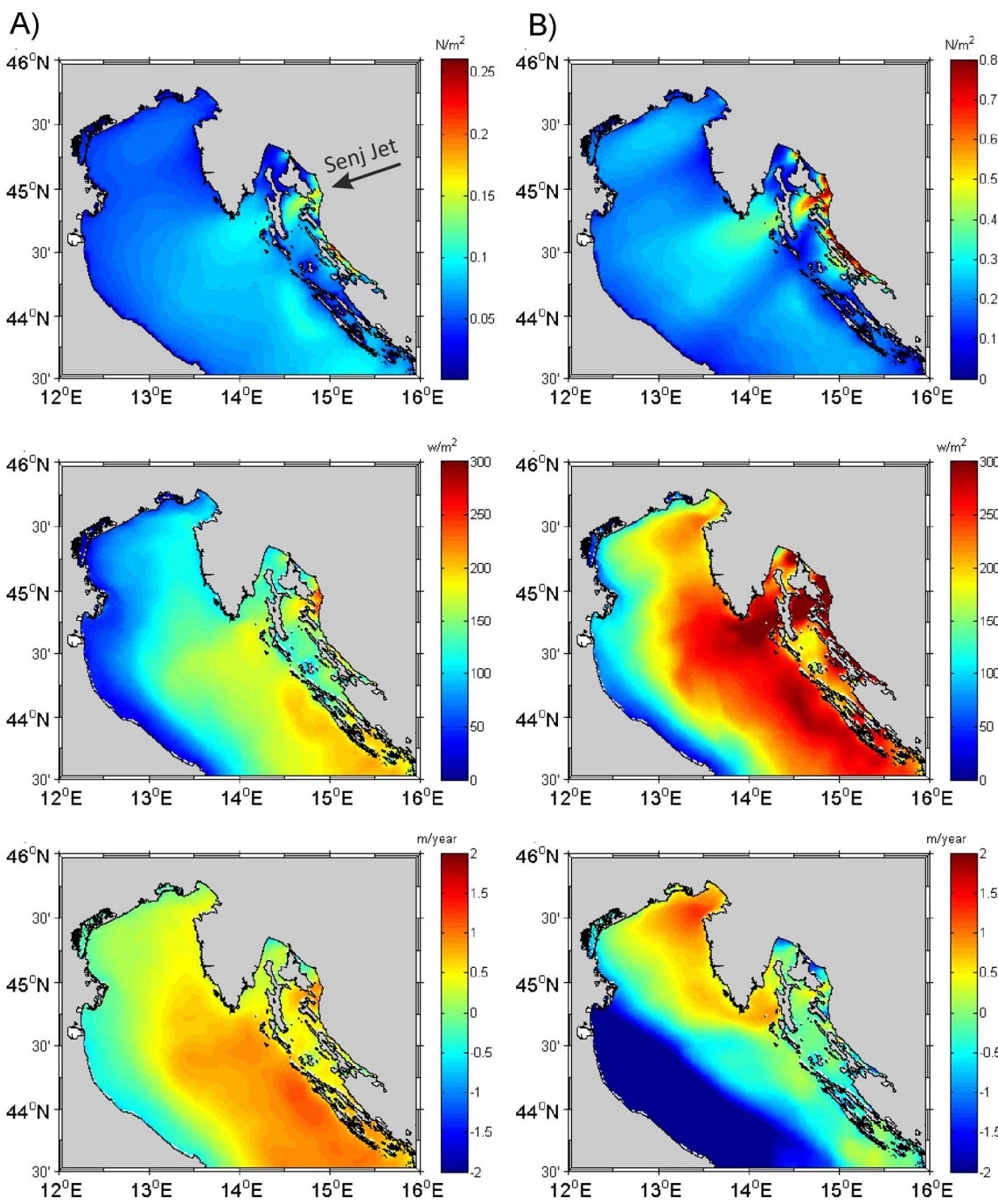

**Figure 2. Wind stress module (top), net heat flux (middle) and water (E-P) flux (bottom) in the middle and northern Adriatic averaged (a) between 15 January and 15 March 2015, and (b) over all bora episodes occurring between 15 January and 15 March 2015. Bora episodes are defined by the wind speed higher than 15 m/s blowing along the major wind axis (from ENE) at G1 location.**

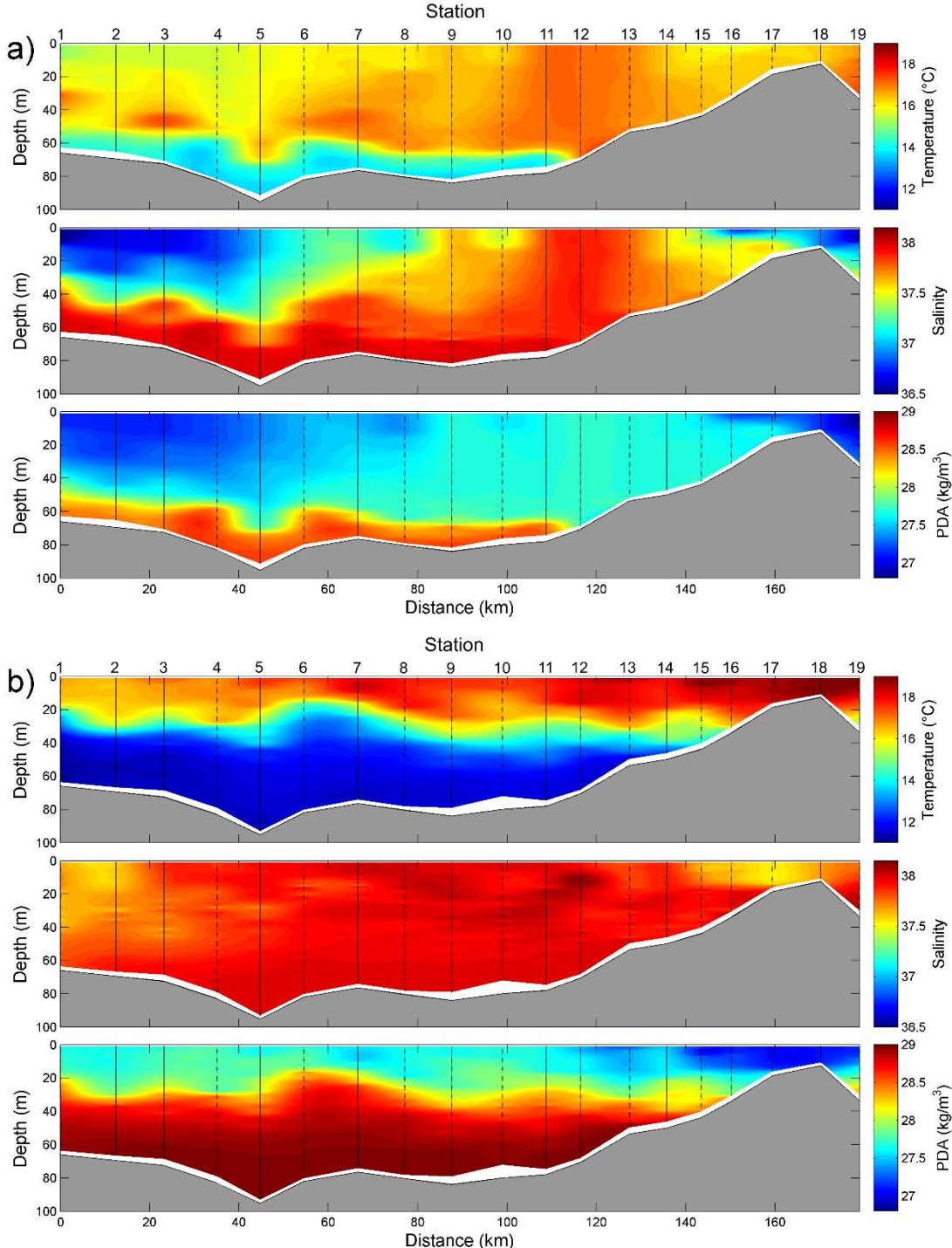

**Figure 3. Temperature, salinity and PDA values measured at the Kvarnerić transect during (a) Leg 1 cruise between 3 and 6 December 2014, and (b) Leg 2 cruise between 26 and 29 May 2015.**

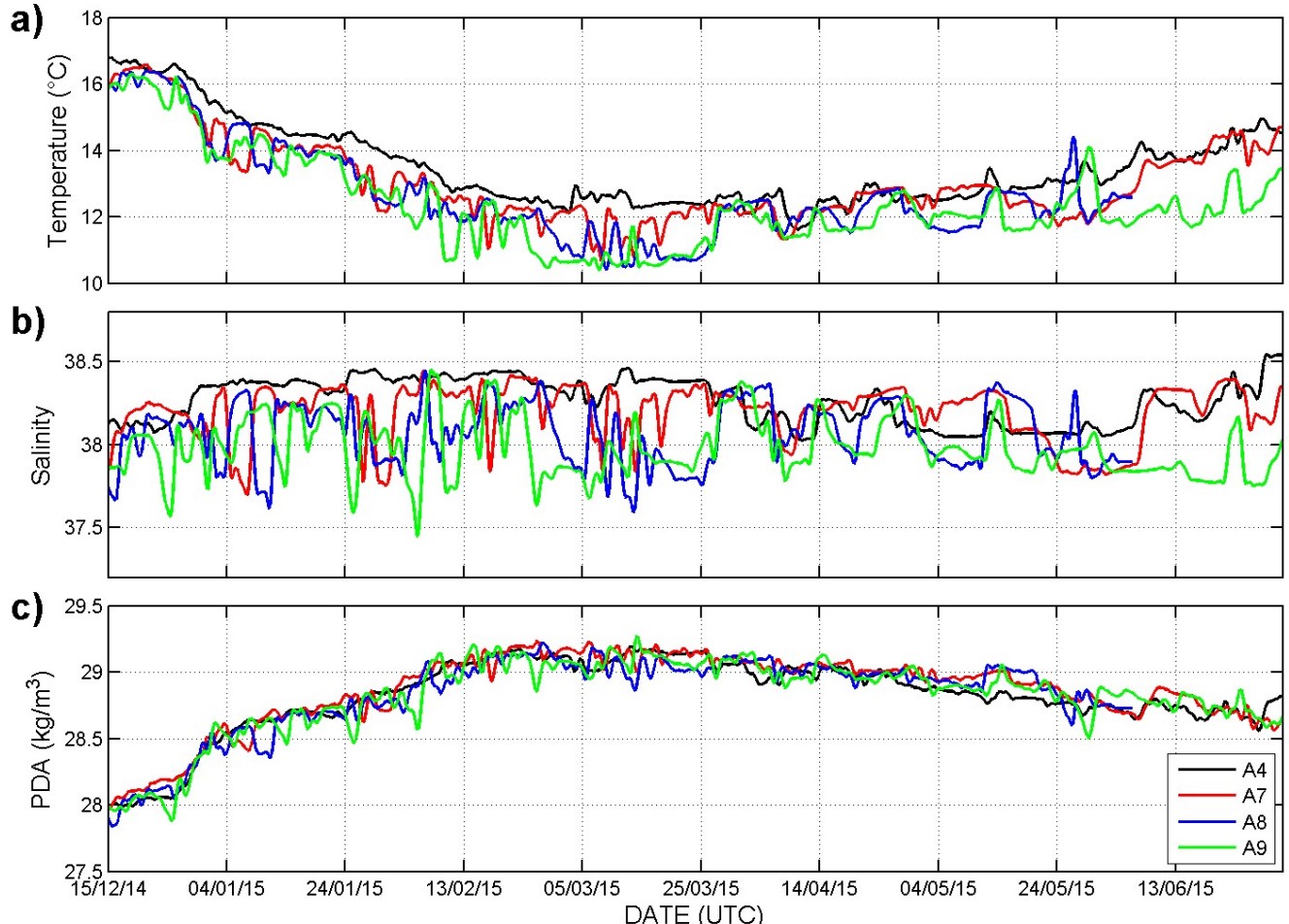

5    Figure 4. (a) Temperature, (b) salinity and (c) PDA series measured at the bottom of stations A4, A7, A8 and A9. The series are
filtered by low-pass Kaiser-Bessel filter with cut-off period at 33 h.

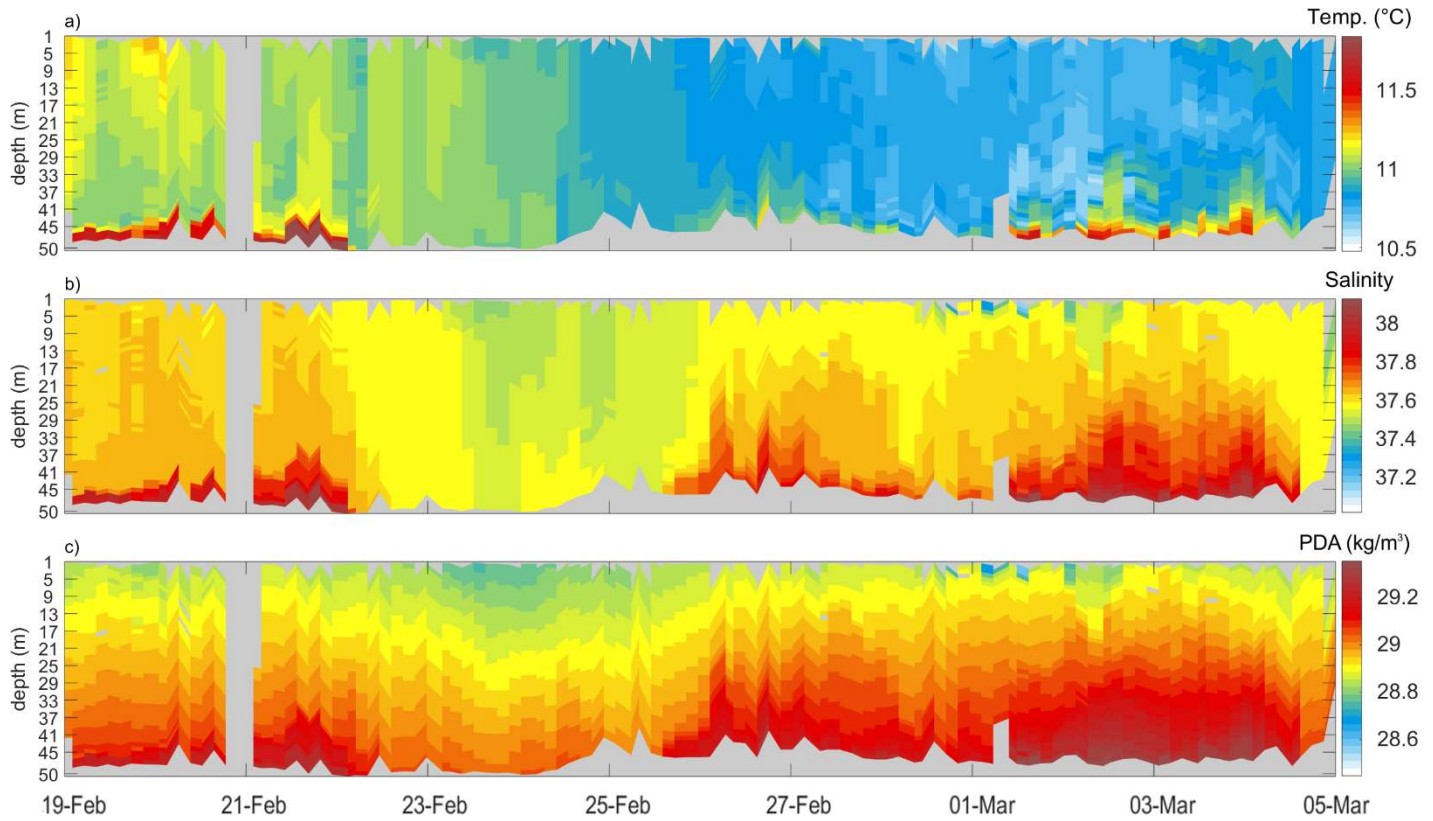

**Figure 5. Temperature, salinity and density values measured between 19 February and 5 March 2015 by Arvor-C profiling float in the Kvarner Bay.**

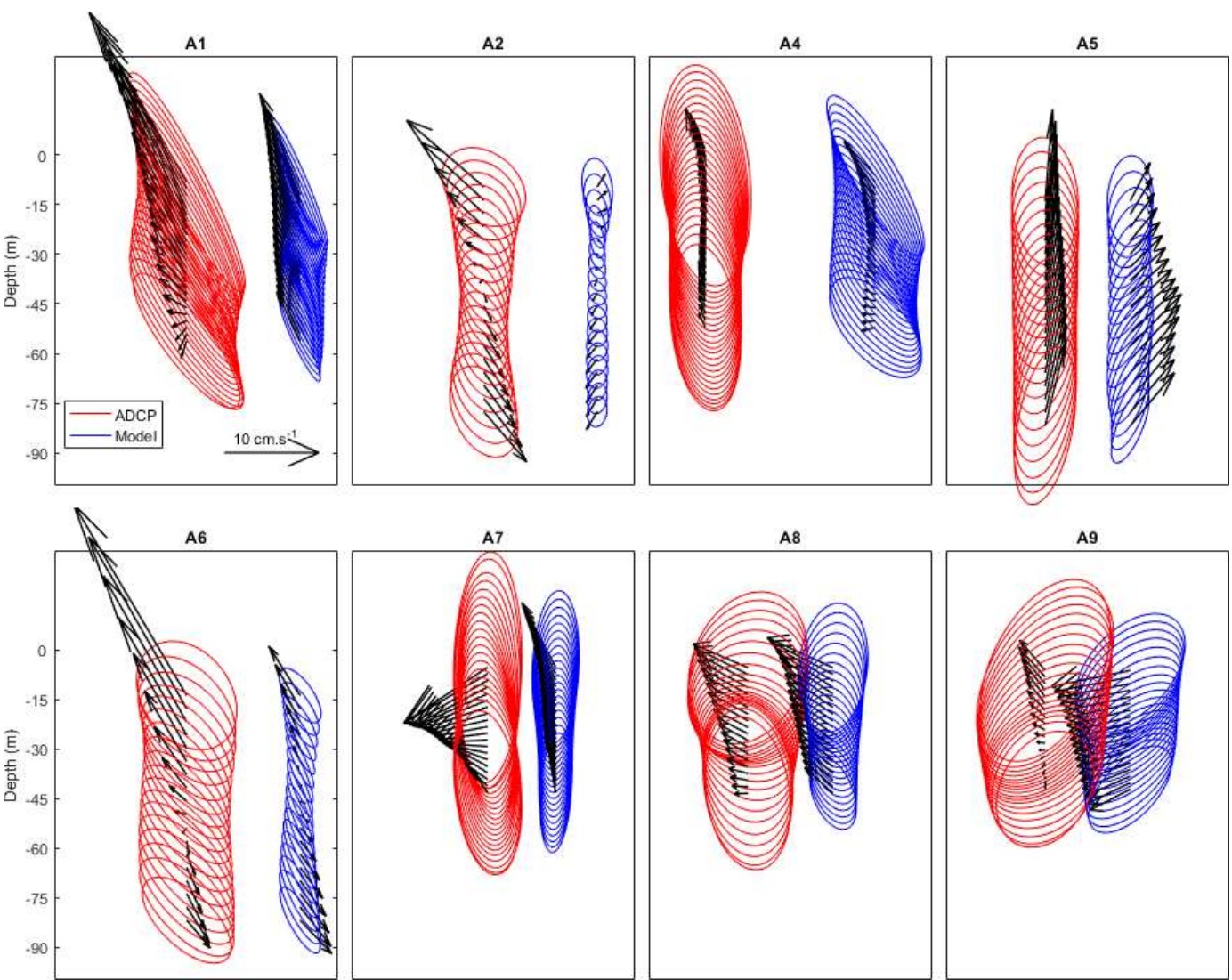

**Figure 6. Mean residual currents and ellipses of standard deviations measured (red) and modelled (blue) at A1 to A9 stations between 1 February and 1 April 2015.**

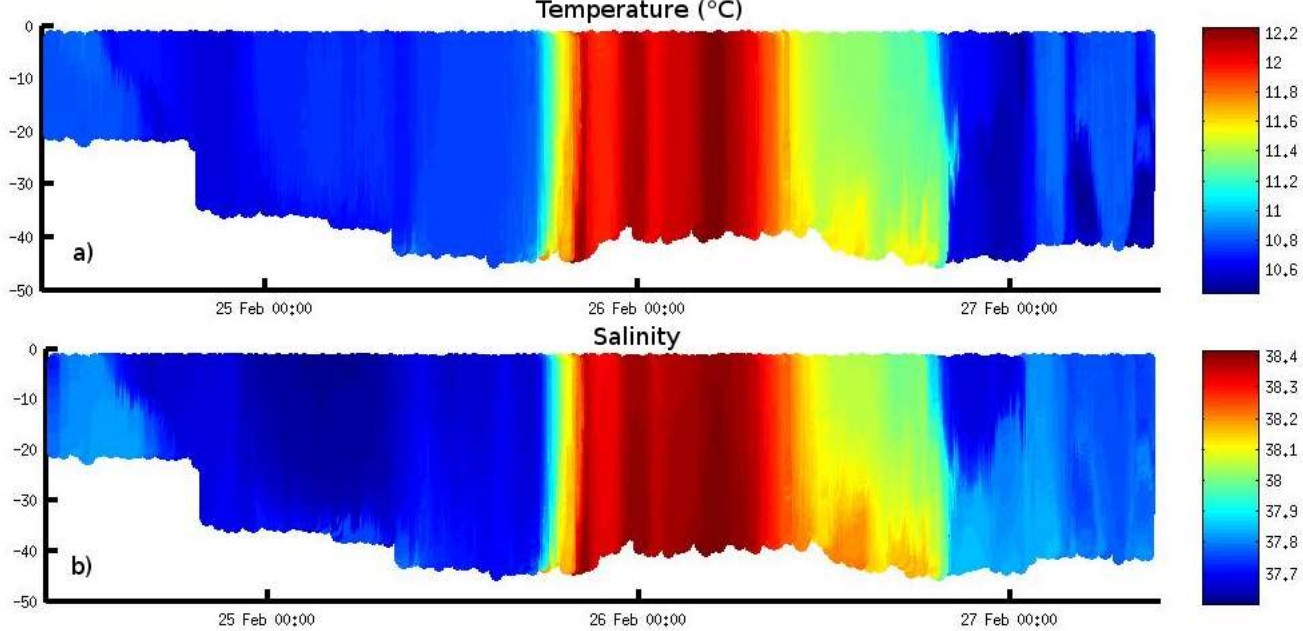

**Figure 7. Temperature and salinity profiles measured by Slocum glider between 24 and 27 February 2015 in front of the Kvarner Bay. Glider trajectory is plotted in Fig. 1.**

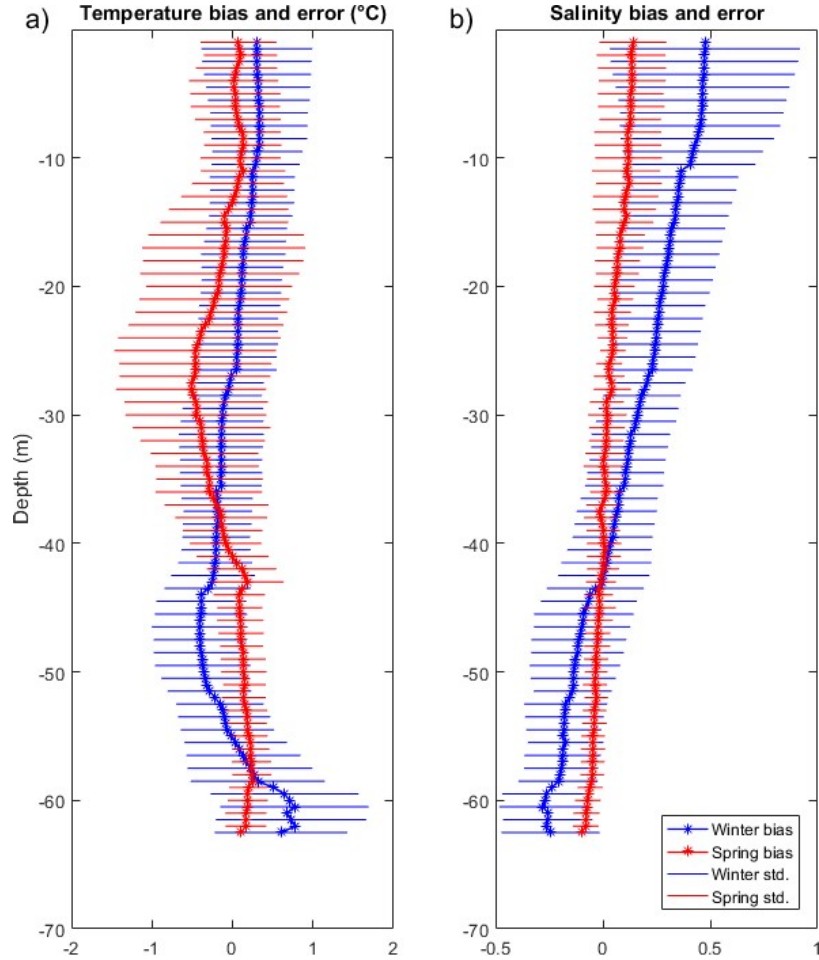

**Figure 8. Vertical profiles of the bias and root-mean-square-error during Leg 1 (December 2014) and Leg 2 (May 2015) cruises.**

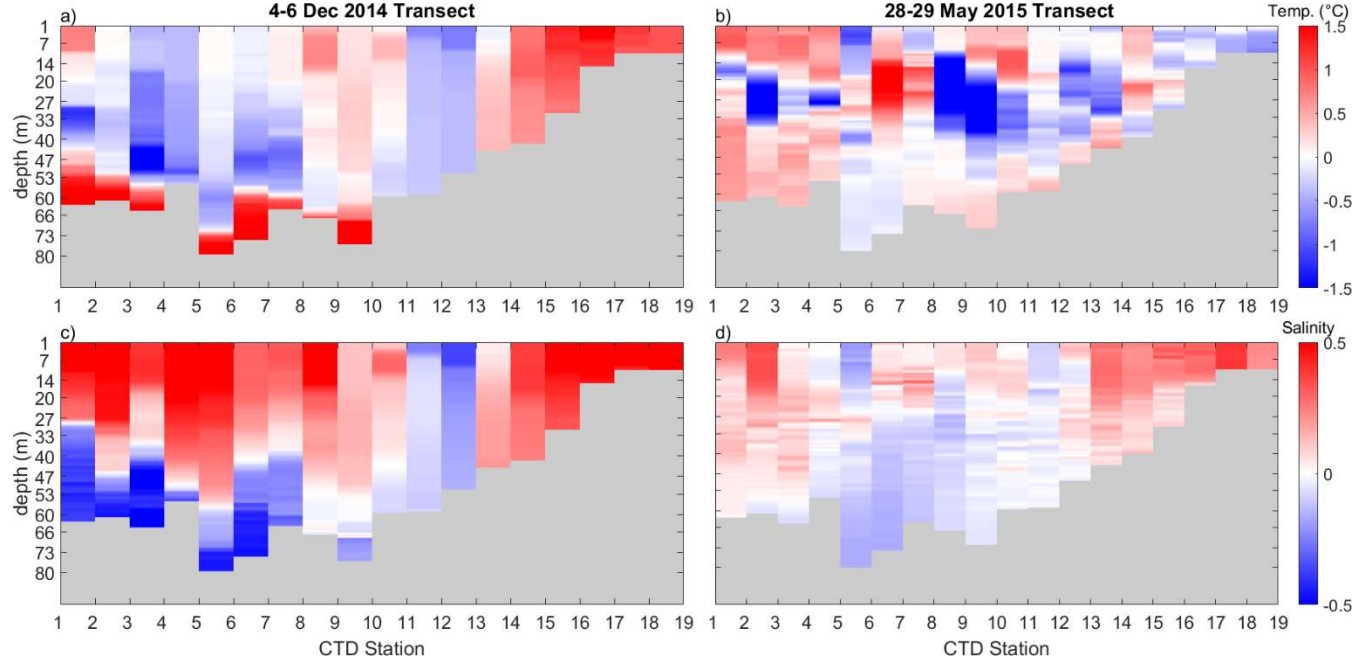

**Figure 9. Model-to-observation difference in (a, b) temperature and (c, d) salinity estimated for (a, c) Leg 1 cruise and (b, d) Leg 2 cruise CTD data.**

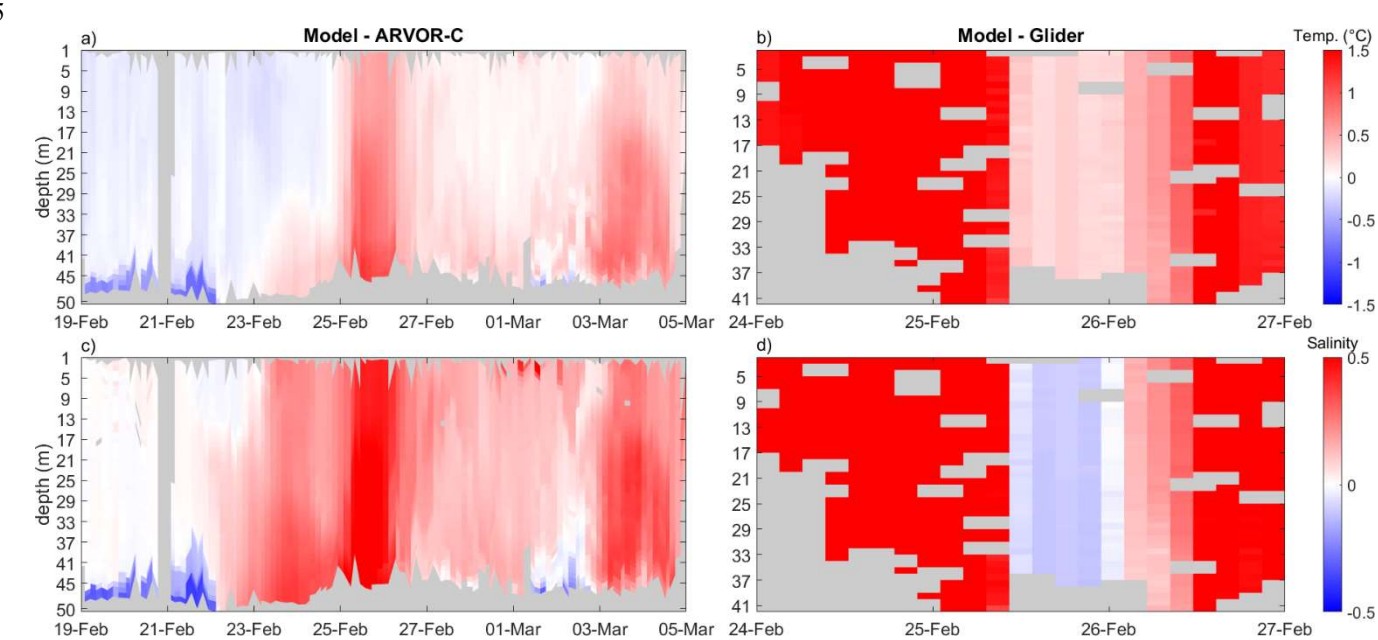

**Figure 10. Model-to-observation difference in (a, b) temperature and (c, d) salinity estimated for (a, c) Arvor-C float and (b, d) glider measurements.**

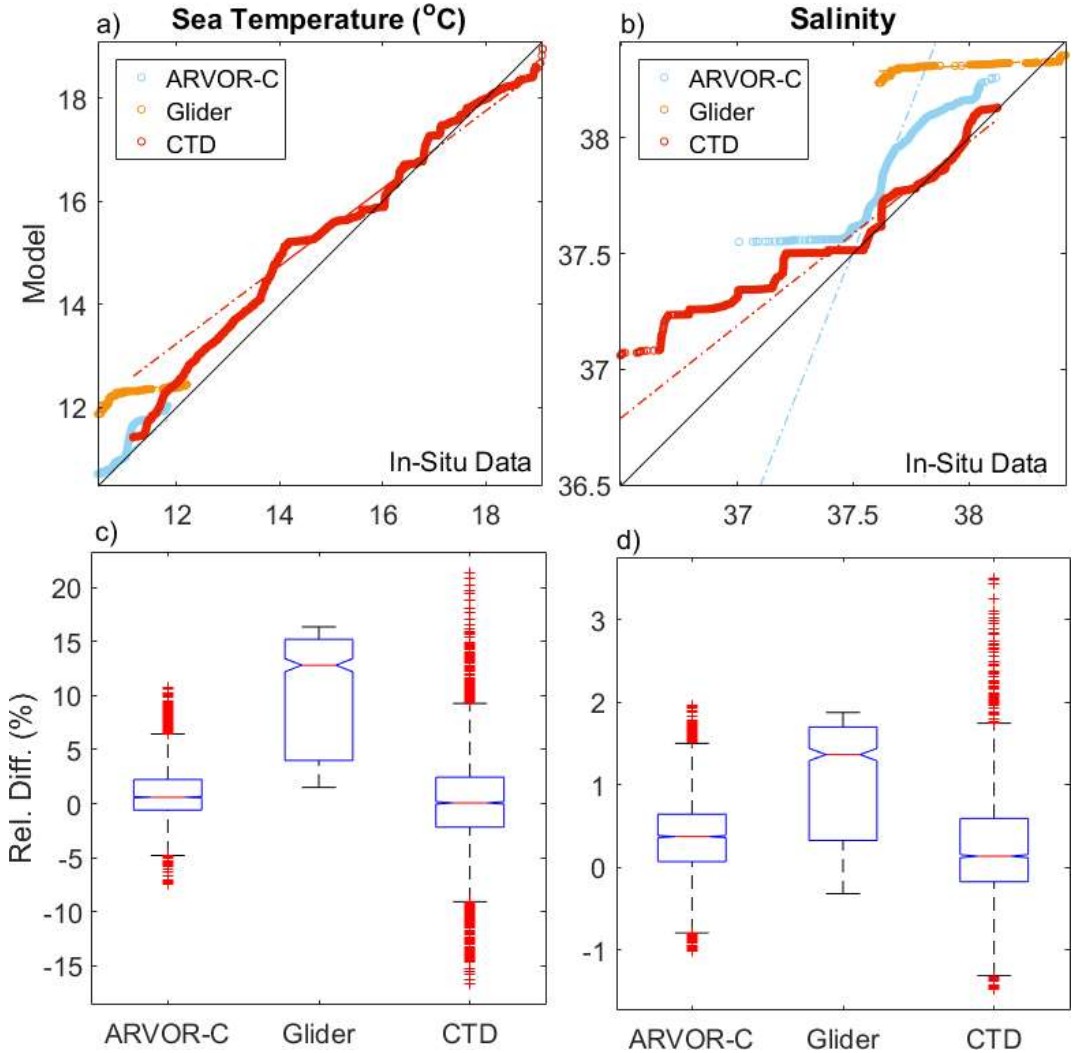

5    **Figure 11. Model-to-observation (a, b) Q-Q plots and (c, d) Box-Whiskers plots of (a, c) temperature and (b, d) salinity measured by CTD, Arvor-C float and glider. Dash-dot lines represent the Q-Q slope of a particular dataset.**

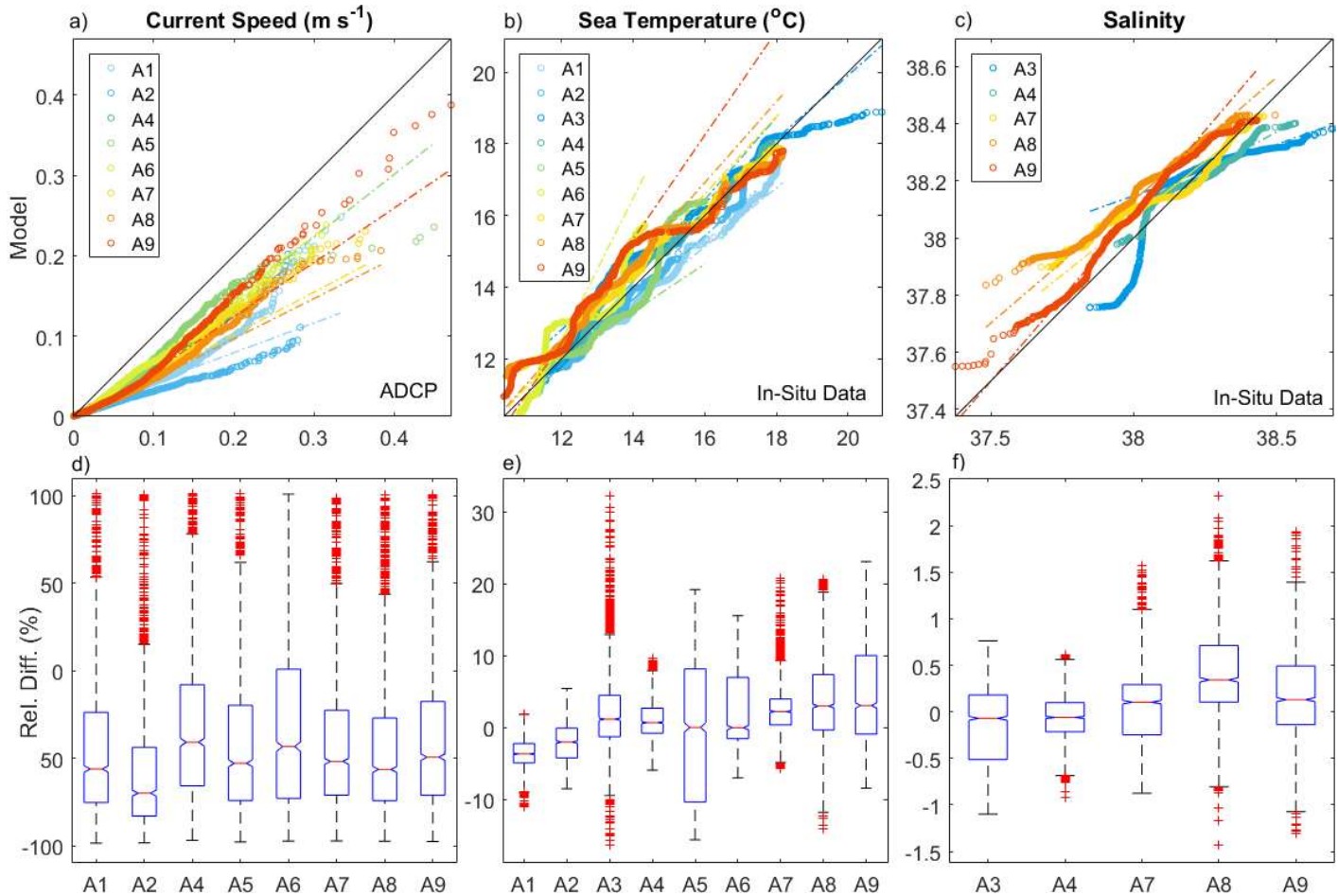

**Figure 12. Model-to-observation (a, b, c) Q-Q plots and (d, e, f) Box-Whiskers plots of (a, d) current speed, and (b, e) temperature and (c, f) salinity measured at the bottom of stations A1 to A9. Dash-dot lines represent the Q-Q slope.**

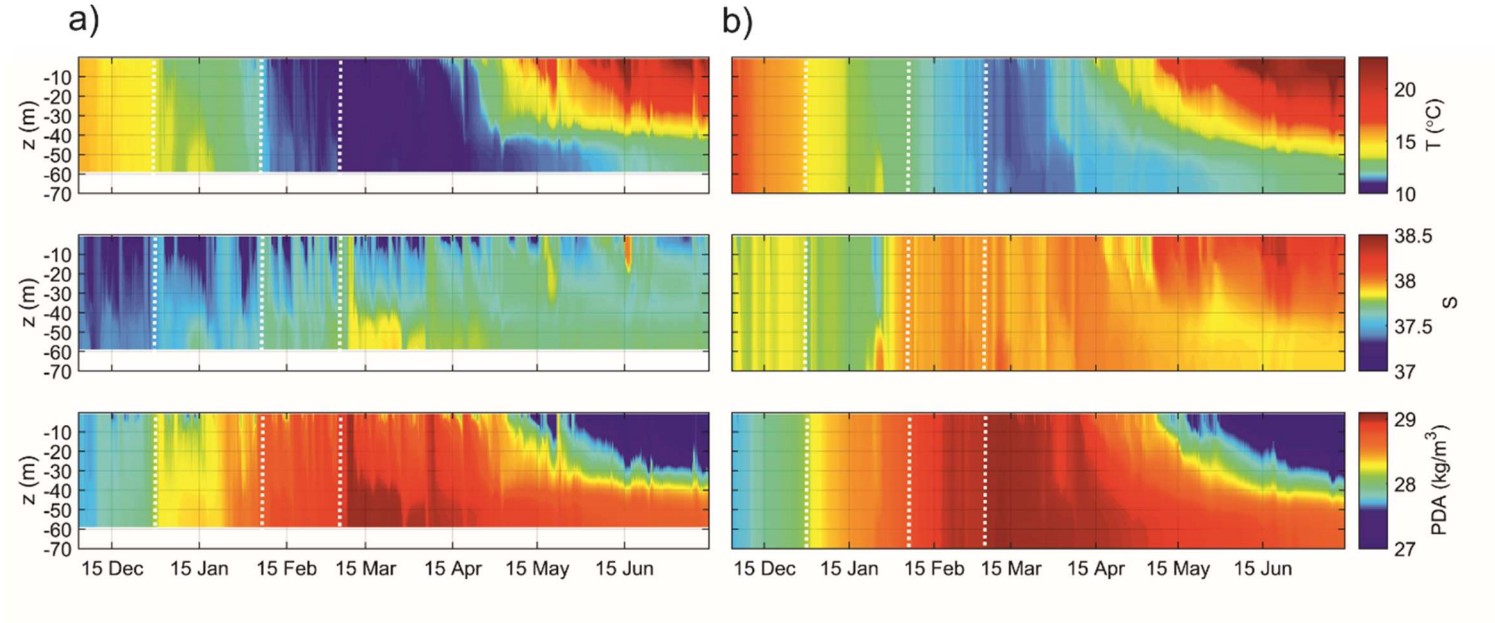

**Figure 13. Hovmöller plot of modelled temperature, salinity and PDA values at grid points (a) G1 and (b) G2. Vertical dashed lines indicate three severe bora episodes.**

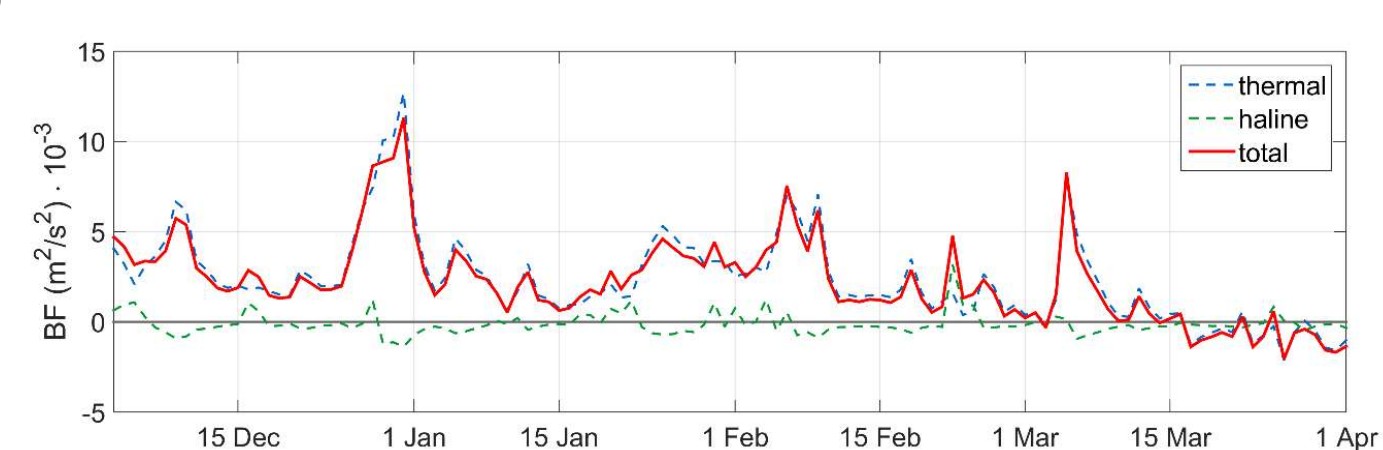

**Figure 14. Daily values of surface buoyancy fluxes (BF) and its components averaged over the nested model domain.**

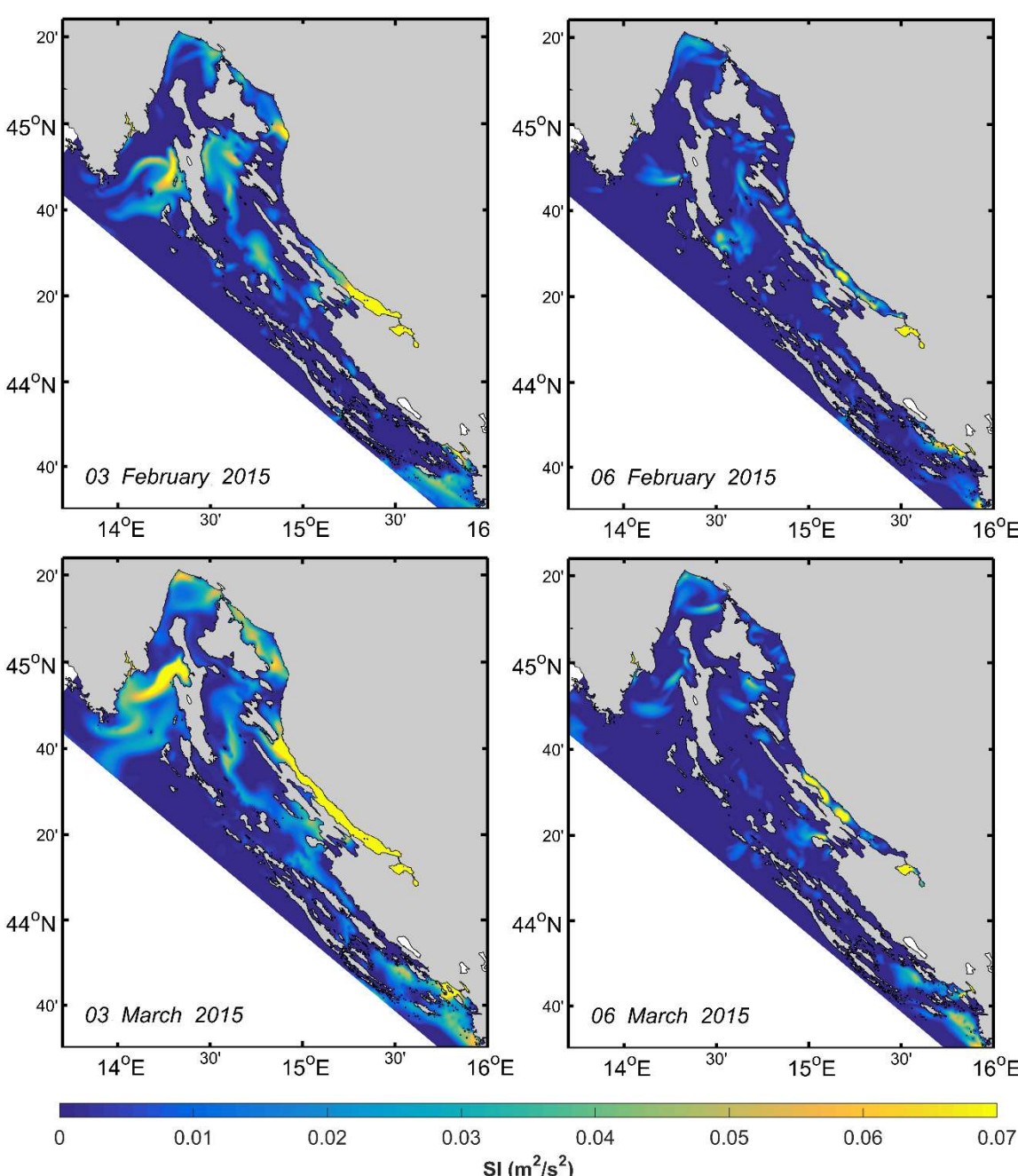

**Figure 15. Stratification index (SI) computed for the nested model domain prior and after the major bora episodes: 3 and 6 February 2015, and 3 and 6 March 2015.**

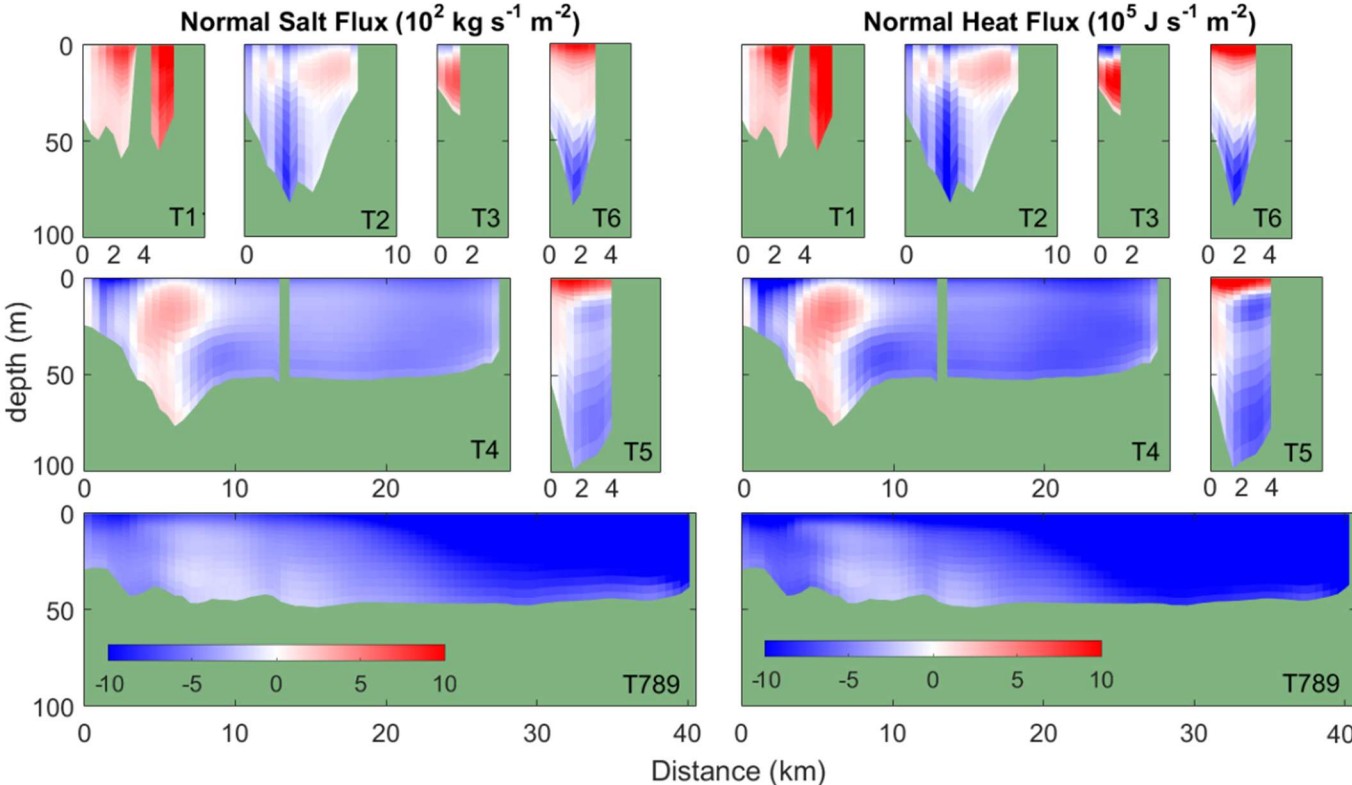

**Figure 16. Modelled heat and salinity fluxes normal to transects T1 to T789 and averaged between 15 December 2014 and 15 May 2015. Positive fluxes are oriented northeastward over the T2, T3, T4 and T789 transects and northwestwards over the T1, T5 and T6 transects. Green areas denote the bathymetry.**

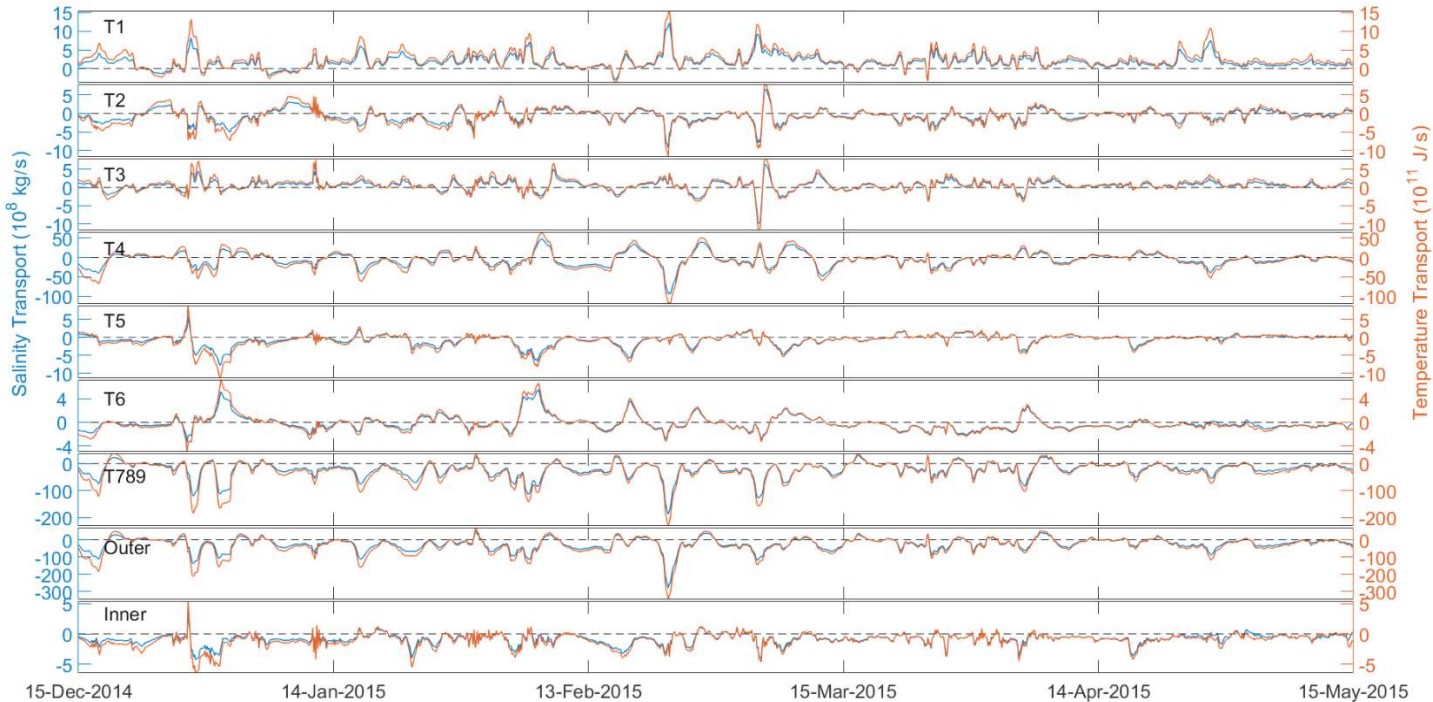

**Figure 17. Time series of heat and salinity transports normal to and integrated over transects T1 to T789 between 15 December 2014 and 15 May 2015. Transports across the outer (T1+T2+T3+T4+T789) and inner (T5+T6) domains are plotted too.**

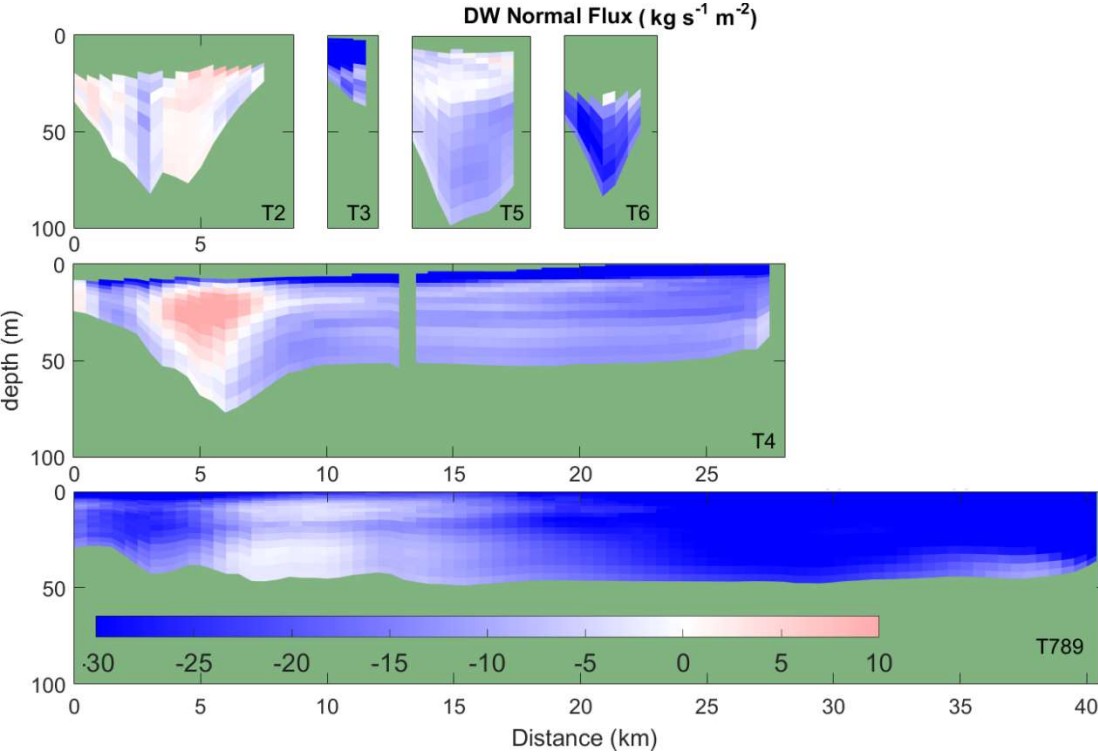

**Figure 18. As in Fig. 16, but for dense water mass fluxes. Green areas denote the bathymetry and model cells where no dense water is modelled. T1 is omitted from the figure, as almost no dense water transport occurred there.**

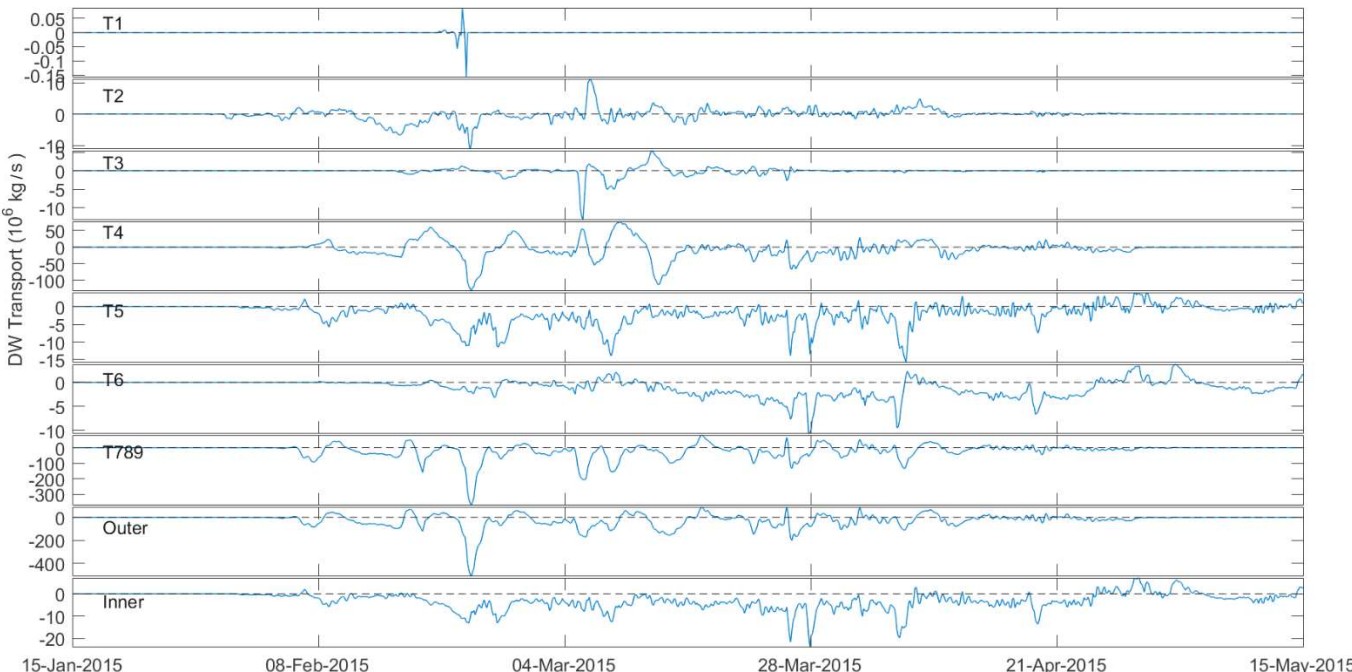

**Figure 19. As in Fig. 17, but for dense water mass transports.**

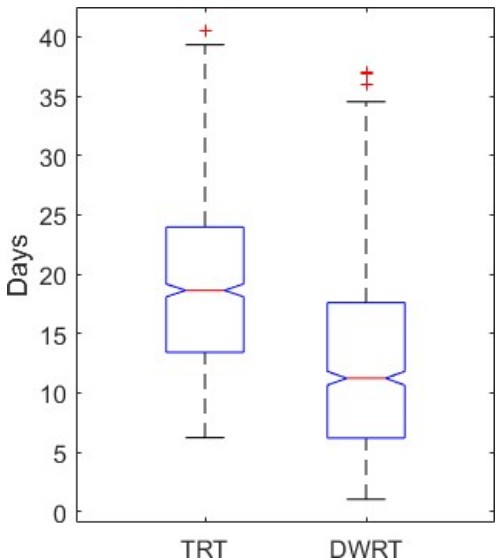

**Figure 20. Box-whiskers statistics of total (RT) and dense water (DWRT) residence times for the coastal northeastern Adriatic bordered by transects T1, T2, T3, T4, and T789.**

