# Peer review of "Dense water formation in the coastal northeastern Adriatic Sea: the NAdEx 2015 experiment"

_Ocean Science, 2017_

## Referee Comment (RC1) · Anonymous Referee #1 · 29 Mar 2017

This is a review of the manuscript "Dense water formation in the coastal northeastern Adriatic Sea: the NAdEx 2015 experiment" by Vilibic et al. The paper provides a modelling and observational study focusing on dense water formation (DWF) in the Northwestern Adriatic Sea. The main scientific questions formulated by the authors are: 1. is DWF frequent or exceptional in this basin ? 2. What are the thermohaline changes associated ? 3. What are the exchanges between this basin and the open-sea Adriatic, and therefore its importance for the Adriatic thermohaline circulation ? 4. What are the associated mesoscale and frontal processes ?  For that purpose, the study focuses on the well-documented 2014-2015 case study based on the NADEX experiment. The relevance of the observing design and of the numerical modelling tool to address those scientific questions is appreciated. The problematic is also well-introduced with a comprehensive bibliography, and the general quality of writing is good.

Out of the 4 main scientific questions, the authors actually address questions 1 to 3, the mesoscale and frontal processes being omitted. However, major concerns arise from the irrelevance of many diagnostics with respect to the authors' commentaries and conclusions. As a consequence, several key conclusions of the study are not demonstrated at all or fragile. I therefore recommend a major review addressing the following points:

- the authors give limited support that winter 2014-2015 was milder than average, Fig.2 not being adapted to this purpose ;

- they do not provide a clear demonstration that DWF occurred within the NADEX area, with respect to the hypothesis that dense waters were imported to this area from the open-sea Adriatic. The lack of any mixed layer depth estimate is a major drawback in the characterization of DWF.

- the velocity section doesn't illustrate, contrary to the authors' claim, that lateral exchanges at the basin boundaries are mostly baroclinic with outcoming waters at the surface and incoming waters at depth ;

- the model evaluation omits major elements of observed variability: is the bottom temperature decrease and density increase during winter trend accurately represented ? Does the model have a baroclinic circulation structure at the transects with incoming waters at depth and outcoming waters at the surface, at least at A2 and A7 locations ?

- the authors don't convincingly show that in the model, DWF occurs locally in the NADEX area and results from Bora wind events. The lack of any mixed layer depth estimate is again a major drawback in the characterization of DWF.

- a major concern arises in the methodology used to calculate residence times within the basin: by separating along-basin and cross-basin fluxes, the authors largely over-estimate the residence time which should include altogether all boundary fluxes. Also, the probability density function is not defined and probably not adapted (too noisy) to

get a robust residence time estimate: instead, a temporal mean residence time estimate would probably give a more convincing result.

Here is a detailed description of major concerns.

M1 There are too many plots: Fig.8b is unnecessary, especially because glider data is not shown, and an average bias for both temperature and salinity would be sufficient ; model-observation comparisons could easily be merged with observation-only plots ; Fig.9 and 10 are probably unnecessary (see comments below).

M2 Fig.2 doesn't prove that winter 2014-2015 was milder than average. For that, an anomaly with respect to an interannual mean should be displayed. Also, a computation of the thermal and saline buoyancy flux at the surface would help to relate surface conditions to DWF. The only support for a mild winter is given with percentiles of surface temperature. However, several points are not clear: the percentiles are computed with respect to what period ? A 30-year climatology ? Is it the near-surface atmospheric temperature over the sea or the sea surface temperature ? Also, the presence of mild periods doesn't prove that the whole winter was on average milder: for that, a time average needs to be calculated.

M3 Fig.3 is valuable but it doesn't prove that DWF occurred within the NADEX area (which corresponds to the nested ROMS area) as the dense waters could also have been imported from the open-sea. The same is true for Fig.4 as the dense and salty anomaly could have been imported from the lateral boundaries. Finally, Fig.5 doesn't prove either that DWF occurred in the NADEX area. Unfortunately, no mixed layer depth was computed in this study, but using a classical density anomaly criterion of 0.01 to 0.05kg/m$^3$ with respect to surface density (e.g. De Boyer Montegut et al 2004, Houpert et al 2015) would show clearly that the mixed layer is shallow throughout the transect. A comparison of the Stratification Index at the bottom with typical surface buoyancy fluxes modelled by ALADIN could be an approach to suggest (but not demonstrate) that DWF occurred during the winter.
M4 Fig.6 doesn't illustrate a general incoming of dense water at depth and outcoming of lighter waters at the surface. Contrary to the authors' statement, only stations A2 and A7 show a predominantly baroclinic vertical structure of velocity. The stations A3, A4 and A7 to A9, which cover the main connecting section between the NADEX area and the open-sea also show a strong barotropic flow. The interpretation of this figure should be more cautiously done.

M5 The model shows a large temperature and salinity bias in the open-sea Adriatic which therefore comes from the low-resolution ROMS model. It should be at least documented and interpreted, or at best reduced in order to produce realistic lateral exchanges between the NADEX and open-sea areas.

M6 If 20 sigma levels are not sufficient to simulate dense water transformations in the NADEX area (the bottom dense water layer is not modelled), how many would be necessary ? What was the rationale in the choice of only 20 sigma levels when regional and coastal Mediterranean Sea models exhibit typically between 50 and 100 vertical levels (e.g. Mediterranean Sea reanalyses in Pinardi et al 2013 and Hamon et al 2016) ?

M7 Fig.10 gives a too raw evaluation to be insightful because there is no spatio-temporal information, and it should be removed as it is currently. Some elements of observed bottom hydrology and velocity profiles were noted in the observations section and not even commented in the model evaluation section. In particular: is the bottom temperature decrease and density increase trend accurately represented ? Is the salinity front also visible at the same area ? Does the model have a baroclinic circulation structure with incoming waters at depth and outcoming waters at the surface, at least at A2 and A7 locations ? By the way, it would have been insightful to also evaluate the current directions.

M8 Fig.11 is also too raw, it could be improved to relate hydrological transformations to DWF and Bora wind events. Once again, no mixed layer depth was computed and the

large range in the colorbar makes it impossible to identify the period of DWF: it would require to read vertical density variations of typically 0.01kg/m$^3$. The Bora wind was mentioned to explain the stepwise cooling, but a comparison between the vertically-integrated heat loss and the surface heat flux would give a much more convincing relation between cooling and Bora events.

M9 Without mentioning the depth at which the SI is computed, it is impossible to interpret Fig.13. Assuming that the SI was computed at the bottom of the water column for each location, Fig.13 could have provided an SI-based mixed layer depth map (threshold at 0.01m$^2$/s$^2$), already used in several Mediterranean studies (e.g. Waldman et al 2017). The link with the Bora event is not clear because: 1. most of the domain already has an SI<0.01m$^2$/s$^2$, the signature of DWF, before the Bora occurs ; 2. only the comparison between the ocean SI loss and the integral surface buoyancy flux would allow to attribute buoyancy losses to the Bora. As a consequence, the authors' conclusion of 6.1 has not been demonstrated.

M10 In Fig.18 the methodology to compute the probability distribution of residence time is not explicited. I assume the residence time is computed for each model day and the probability distribution is therefore temporal and daily based on the winter period (which period ?) Results are very noisy which makes them difficult to interpret: I strongly recommend to compute a temporal mean residence time and not a probability distribution function in order to increase the robustness of results. Also, even though it is interesting to differentiate between along-basin and cross-basin fluxes, the residence time of a water parcel is impacted by all boundary fluxes, which cannot be considered separately: by doing so, the authors artificially increase the residence time. I therefore recommend the authors to compute a residence time including all boundary fluxes, and to compare along-channel and cross-channel fluxes without deducing residence times from them.

Here is a description of minor concerns:

Abstract

m1 Do not use the term one-way coupling. There is no feedback of the ocean models toward the atmospheric model, therefore there is no ocean-atmosphere coupling.

2.3.

m2 What is the consistency between the sea surface temperature at the boundary of the forcing atmospheric model and that of the ocean model ?

m3 What is the time resolution of the atmospheric forcing and how does it impact the representation of extreme heat fluxes associated with Bora events ?

m4 Be more explicit about how bathymetry smoothing ensures the run stability.

m5 The mention of an operational integration for the large-scale ROMS model usually implies that data assimilation was done within the model domain. Is it the case ?

m6 Was the nesting 2-way (coupling between both ocean models) or 1-way (forcing of the nested model by the large-scale model) ? If it was 1-way, do you believe that the absence of any feedback toward the large-scale model alters the estimated lateral transports at the boundary of the NADEX domain ?

3.

m7 Missing mention top/middle/bottom in Fig.2.

m8 No critical assessment of the gust wind event was made. Which is the Bulk turbulent flux formulation used in the ALADIN configuration ? Most state-of-the-art Bulk formulations have not been calibrated for winds >20m/s because of the scarce observations (e.g. Fairall et al 2003), which makes them unrealistic and divergent between each other at such wind regimes. Therefore interpretations of ALADIN outputs should be more cautious and critical.

4.

m9 Regarding the presence of a salinity front between A7 and A9, the large high-frequency variability gives some doubts about the significance level of such a signal. Could you provide it please ?

5.

m10 Quantify the model bias decrease between the winter and spring cruises.

m11 Comment the persistent low salinity bias around station 18.

m12 When assessing the model bias, it is clearer to display the difference model minus observations, so that Fig.7 to 10 can be interpreted directly as model bias.

m13 Fig.9 is unnecessary as it reiterates the evaluation done in Fig.7 and 8, without a clear added-value in the interpretation of using Q-Q plots or box whiskers diagnostics. Also, the mention of an ideal distribution in the Q-Q plots is unclear, as all observations should fall within this slope 1 line (e.g. Herrmann and Somot 2008).

m14 It would be more relevant to compare temperature and salinity biases to their variability for instance, rather than to their absolute value, in order to compare biases to the typical hydrological variability in the area.

6.1

m15 Fig.12 shows repetitive timeseries: instead, an area-averaged surface buoyancy flux timeseries (in $m^2/s^3$/day, and not $m^2/s^3$) would give qualitatively the same result, but more integrated. The difference between locations should be documented quantitatively (how does each location compare to the area-average ?) in the text, without displaying the 7 plots.

6.2.

m16 The so-called temperature flux should be named heat flux.

m17 Specify when mentioning Fig.14 and in its legend that it is modelled results.

m18 Fig.14 is misleading because of a factor up to ~30 between transect widths: a constant scaling would be more appropriate to compare heat and salt fluxes. Same comment for the vertical scaling.

m19 How do you interpret that the NADEX area is losing heat through its lateral boundaries despite the stronger Bora heat loss than in the open-sea (Fig.2 middle panels) ?

m20 It is not intuitive that dense water mass fluxes should be directed outward as Fig.3 suggests that the densest waters with a high salinity signature came in from the open Adriatic and were not formed locally. How do you explain this appearent discrepancy between the model and observations ?

m21 The interpretation of some high-frequency transport variability in Fig.15 being driven by Bora events is interesting and could be tested by calculating the transports induced by Ekman currents.

m22 How do you interpret that the residence time for dense waters is lower than that for the total volume, when it is visible from Fig.6 that in observations (but the same is probably true for the model), velocities are lowest in the deep layers where dense waters are located ? One should therefore expect a longer residence time for those dense waters.

References mentioned in this review:

Clément de Boyer Montégut, Gurvan Madec, Albert S Fischer, Alban Lazar, et Daniele Iudicone. Mixed layer depth over the global ocean: An examination of profile data and a profile-based climatology. Journal of Geophysical Research: Oceans, 109(C12), 2004.

CW Fairall, Edward F Bradley, JE Hare, AA Grachev, et JB Edson. Bulk parameterization of air-sea fluxes: Updates and verification for the coare algorithm. Journal of climate, 16(4): 571–591, 2003.

M. Hamon, J. Beuvier, S. Somot, J.-M. Lellouche, E. Greiner, G. Jordà, M.-N. Bouin, T.

Arsouze, K. Béranger, F. Sevault, C. Dubois, M. Drevillon, et Y. Drillet. Design and validation of medrys, a mediterranean sea reanalysis over the period 1992-2013. Ocean Science, 12(2):577–599, 2016. doi: 10.5194/os-12-577-2016. URL http://www.ocean-sci.net/12/577/2016/.

M. Herrmann et S. Somot. Relevance of ERA40 dynamical downscaling for modeling deep convection in the Mediterranean Sea. Geophys. Res. Lett., 35(L04607), 2008. doi: 10.1029/2007GL032442.

L. Houpert, X. Durrieu de Madron, P. Testor, A. Bosse, F. D'Ortenzio, M.N. Bouin, D. Dausse, H. Le Goff, S. Kunesch, M. Labaste, L. Coppola, L. Mortier, et P. Raimbault. Observations of open-ocean deep convection in the northwestern mediterranean sea: Seasonal and interannual variability of mixing and deep water masses for the 2007-2013 period. Journal of Geophysical Research: Oceans, pages n/a–n/a, 2016. ISSN 2169-9291. doi: 10.1002/2016JC011857. URL http://dx.doi.org/10.1002/2016JC011857.

N. Pinardi, M. Zavatarelli, M. Adani, G. Coppini, C. Fratianni, P. Oddo, S. Simoncelli, M. Tonani, V. Lyubartsev, S. Dobricic, et A. Bonaduce. Mediterranean sea large-scale lowfrequency ocean variability and water mass formation rates from 1987 to 2007: A retrospective analysis. Prog. Oceanogr., 2013.

Waldman, R., S. Somot, M. Herrmann, A. Bosse, G. Caniaux, C. Estournel, L. Houpert, L. Prieur, F. Sevault, and P. Testor (2017), Modeling the intense 2012–2013 dense water formation event in the northwestern Mediterranean Sea: Evaluation with an ensemble simulation approach, J. Geophys. Res. Oceans, 122, doi:10.1002/2016JC012437.
* * *

---

## Author Comment (AC1) · 16 May 2017

**Response to the Reviewer #1 comments on the manuscript "Dense water formation in the coastal northeastern Adriatic Sea: the NAdEx 2015 experiment" by I. Vilibic et al.**

*This is a review of the manuscript "Dense water formation in the coastal northeastern Adriatic 2015 experiment" by Vilibic et al. The paper provides a modelling and observational study focusing on dense water formation (DWF) in the Northwestern Adriatic Sea. The main scientific questions formulated by the authors are: 1. is DWF frequent or exceptional in this basin? 2. What are the thermohaline changes associated? 3. What are the exchanges between this basin and the open-sea Adriatic, and therefore its importance for the Adriatic thermohaline circulation? 4. What are the associated mesoscale and frontal processes? For that purpose, the study focuses on the well-documented 2014-2015 case study based on the NADEX experiment. The relevance of the observing design and of the numerical modelling tool to address those scientific questions is appreciated. The problematic is also well-introduced with a comprehensive bibliography, and the general quality of writing is good.*

- We appreciate the words by the reviewer.

*Out of the 4 main scientific questions, the authors actually address questions 1 to 3, the mesoscale and frontal processes being omitted. However, major concerns arise from the irrelevance of many diagnostics with respect to the authors' commentaries and conclusions. As a consequence, several key conclusions of the study are not demonstrated at all or fragile. I therefore recommend a major review addressing the following points:*

- We plan to substantially revise the manuscript following reviewer's suggestions, taking into account all provided comments. A small number of comments on which we disagree are largely result of bad explanations and insufficient details provided in the text. Robust arguments for these statements will be provided in the revised manuscript. Also, we will add more material – like glider profiles – which are addressing mesoscale and frontal processes in the area.

*- the authors give limited support that winter 2014-2015 was milder than average, Fig. 2 not being adapted to this purpose;*

- There are two official reports of the Meteorological and Hydrological Service of the Republic of Croatia which are quoted as references (MHS, 2015, 2016), which classify the winter of 2015 (DJF) as warmer-than-average compared to the base period of 1961-1990. The classification is also available at http://klima.hr/ocjene_arhiva.php. Yet, both reviewers bring out as a weakness in our analyses, so we concentrated on the model period (2008-2015) and estimated average wintertime heat fluxes in the NAdEx area from the model for all winters between 2008 and 2015. For these estimates we used average January and February fluxes (as the DWF occurs in the northern Adriatic exclusively in these months) over the nested ROMS domain (marked by dots in Fig. 1). These estimates put the winter of 2015 (January-February) to be ranked as a normal (see Table 1 below) in terms of average heat losses in the NAdEx area. Relevant changes in the text and additional explanations will be inserted in the manuscript.

Table 1. Cumulative January-February net heat losses (in $MJ/m^2$) estimated from the Aladin/ROMS model for winters between 2008 and 2015.

| 2008 | 2009 | 2010 | 2011 | 2012 | 2013 | 2014 | 2015 |
|------|------|------|------|------|------|------|------|
| -0.53 | -0.76 | -0.78 | -0.75 | -1.20 | -0.80 | -0.49 | -0.80 |

*- they do not provide a clear demonstration that DWF occurred within the NADEX area, with respect to the hypothesis that dense waters were imported to this area from the open-sea Adriatic. The lack of any mixed layer depth estimate is a major drawback in the characterization of DWF.*

- Thanks for this comment, raised also by Reviewer #2, which is a result of insufficient presentation of our results. Namely, the NAdEx region is a shallow region (up to 100 m) and is completely mixed over the most of the domain during wintertime, particularly if there are strong bora events as occurring in winter of 2015 (see details in Section 3.1). Also, previous modelling studies indicate that the DWF is occurring in the area and that dense waters are outflowing towards the open sea (Mihanović et al., 2013; Janeković et al., 2014). Next, major DW pathways from the open northern Adriatic shelf are placed along the

western shore (due to Coriolis force, there is a lot of literature on that), being just hypothesized to occur along the eastern shore (Vilibić, 2003). In fact, our Arvor-C findings are the first proof of the DW coming from open Adriatic to the eastern shore, but this area (Kvarner Bay) is characterized by a gentle slope in its outer part, while the rest of connecting passages has a barrier which restrict eventual near-bottom DW inflow to the coastal region.

As coming from our model results, the most of the area was completely mixed in 2015 and therefore the stratification index presented in Fig. 13 is approaching zero in most of the domain, while the stratification is somehow being increased between bora events and DWF events due to relaxing horizontal advection that may establish a weak stratification at some of the NAdEx area. Vertical homogeneity over the whole water column can be also seen in Hovmoller plots of model results coming at two locations (old Fig. 11). So that, we anticipated an estimation of MLD as not straightforward to reach any conclusion about the DWF. However, we estimated and presented in old Figs. 16 and 17 temporal and spatial properties of the DWF fluxes and transports at connecting passages, which are mostly directed outward. In addition to that, we estimated average DW fluxes at the connecting passages and summarized them in the last row of Table 1 – all of them are directed outward, i.e. from the coastal region towards the open sea.

We believe that these estimates are a proof that the DWF occurred in the NAdEx area, and not advected from the outside (except at the very bottom in the Kvarner Bay). Yet, our arguments are obviously not properly presented in the manuscript, so that plan to add a separate section 6.3 in the manuscript, which is summarizing all the results and arguments about the DWF. Also, we will add new material in Section 1 and add a paragraph in Section 7, which discusses this question and summarizes our results with respect to previous DWF studies in the NAdEx area.

*- the velocity section doesn't illustrate, contrary to the authors' claim, that lateral exchanges at the basin boundaries are mostly baroclinic with outcoming waters at the surface and incoming waters at depth;*

- Ok, we will rephrase the statement, concentrating to the description of the lateral exchanges and not going into explanation of processes.

*- the model evaluation omits major elements of observed variability: is the bottom temperature decrease and density increase during winter trend accurately represented? Does the model have a baroclinic circulation structure at the transects with incoming waters at depth and outcoming waters at the surface, at least at A2 and A7 locations?*

- We added more on the model performance at the lateral boundaries, particularly commenting on A2 observations. The connecting channel Sedmovraće, off which the station A2 has been positioned toward the open sea, is very narrow, about 600 m in its deep section, and the model with horizontal resolution of 500 m is obviously not capable to reproduce the dynamics at these spatial scales. Nevertheless, this channel is connecting outer waters with inner in the channel area between A1 and A2, while the latter are connected with the rest of the NAdEx area through shallow passages only (< 40 m), which physically prohibit the inflow of near-bottom dense waters from the open sea. Therefore, it is not physically possible that dense waters come from the open sea to Kvarnerić and Velebit Channels through Sedmovraće (where A2 station has been moored).

*- the authors don't convincingly show that in the model, DWF occurs locally in the NADEX area and results from Bora wind events. The lack of any mixed layer depth estimate is again a major drawback in the characterization of DWF.*

- As said, the computation of the mixed layer depth is straightforward in deep environments (like Southern Adriatic or Gulf of Lions), but not for a shallow area which is mixed to the bottom, what is the case for the northern Adriatic between November and March (including NAdEx area) (Franco et al., 1992; Artegiani et al., 1997; Viličić et al., 2009). We will put more estimates and arguments about DWF in separate section 6.3 of the manuscript.

  Artegiani, A., Bregant, D., Paschini, E., Pinardi, N., Raicich, F., Russo, A., 1997. The Adriatic Sea general circulation Part I Air–sea interactions and water mass structure. Journal of Physical Oceanography, 14, 1492-1514.

Franco, P. and Michelato, A., 1992. Northern Adriatic Sea: Oceanography of the basin proper and of the western coastal zone, Science of the Total Environment, Suppl., 35-62.

Viličić, D., Kuzmić, M., Bosak, S., Šilović, T., Hrustić, E., Burić, Z., 2009. Distribution of phytoplankton along the thermohaline gradient in the north-eastern Adriatic channel; winter aspect. Oceanologia, 51, 100-118.

*- a major concern arises in the methodology used to calculate residence times within the basin: by separating along-basin and cross-basin fluxes, the authors largely overestimate the residence time which should include altogether all boundary fluxes. Also, the probability density function is not defined and probably not adapted (too noisy) to get a robust residence time estimate: instead, a temporal mean residence time estimate would probably give a more convincing result.*

- Ok, we simplified the computation of the residence time and present result in the form of box-whiskers diagram (Fig. 1). A simple estimate of the residence time within the studied domain was obtain using equation:

$$RT = \frac{\iiint_{x,y,z} \rho(x,y,z)\, dx\, dy\, dz}{\oiint_{C,z} \rho(x,y,z)\, \vec{u}(x,y,z).\vec{n}\, dC\, dz}$$

where $\rho(x,y,z)$ is the density of the water at each point $(x,y)$ and for each depth $z$ of the domain, while $\vec{u}(x,y,z).\vec{n}$ is the normal velocity along the contour $C$ of the domain. Such an approach assumes that (i) only the velocities at the border of the domain are used to calculate the residence time, (ii) only the outflow of water at the border are taking into account; the goal here is to only look at how long it would take for the water masses to leave the domain assuming that there is no income of water within the studied domain, and (iii) the residence time is calculating with a time step of 3 h over the period of the studied event, assuming a steady state of the dynamical conditions at each time step. The box-whiskers diagram (Fig. 1) is thus showing the statistical properties of the estimated residence time for each 3 h model results.

[Figure]

Fig. 1. Simple estimate of the residence time of both total water (TRT) mass and dense water (DWRT) within the studied domain.

*Here is a detailed description of major concerns.*
*M1 There are too many plots: Fig. 8b is unnecessary, especially because glider data is not shown, and an average bias for both temperature and salinity would be sufficient; model-observation comparisons could easily be merged with observation-only plots; Fig. 9 and 10 are probably unnecessary (see comments below).*

- We don't agree with this reviewer comment. Particularly, as Reviewer #2 asks for inclusion of a glider data figure, which we did as the figure is showing the existence of the thermohaline front. By doing that we provided new results to support the existence of the thermohaline front stretching from the Kvarner Bay towards the open sea, what is recognized as a drawback by the reviewer.

Regarding Figs. 9 and 10, they provide an information about the reliability of the model versus all measured data, which are essential for assessing the quality of the model discussed in Section 7. For example, Fig. 10 clearly show an underestimation of currents by the model, not provided by any other figure, which is quite important result highlighted in the abstract and being the base for discussion on ways to improve modelling in such a complex area. So that, we are in favour for keeping them in the manuscript. See also the response to the M7 comment.

*M2 Fig. 2 doesn't prove that winter 2014-2015 was milder than average. For that, an anomaly with respect to an interannual mean should be displayed. Also, a computation of the thermal and saline buoyancy flux at the surface would help to relate surface conditions to DWF. The only support for a mild winter is given with percentiles of surface temperature. However, several points are not clear: the percentiles are computed with respect to what period? A 30-year climatology? Is it the near-surface atmospheric temperature over the sea or the sea surface temperature? Also, the presence of mild periods doesn't prove that the whole winter was on average milder: for that, a time average needs to be calculated.*

- Yes, we agree - that was claimed following official publications of the Meteorological and Hydrological Service of the Republic of Croatia (MHS, 2015, 2016). We do not claim such a statement from Fig. 2. Anyway, we computed some statistics and comparison between winters on heat fluxes from the model – the details are provided at the beginning of this document. We will insert these computations and analyses into the revised manuscript.

*M3 Fig. 3 is valuable but it doesn't prove that DWF occurred within the NADEX area (which corresponds to the nested ROMS area) as the dense waters could also have been imported from the open-sea. The same is true for Fig. 4 as the dense and salty anomaly could have been imported from the lateral boundaries. Finally, Fig. 5 doesn't prove either that DWF occurred in the NADEX area. Unfortunately, no mixed layer depth was computed in this study, but using a classical density anomaly criterion of 0.01 to 0.05 kg/m3 with respect to surface density (e.g. De Boyer Montegut et al 2004, Houpert et al 2015) would show clearly that the mixed layer is shallow throughout the transect. A comparison of the Stratification Index at the bottom with typical surface buoyancy fluxes modelled by ALADIN could be an approach to suggest (but not demonstrate) that DWF occurred during the winter.*

- To repeat, the comment and all of the quoted references are dealing with the dense water formation occurring through deep ocean convection, for which the computation of mixed layer depth is essential. Here, the NAdEx area is shallow and mixed to the bottom in the winter of 2015. There is a substantial literature on that for the shallow northern Adriatic shelf (already presented in introduction, but we add a few more references), none of them using MLD in DWF analyses as not being relevant. However, we agree that the proof for the DWF in the NAdEx area and the associated DW dynamics in the connecting channels is not explained well, so we further developed the analysis, which will be presented in separate section 6.3 of the revised manuscript. See also previous comments on the DWF issues.

*M4 Fig. 6 doesn't illustrate a general incoming of dense water at depth and outcoming of lighter waters at the surface. Contrary to the authors' statement, only stations A2 and A7 show a predominantly baroclinic vertical structure of velocity. The stations A3, A4 and A7 to A9, which cover the main connecting section between the NADEX area and the open-sea also show a strong barotropic flow. The interpretation of this figure should be more cautiously done.*

- We agree and we will rewrite these statements in the revised manuscript.

*M5 The model shows a large temperature and salinity bias in the open-sea Adriatic which therefore comes from the low-resolution ROMS model. It should be at least documented and interpreted, or at best reduced in order to produce realistic lateral exchanges between the NADEX and open-sea areas.*

- Ok, we added more discussion on that, based on previous works by Janeković et al. (2014) and Vilibić et al. (2016). Both studies document the negative salinity bias observed in the whole Adriatic, being presumably a result of "lateral AREG model boundary conditions that use river discharges based on the Raicich climatology" (Vilibić et al., 2016). Some other studies apply a bias correction for their simulations of DWF in the Adriatic (e.g. Benetazzo et al., 2014), but we are not in favour of such an approach; we think that it is better to find a feasible explanation for such a model behaviour, which may then be corrected and lead to an improvement of the model.

*M6 If 20 sigma levels are not sufficient to simulate dense water transformations in the NADEX area (the bottom dense water layer is not modelled), how many would be necessary? What was the rationale in the choice of only 20 sigma levels when regional and coastal Mediterranean Sea models exhibit typically between 50 and 100 vertical levels (e.g. Mediterranean Sea reanalyses in Pinardi et al 2013 and Hamon et al 2016)?*

- The Adriatic Sea, and in particular northern Adriatic Sea, is a shallow place and it does not need 100 sigma levels as for the Mediterranean. Some of previous studies reproduced reliably the DWF using 20-30 sigma layers (e.g. Janeković et al., 2014; Benetazzo et al., 2014). Also, our model run is a multi-annual run (2008-2015), not just focused on a single dynamics events but more to get a broader picture on the DWF and other processes in the northern Adriatic. Indeed, our study documents very fine details in DWF-related processes for the very first time, as some of NAdEx measurements are the very first of such kind in the Adriatic (e.g. Arvor-C and capturing of fine structures near the bottom of the Kvarner Bay). So that, our results are opening a direction to improve the DWF modelling, also by increasing number of sigma layers to better reproduce near-bottom dynamics. Yet, this is beyond the scope of this paper. Discussion on that will be added in Section 7, when commenting model drawbacks and ways how to resolve them in the future.

*M7 Fig. 10 gives a too raw evaluation to be insightful because there is no spatiotemporal information, and it should be removed as it is currently. Some elements of observed bottom hydrology and velocity profiles were noted in the observations section and not even commented in the model evaluation section. In particular: is the bottom temperature decrease and density increase trend accurately represented? Is the salinity front also visible at the same area? Does the model have a baroclinic circulation structure with incoming waters at depth and outcoming waters at the surface, at least at A2 and A7 locations? By the way, it would have been insightful to also evaluate the current directions.*

- We don't agree that Fig. 10 (and Fig. 9) in the manuscript are not providing substantial information about the model performance – the presented analysis is a statistical evaluation of model-to-observations performance, normally used in such studies and using standard presentations (Q-Q plots and box-whiskers diagrams). There is a large number of conclusions in the text derived from these figures, e.g. about an underestimation of currents in connecting passages, a good reproduction of temperature over a range of percentiles (what is quite important for assessment of heat losses and the DWF), and other. Yes, we agree that temporal evolution of the model performance cannot be achieved with such an approach, but such a detailed assessment is beyond the scope of this study. Also, some of raised questions (e.g. the last one) – on which we agree that it would be quite useful to discuss in the paper - is in fact extending the analysis compared to the presented one, so we plan to put extra material and the text in the revised manuscript.

  In particular, we plan to add modelled residual currents and ellipses of standard deviation at all A stations (Fig. 2). It can be seen that (i) the model reproduces satisfactorily the observations (bearing in mind that the bathymetry and the atmospheric forcing is quite complex in the area) except at A2 station, but this station is placed in quite narrow connecting passage (about 600 m) not properly resolved with the model (resolution of 500 m) – more comments on that are already placed above, (ii) the model reproduce fairly well vertical changes in currents (again except at A2), aside the fact that the direction may differ from observations due to differences between real and model shorelines and bathymetry (like at A7 and A9). Anyway, we think that the model results are usable for investigations of DWF and transports in the area, while their shortcomings are discussed in the discussion section.

[Figure]

Figure 2. Mean residual currents and ellipses of standard deviations measured (in red) and modelled (in blue) at A1 to A9 stations between 1 February and 1 April 2015.

*M8 Fig. 11 is also too raw, it could be improved to relate hydrological transformations to DWF and Bora wind events. Once again, no mixed layer depth was computed and the large range in the colorbar makes it impossible to identify the period of DWF: it would require to read vertical density variations of typically 0.01kg/m3. The Bora wind was mentioned to explain the stepwise cooling, but a comparison between the vertically-integrated heat loss and the surface heat flux would give a much more convincing relation between cooling and Bora events.*

- We replot the figure by using different colobar, which provides much more details to the cooling events and the associated DWF (Fig. 3). Again, we did not compute MLD as not relevant for the DWF in shallow (shelf) areas. It can be seen clearly that no dense waters are advected to the region after the bora events. Also, we added simple computations of bora-driven cooling to make the relationship more robust, using a simple box model (e.g. Gill, 1982):

$$\Delta T = \frac{1}{Hc\rho_0} \int_{t_1}^{t_2} Q \, dt$$

where $\Delta T$ is the temperature changes induced by surface heat loss $Q$ in the time interval between $t_1$ and $t_2$, while $H$ is the ocean depth. This model assumes no lateral exchange of energy between the box and the adjacent sea, what is a fair approximation for short and transient events like the bora. At station G1 this simplified formula gives $\Delta T$ = -0.96°C, -0.48°C and -0.03°C for three bora events (the third bora was very weak at G1), respectively, while the respective cooling rate as provided by the model is -1.04 °C, -0.51 °C and -0.07 °C. Given the assumptions of this simple box model, one can say that the cooling in the area is dominantly driven by the bora wind.

More on that will be added to the manuscript.

[Figure]

Figure 3. New Fig. 11 from the discussion paper, with colour bar adopted to show more details.

*M9 Without mentioning the depth at which the SI is computed, it is impossible to interpret Fig. 13. Assuming that the SI was computed at the bottom of the water column for each location, Fig. 13 could have provided an SI-based mixed layer depth map (threshold at 0.01 m2/s2), already used in several Mediterranean studies (e.g. Waldman et al. 2017). The link with the Bora event is not clear because: 1. most of the domain already has an SI<0.01m2/s2, the signature of DWF, before the Bora occurs; 2. only the comparison between the ocean SI loss and the integral surface buoyancy flux would allow to attribute buoyancy losses to the Bora. As a consequence, the authors' conclusion of 6.1 has not been demonstrated.*

- The SI is computed for the whole water column – the info is added in the text - to present the level of stratification and to assess if stratification may prevent the DWF. Again, MLD is not the parameter considered relevant if DWF studies over a shelf, while the suggested literature is dealing with deep-convection processes occurring over a few thousand meter depths.

*M10 In Fig. 18 the methodology to compute the probability distribution of residence time is not explicited. I assume the residence time is computed for each model day and the probability distribution is therefore temporal and daily based on the winter period (which period?) Results are very noisy which makes them difficult to interpret: I strongly recommend to compute a temporal mean residence time and not a probability distribution function in order to increase the robustness of results. Also, even though it is interesting to differentiate between along-basin and cross-basin fluxes, the residence time of a water parcel is impacted by all boundary fluxes, which cannot be considered separately: by doing so, the authors artificially increase the residence time. I therefore recommend the authors to compute a residence time including all boundary fluxes, and to compare along-channel and cross-channel fluxes without deducing residence times from them.*

- Ok, we changed the methodology of residence time estimates as suggested. Also, we switch the presentation from probability distribution to simple box-whiskers statistics. The details are provided in introductory part of this document.

*Here is a description of minor concerns:*
*Abstract*
*m1 Do not use the term one-way coupling. There is no feedback of the ocean models toward the atmospheric model, therefore there is no ocean-atmosphere coupling.*

- Ok, corrected.

*2.3.*
*m2 What is the consistency between the sea surface temperature at the boundary of the forcing atmospheric model and that of the ocean model?*

- NWP ALADIN/HR model used in our study uses SST fields coming from the IFS (Integrated Forecast System) operational forecast run in ECMWF (European Centre for Medium Range Forecast). These SSTs have a positive bias towards in situ measurements during the winter and during bora episodes in the Kvarner Bay. When compared to the SST satellite observations at the open Adriatic, the bias is much

lower. It has been found that these SST differences have an impact on the value and location of the precipitation maximum in the domain, but affect wind speed only slightly, as strong wind episodes (such as bora) driven by a strong synoptic forcing are primarily controlled by surrounding topography. There is paper on the subject has been submitted only recently but there are few reports available (ALADIN/HIRLAM Newsletter No8. http://www.umr-cnrm.fr/aladin/meshtml/NL8-final.pdf and several posters and Workshop presentations)

*m3 What is the time resolution of the atmospheric forcing and how does it impact the representation of extreme heat fluxes associated with Bora events?*

- The ALADIN/HR output is every 3 hours, as being a standard of the operational product of the Meteorological and Hydrological Service. Although bora wind may have a substantial variability on periods from several minutes to a few hours, previous modelling studies that used 3-h ALADIN/HR forcing provided reliable results (e.g. Janeković et al., 2014). Unfortunately, we cannot give a response to the second part of the question, for which sensitivity studies should be carried out.

*m4 Be more explicit about how bathymetry smoothing ensures the run stability.*

- The bathymetry smoothing has been performed using a linear programming technique (Dutour Sikirić et al., 2009) that suppresses horizontal pressure gradient errors. Such approach is required for complex bathymetry with steep slopes such as the Adriatic Sea, to get an unbiased result for longer multiyear integration. The explanations will be provided in the revised manuscript.

  Dutour Sikirić, M., Janeković, I., Kuzmić, M., 2009. A new approach to bathymetry smoothing in sigma-coordinate ocean models. Ocean Modelling, 29, 128-136.

*m5 The mention of an operational integration for the large-scale ROMS model usually implies that data assimilation was done within the model domain. Is it the case?*

- Operational integration is mentioned just for ALADIH/HR which is numerical weather prediction model, not for ROMS, which has no assimilation imposed. The reference about the ALADIN/HR assimilation will be added in the manuscript (Stanešić, 2011).

  Stanešić, A., 2011. Assimilation system at DHMZ: Development and first verification results. Hrvatski meteorološki časopis, 44/45, 3-17.

*m6 Was the nesting 2-way (coupling between both ocean models) or 1-way (forcing of the nested model by the large-scale model)? If it was 1-way, do you believe that the absence of any feedback toward the large-scale model alters the estimated lateral transports at the boundary of the NADEX domain?*

- It is the one-way nesting. For sure, the two-way nesting at lateral boundaries will change the transports there, yet we consider these effects not substantial for description of the DWF. What is more important, in our humble opinion, are feedback processes at the ocean surface (two-way coupling between ALADIN and ROMS), which is known to alter modelled wind and ocean currents and heat fluxes in the open northern Adriatic up to 10-20 % during bora events (Pullen et al., 2007, Ličer et al., 2016). Then, the changes in the wind field will for sure change water exchange in connecting passages. Some of the discussion regarding these issues is already in the manuscript, but we will extend the discussion in the revised manuscript and add new references.

  Pullen, J., Doyle, J., Haack, T., Dorman, C., Signell, R., Lee, C.M., 2007. Bora event variability and the role of air-sea feedback. Journal of Geophysical Research, 112, 33407, doi:10.1029/2006JC003726.

  Ličer, M., Smerkol, P., Fettich, A., Ravdas, M., Papapostolou, A., Mantziafou, A., Strajnar, B., Cedilnik, J., Jeromel, M., Jerman, J., Petan, S., Malačič, V., Sofianos, S., 2016. Modeling the ocean and atmosphere during an extreme bora event in northern Adriatic using one-way and two-way atmosphere–ocean coupling. Ocean Science, 12, 71-86.

*3.*
*m7 Missing mention top/middle/bottom in Fig. 2.*

- Ok, added.

*m8 No critical assessment of the gust wind event was made. Which is the Bulk turbulent flux formulation used in the ALADIN configuration? Most state-of-the-art Bulk formulations have not been calibrated for winds >20m/s because of the scarce observations (e.g. Fairall et al 2003), which makes them unrealistic and divergent between each other at such wind regimes. Therefore interpretations of ALADIN outputs should be more cautious and critical.*

- Yes, we are aware that bulk estimates may not be appropriate for some winds and ocean regions, also for the Velebit Channel which exhibit strong and gusty bora, with a lot of sea spray during extreme events. Yet, ALADIN products (winds, temperature, …) has been verified also in the NAdEx area during a number of severe bora episodes (e.g. Tudor et al., 2012, 2013). We will add some text and the reference on that issue in the discussion section.

  The wind gusts in ALADIN/HR are computed according to Boržkova et al. 2006. This particular modification was introduced as ALADIN was used in support of the MFSTEP oceanographic campaign. This is why we find it most suitable for this purpose. The problem is more complex. There are few measurements to calibrate (as pointed by the reviewer) and currently no measurements above the sea surface in this area. However, this area is composed of many small basins where waves have to build up as the wind blows from the coastline. Any relation derived for the open ocean would probably fail here. Coupling to a wave model seems more promising solution.

  Brožková, R., Derková, M., Belluš, M., Farda, A., 2006. Atmospheric forcing by ALADIN/MFSTEP and MFSTEP oriented tunings, Ocean Science, 2, 113-121.

  Tudor, M., Ivatek-Šahdan, S., Stanešić, A., 2012. February 2012 winter conditions in Croaria. 6[th] HymMeX Workshop, Primošten, poster available at https://www.researchgate.net/publication/283897781_February_2012_winter_conditions_in_Croatia.

  Tudor, M., Ivatek-Šahdan, S., Stanešić, A., Horvath, K., Bajić, A., Zhang, Y., and Ray, P., 2013. Forecasting weather in Croatia using ALADIN numerical weather prediction model. Climate Change and Regional/Local Responses. InTech, Rijeka, 59-88.

*4.*
*m9 Regarding the presence of a salinity front between A7 and A9, the large high-frequency variability gives some doubts about the significance level of such a signal. Could you provide it please?*

- We estimated significance between temperature and salinity series at A7, A8 and A9 stations by applying t-test for mean and std values (Table 2). Both temperature and salinity series are significantly differing between all stations with p<0.001. In addition, to put the conclusion about thermohaline front more robust, we will add a figure with glider data to the manuscript, where the front can be clearly seen (Fig. 4). Of course, the front may not be a permanent feature during the whole wintertime period. These results and discussion are going to be added to the revised manuscript.

Table 2. Statistic of temperature and salinity data at stations A7 to A9 between 1 February and 31 March 2015.

| Station | Mean temperature | Std temperature | Mean salinity | Std salinity |
|---|---|---|---|---|
| A7 (8497 data) | 12.128 | 0.570 | 38.265 | 0.133 |
| A8 (16993 data) | 11.619 | 0.706 | 38.027 | 0.204 |
| A9 (8497 data) | 11.302 | 0.737 | 37.979 | 0.213 |

[Figure]

Figure 4. Temperature and salinity profiles measured by Slocum glider between 24 and 27 February 2015 in front of the Kvarner Bay. Glider trajectory is provided in Fig. 1 of the discussion paper.

*5.*
*m10 Quantify the model bias decrease between the winter and spring cruises.*

- The bias is depth-dependant, so we analysed vertical distribution of the bias (Fig. 5). It can be clearly seen that winter bias – both in temperature and salinity - is much larger during over the most of the water column, except sporadically in deeper layers. Some text on that will be added to the manuscript.

[Figure]

Figure 5. Vertical profiles of the bias and root-mean-square-error during spring (May 2015) and winter (December 2014) cruises.

*m11 Comment the persistent low salinity bias around station 18.*

- This minimum could be due to submarine springs which are discharging freshwater from neighbouring Vrana Lake to the sea and which are not introduced to the model. The area around station 18 is quite shallow, so such an effect may be seen on salinity data. A comment on that will be added to the text.

*m12 When assessing the model bias, it is clearer to display the difference model minus observations, so that Fig. 7 to 10 can be interpreted directly as model bias.*

- Ok, we agree, that would be done in revised manuscript.

*m13 Fig. 9 is unnecessary as it reiterates the evaluation done in Fig. 7 and 8, without a clear added-value in the interpretation of using Q-Q plots or box whiskers diagnostics. Also, the mention of an ideal distribution in the Q-Q plots is unclear, as all observations should fall within this slope 1 line (e.g. Herrmann and Somot 2008).*

- We don't agree with the reviewer, as Fig. 9 does provide additional information about average temperature and salinity bias of the CTD data, not attainable from Figs. 7 and 8. Also, the figure present comparative statistics for datasets coming from different measuring platforms. So, we decided to keep the figure. We will rewrite the caption of Fig. 9 not mentioning ideal distribution any more.

*m14 It would be more relevant to compare temperature and salinity biases to their variability for instance, rather than to their absolute value, in order to compare biases to the typical hydrological variability in the area.*

- Yes, we agree that bias is not describing the variability of the temperature and salinity, so that we computed rmse (root-mean-square-error) values as well. Their overall values are provided in Table 3. Yet, we decided not to present all of these computations in the manuscript, as the model verification part of the manuscript will be too large compared to the rest of the results, so we are planning to add a few results describing the rmse variability in the manuscript.

  Table 3. Model-to-observations bias and rmse values at A stations (the lowest layer has been taken for current speed estimates) for the whole measuring period.

| station | current speed (m s$^{-1}$) | | temperature (°C) | | salinity | |
|---|---|---|---|---|---|---|
| | bias | rmse | bias | rmse | bias | rmse |
| A1 | -0.05 | 0.06 | -0.54 | 0.35 | | |
| A2 | -0.04 | 0.05 | -0.30 | 0.38 | | |
| A3 | | | 0.29 | 0.85 | -0.05 | 0.14 |
| A4 | -0.03 | 0.05 | 0.15 | 0.39 | -0.02 | 0.09 |
| A5 | -0.02 | 0.04 | -0.09 | 1.29 | | |
| A6 | -0.02 | 0.05 | 0.32 | 0.67 | | |
| A7 | -0.04 | 0.05 | 0.36 | 0.44 | 0.02 | 0.16 |
| A8 | -0.04 | 0.05 | 0.46 | 0.65 | 0.16 | 0.20 |
| A9 | -0.04 | 0.05 | 0.56 | 0.88 | 0.07 | 0.19 |

*6.1*
*m15 Fig. 12 shows repetitive time series: instead, an area-averaged surface buoyancy flux time series (in m2/s3/day, and not m2/s3) would give qualitatively the same result, but more integrated. The difference between locations should be documented quantitatively (how does each location compare to the area-average?) in the text, without displaying the 7 plots.*

- No, we are not presenting temporal changes of the buoyancy flux, but the flux itself, so the unit should be m$^2$/s$^3$. Thus we will change the caption to "daily values" instead of "daily changes". Also, we followed the reviewer's suggestion and computed the time series of averaged buoyancy fluxes over the whole NAdEx domain (Fig. 6), while spatial differences will be commented in the text only. The new figure will be placed in the revised manuscript.

[Figure]

Fig. 6. Daily values of surface buoyancy fluxes and its components modelled over the whole nested model domain.

6.2.

*m16 The so-called temperature flux should be named heat flux.*

- Ok, changed.

*m17 Specify when mentioning Fig. 14 and in its legend that it is modelled results.*

- Ok, specified.

*m18 Fig. 14 is misleading because of a factor up to _30 between transect widths: a constant scaling would be more appropriate to compare heat and salt fluxes. Same comment for the vertical scaling.*

- Ok, done, the figures are changed – see below (Figs. 7 and 8).

[Figure]

Figure 7. Replotted Fig. 14 from the discussion paper.

[Figure]

Figure 8. Replotted Fig. 16 from the discussion paper.

*m19 How do you interpret that the NADEX area is losing heat through its lateral boundaries despite the stronger Bora heat loss than in the open-sea (Fig. 2 middle panels)?*

- That is due to wind-driven dynamics, that obviously dominates over the thermohaline circulation during strong bora events.

*m20 It is not intuitive that dense water mass fluxes should be directed outward as Fig. 3 suggests that the densest waters with a high salinity signature came in from the open Adriatic and were not formed locally. How do you explain this apparent discrepancy between the model and observations?*

- We don't agree that Fig. 3 show an inflow of any water through the open boundary. It simply shows that the DWF occurred somewhere and that dense waters are present in bottom layers. We will now put more evidences that these waters are generated in the NAdEx area, while the open Adriatic dense waters came only at the very bottom of the Kvarner Bay (where CTD sampling has not been carried out).

*m21 The interpretation of some high-frequency transport variability in Fig. 15 being driven by Bora events is interesting and could be tested by calculating the transports induced by Ekman currents.*

- We agree, however such a computation will direct the paper to new investigations, which in our opinion deserve a throughout analysis and may go into a separate paper. Thus, we will just discuss the idea in the text.

*m22 How do you interpret that the residence time for dense waters is lower than that for the total volume, when it is visible from Fig. 6 that in observations (but the same is probably true for the model), velocities are lowest in the deep layers where dense waters are located? One should therefore expect a longer residence time for those dense waters.*

- Yes, the velocity of dense waters are generally smaller than overall velocities, however, the volume of dense water is much smaller than of overall NAdEx volume, while its outflow is occurring not just at the bottom (if density threshold is used for definition of dense waters). For that reason, DW residence time is smaller than total volume residence time.

*References mentioned in this review:*

*Clément de Boyer Montégut, Gurvan Madec, Albert S Fischer, Alban Lazar, et Daniele Iudicone. Mixed layer depth over the global ocean: An examination of profile data and a profile-based climatology. Journal of Geophysical Research: Oceans, 109(C12), 2004.*

*CW Fairall, Edward F Bradley, JE Hare, AA Grachev, et JB Edson. Bulk parameterization of air-sea fluxes: Updates and verification for the coare algorithm. Journal of Climate, 16(4): 571–591, 2003.*

*M. Hamon, J. Beuvier, S. Somot, J.-M. Lellouche, E. Greiner, G. Jordà, M.-N. Bouin, T. Arsouze, K. Béranger, F. Sevault, C. Dubois, M. Drevillon, et Y. Drillet. Design and validation of medrys, a mediterranean sea reanalysis over the period 1992-2013. Ocean Science, 12(2):577–599, 2016. doi: 10.5194/os-12-577-2016. URL http://www.oceansci.net/12/577/2016/.*

*M. Herrmann et S. Somot. Relevance of ERA40 dynamical downscaling for modelling deep convection in the Mediterranean Sea. Geophys. Res. Lett., 35(L04607), 2008. doi: 10.1029/2007GL032442.*

*L. Houpert, X. Durrieu de Madron, P. Testor, A. Bosse, F. D'Ortenzio, M.N. Bouin, D. Dausse, H. Le Goff, S. Kunesch, M. Labaste, L. Coppola, L. Mortier, et P. Raimbault. Observations of open-ocean deep convection in the northwestern mediterranean sea: Seasonal and interannual variability of mixing and deep water masses for the 2007-2013 period. Journal of Geophysical Research: Oceans, pages n/a–n/a, 2016. ISSN 2169-9291. doi: 10.1002/2016JC011857. URL http://dx.doi.org/10.1002/2016JC011857.*

*N. Pinardi, M. Zavatarelli, M. Adani, G. Coppini, C. Fratianni, P. Oddo, S. Simoncelli, M. Tonani, V. Lyubartsev, S. Dobricic, et A. Bonaduce. Mediterranean sea large-scale low-frequency ocean variability and water mass formation rates from 1987 to 2007: A retrospective analysis. Prog. Oceanogr., 2013.*

*Waldman, R., S. Somot, M. Herrmann, A. Bosse, G. Caniaux, C. Estournel, L. Houpert, L. Prieur, F. Sevault, and P. Testor (2017), Modeling the intense 2012–2013 dense water formation event in the northwestern Mediterranean Sea: Evaluation with an ensemble simulation approach, J. Geophys. Res. Oceans, 122, doi:10.1002/2016JC012437.*

- The most of references are relevant for the deep convection, which is here not the case. We will add a few of them to the revised manuscript as general references on the dense water formation.

---

## Author Comment (AC2) · 16 May 2017

**Response to the Reviewer #2 comments on the manuscript "Dense water formation in the coastal northeastern Adriatic Sea: the NAdEx 2015 experiment" by I. Vilibic et al.**

General comment to the Authors: The manuscript investigates the dynamics of the coastal areas in the northeastern Adriatic Sea during winter 2015, using numerical models and in situ data. The data were collected during an intense fieldwork that was conducted using a multiplatform approach, involving ADCPs, CTDs, glider and a profiling float. In particular, the authors focus on the possibility that dense water forms even in this area of the northern Adriatic, and not only during severe winters, but also during relatively mild winter (winter 2015 in fact was one of those). The objectives of the paper are sufficiently clear but not well discussed. The structure of the paper could be better organized and the figures and captions are all relevant, but not all the data were shown. There are a number of aspects that need to be clarified to the reader, before the paper would be publishable in Ocean Science.

• Thanks for comments, we revised substantially the paper following reviewer's suggestions.

A major comment is that I don't think the authors have uniquely demonstrated that the formation of dense water occurred in the investigated area. Further you have not shown to the reader how mild was winter 2015 compared to other winters. With this in mind, I think the paper deserves publication after a major revision.

• These two issues, which are also raised by Reviewer #1, are now demonstrated. New section on DWF will be added to the revised manuscript, synthesizing all arguments regarding the DWF. The details are provided in Response to comments of the Reviewer #1.

Some more detailed comments are: - Page 1, Line 17: should be "accompanied by"

- Ok, to be corrected.
- Page 1, Line 18: do not define acronyms in the abstract, but only later on (DWF)
  - Ok, to be done.
- Page 1, Line 25: should be "to be about 1-2"
  - To be corrected.
- Page 2, Line 2: should be "mixing on the"
  - Ok, to be corrected.

- Page 2, Line 5: in addition to heat losses, also evaporation should be mentioned as an important contributing factor.

• Ok, to be added to the text.

- Page 2, Line 17: should be "from the eastern coastal areas".

- Ok, to be corrected.
- Page 4, Line 26: should be "The atmospheric".
  - Ok, to be corrected.

- Page 7, Line 16: I think the glider measurements would be important and should be described, and shown.

• Ok, we will add to the revised manuscript a figure showing glider measurements and the text describing the measurements (Fig. 1). We agree that this figure is quite important to show the existence of the thermohaline front (among other things), which is recognized as a drawback in the manuscript.

Figure 1. Temperature and salinity profiles measured by Slocum glider between 24 and 27 February 2015. The path of the glider is shown in Fig. 1 of the discussion paper.

- Page 7, Lines 9-18: My main concern here is how can you exclude that advection is the cause of what you observe here?

• Ok, we rewrote the paragraph, as the assessment of the DWF and dense water dynamics is performed later in the manuscript.

- Page 10, Lines 9-16: there seems to be a contradiction since in the first part you speak about "horizontal salinity gradients" and of the fact that "cooled waters were largely advected to this area", while afterwards you speak about "DWF in the area", which for me has not been proven in this paper.

• DWF has been occurring during transient bora episodes, lasting for a few days. As DWF is spatially inhomogeneous due to strong variation in heat losses and in freshwater load, the ocean started to relax between bora episodes through horizontal advection. We will clarify this issue in the manuscript, particularly in new section 6.3 on dense water formation.

- Page 10, Line 24-25: this sentence "acted mostly in opposite to the thermally driven buoyancy changes" is not clear at all.

• To be clarified.

- Page 12, Lines 9-22: I don't understand why you decompose the residence time in along and across and not just use the standard residence time. Besides the mathematical formulation you should give the reader a physical explanation on why you do that and why it should be important.

• Following also suggestion by Reviewer #1, we simplified the estimate of residence time and provided the result in the form of box-whiskers diagram. See Response to Reviewer #1 comments.

- Page 12, Lines 32-33: you say the in the outer basin the residence times "are much lower" (than what) and after this sentence you say that "for the inner basin ... residence times are much shorter": : :.so they are short in and out, but with respect to what: : :? Really unclear!

• To be rewritten, also following changes in computations of the residence time.

- Page 13, Line19: what is the meaning of "This is a baseline Nadex 2015 paper"???

- To be deleted as not providing any information.
- Page 13, Line 22: I don't think you have demonstrated item (i)!
  - We extended the analysis and provided arguments which demonstrate the occurrence of the DWF in coastal northeastern Adriatic in winter of 2015. The details are provided in the Response to comments of the Reviewer #1.
- Page 13, Line 32: what is the meaning of "has still excited thermohaline circulation"?? It sounds odd.
  - To be clarified.

---

## Author Response (AR2)

**Response to the reviewers' comments on the manuscript "Dense water formation in the coastal northeastern Adriatic Sea: the NAdEx 2015 experiment" by Vilibić and co-authors, submitted to Ocean Science (osd-2017-6)**

We thank to both reviewers for their comments which we believe that improved significantly quality of the manuscript.

Reviewer #1 comments:

*I still think that the paper needs major work, I find it very confusing, surely because there are a lot of data to be discussed and shown, but the reader needs to be guided through all these information in a more effective way.*

- We performed the major revision of the paper following comments raised by the reviewer, whose comments are along the line with comments of Reviewer #1. So, we hope that the new version of manuscript is more coherent, structured and fluid to read.

*English must be revised possibly by a native English-speaking person.*

- The language is corrected by language-correcting service, American Journal Experts.

*The additional section 6.3 that was promised in the answers to the reviewers is not in the new version of the manuscript. I am still not convinced that the DWF occurrence has been demonstrated, and even less that the involved volumes may be relevant for the northern Adriatic system. I think that the model results do not perform very well, especially considering that the authors explicitly show that overall the model did not reproduce properly the observed 2-layer structure as well as under- and overestimation of thermohaline properties. The authors themselves say that the model did not have sufficient resolution to to reproduce the bottom density current (which should be important here, given the aim of the paper)...*

- Although we promised new Section 6.3 to lay down the arguments about the DWF in the NAdEx area, we decided not to do so, as Section 6 is solely dedicated to results. Section 6.3 was supposed to contain predominantly the discussion of the results. Therefore, we decided to reformulate and strengthen the argument pro DWF in Section 7. However, these arguments are still not fully recognized as reliable, also raised by Reviewer #2, so that we rewrote the whole discussion part on the DWF, following the suggestions given by Reviewer #2. Also, we agree that the model performance is not a perfect one, particularly for currents at constrictions influenced by rapidly changing bathymetry or narrow channels resolved only by one or two model grid cells, but we provided extended discussion on that. This discussion will hopefully be of great help for others applying ocean models in regions with complex bathymetry and forcing, including future modelling exercises that for sure will be applied to the NAdEx area, as containing really nice and diverse dataset useful for validation of models. We hope that the text and statements placed in the text are reliable now and following the provided results.

*To conclude, major revision is required to 1) improve the structure of the paper (paragraphs presenting the dataset and guiding the reader through the information, addressing one by one all questions that you want to answer here), 2) convince the reader how important DWF in this area is respect to DWF outside the area, 3) if possible, improve the model results (maybe not feasible).*

- We did the major revision following comments 1) and 2), while for 3) it would be necessary to set up completely new modelling setup, both for atmosphere and ocean. So, we decided to keep the present model, while encouraging further modelling studies in this complex area which might be evaluated on the presented comprehensive observations.

Reviewer #2 comments:

*This is the second review of the manuscript «Dense water formation in the coastal northeastern Adriatic Sea: the NAdEx 2015 experiment» by Vilibic et al. The authors have provided a comprehensive response to the previous comments which is highly appreciated. They have convincingly responded to a majority of the major remarks and to most of the minor ones (although new minor concerns arise), which is a significant step forward toward the publication of their work. However, several major issues previously arosen still remain, which either fragilize some of the authors' conclusions or complexify the reading. I therefore recommend a major revision of the manuscript.*

*The following major concerns still remain:*

*M1 Although the authors disagreed with me on the too large number of Figures, I believe this remark is even truer now that the manuscript reaches 20 Figures. Some of them provide repetitive informations and others are too raw to support the authors' claims, fragilizing them.*

- We reduced the number of figures following suggestions of the reviewer (now we have 17 figures), but also following general comment of the other reviewer, to improve the flow and readability of the manuscript.

*The authors should seek to be more synthetic and to add a few (simple) diagnostics to some Figures to be more convincing. In particular:*

*- I still believe Figures 11 and 12 repeat already-existing informations and the added value of showing statistical distributions is very low as it is presented. Also, as mentioned in former remark m14 (which has not been addressed), I think displaying relative differences (Fig.11c,d and Fig.12d,e,f) is irrelevant, especially for salinity, it gives the misleading impression that the bias is low and it makes it hard to read the actual salinity bias. As mentioned in the previous m14, absolute differences would be easier to read and interpret. For the q-q plots (Fig.11a,b and Fig.12a,b,c), the dashed-dotted line is still not labelled, please explain how it is obtained.*

- We removed Figures 11c,d and Figures 12d,e,f, while the rest is merged into the single figure. Also, we added explanation for the dash-dotted line in figure caption.

*- As mentioned in the previous review, model-obs Figures could easily be merged with obs-only Figures, which would make the reading much simpler. The authors have already*

*successfully done it for Figure 6, they could do the same by removing Figures 9 and 10 and putting them aside their observational counterparts.*

- Done.

*- As already mentioned before, Figure 15 is not insightful as it is now and could also be removed. The SI difference doesn't read well and plotting directly both time differences would have been easier, and most of the domain already has a very low (<0.05m²/s²) stratification before the Bora events occur. Also, displaying the MLD would be very insightful (see comment M3).*

- We changed SI to MLD in Fig. 15 (now Fig. 11).

*- The new Table 2 is too instantaneous to be insightful and could also be removed. Instead, the intense transports during Bora events could be compared in the text to the winter mean transport (e.g. on average during the 6 Bora events, the T, S and volume transports are respectively X times, Y times and Z times larger than their winter mean value). Also, mass and volume transports are almost identical so that the mass transport could be removed from Table 1.*

- We removed Table 2 and provided some data in the text as suggested. Also, we removed mass transport from Table 1.

*- The very large number of geographic locations makes the reading very confusing. In particular, the following locations should appear in Figure 1: Istria, Sedmovrace, Senj, Kvarneric (confusion between A4, T4 and Kvarneric transect).*

- We tried to remove some locations by replacing them by station/transect codes, to clarify others (e.g. Kvarnerić Channel is used everywhere instead Kvarnerić), while adding the missing locations to Figure 1 (like Senj, Istria was already there, while Istrian Peninsula is changed by Istria).

*M3 Although the authors have put forward more explicitly both hypotheses of a DWF occurring locally or dense waters from the open Adriatic shelf, I still find the conclusions confusing, including those stated in the abstract. In addition, as in coastal areas DWF occurs exclusively when the mixed layer reaches the bottom of the water column, the lack of any mixed layer computation is still a major drawback of the paper.*

- We cleared up the text, particularly abstract, and added MLD as suggested.

*- Regarding the first point, here is the conclusion I have reached, which is clearly stated nowhere in the paper : « Observations at transect T4 and station A9 show the incoming of dense water from the open shelf, and current profiles only show marginal outward transport of dense water from the NADEX area to the open shelf. They support balanced inward-outward dense water exchanges or even a dominating inward transport. However, modelling results show the domination of outward dense water transport, strongly supporting the hypothesis that the NADEX area is a source of dense waters for the open Adriatic. » Please re-formulate more clearly the respective contributions of the local formation, outward and*

*inward transport in the NADEX area to the DWF phenomenon, in particular in the abstract, the conclusion and section 4 (p.9, L.4-25).*

- We reformulated the text following the above suggestions. We hope that it sounds much better and coherent now.

*- Regarding the lack of any mixed layer computation: I believe that comments of Figure 13 are not convincing without an MLD being displayed (see also comments m9 and m11), and the color range change obviously didn't fix the issue (it is still impossible to read with the eye a 0.01kg/m³ density variation). I also believe (former M8) that Figure 15 should be removed if no mixed layer is displayed (eg: relevant contours).*

- We added mixed layer depth to Figure 13 (now Figure 10), to existing PDA Hovmoller as a thick line. The methodology for MLD computation is also added in the text. Now we present MLD instead of SI in Fig. 15.

*Figure 15 would be insightful if the IS time difference map was displayed together with a relevant MLD contour (e.g. 50m depth) before and after the Bora event, so that the reader can objectively identify the regions of DWF created by the Bora event.*

- We computed the respective MLD for dates presented in old Fig. 15 (now Fig. 11). MLD has been following the SI distribution, i.e. it is reaching the bottom at the most of the domain. However, following the suggestion of the reviewer, we put MLD instead of SI to the figure (putting both at the same figure will be too messy).

*M11 (new) The authors claim in their response to have addressed their general question 4 (What are the associated mesoscale and frontal processes associated?), which I believe is not the case: no actual mesoscale or frontal process has been studied, they have just identified a front. As questions 1 to 3 are enough, I believe question 4 should be removed.*

- We removed it as requested.

*Here is a list of the new minor concern:*

*m1 The authors have identified an interesting bottom hydrological trend in observations which is the main signature of DWF: they should evaluate it in the model.*

- Done, the model really reproduced the T-S trends at all A stations. We decided not to provide additional figures, as the reviewer comments are asking for their reduction. Some of the text was already in the manuscript, but a few statements are added to the Section 5.

*m2 p.6, L.15 Compare year 2015 to the 10-year average rather than to specific years, because the fact that it is lower than 2012 and higher than 2014 doesn't prove that year 2015 was average.*

- We also compared it to the average and we kept comparison with specific years.

*m3 p.7 L.27 How do you know that this incoming occurred dominantly through T4 connecting passage?*

- To be perfectly honest, we really don't know, and it is not perfectly clear from the Leg 2 salinity distribution, hence we omitted the statement.

*m4 p.9, L.4-5 Question 2 is not answered to in the whole manuscript (see comment M3).*

- We've changed the text, including discussion and conclusions, following the suggestion from M3. It is basically softening our conclusions and put the accent on additional research to get the answer on Question 2.

*m5 p.8, L.28 Fig. 5 not Fig. 6.*

- Corrected.

*m6 p.9, L.5 Fig. 6 not Fig. 7. The orientation of all channel cross-sections would help a lot to interpret Fig. 6 as each section has a different orientation (e.g. just display for instance a horizontal line on top of the current measurements if the section is East-West).*

- Done, inserted in Fig. 7.

*m7 p.10, L.15-16 Positive bias, not negative.*

- Corrected.

*m8 p.11, L.11-21 The model evaluation is too qualitative, be more quantitative (e.g. bias in the module and in the angle). For now it is not convincing that the model performs overall well.*

- We computed it and added details in the text.

*m9 p.11, L30 The water column was not homogeneous all the time from December to early April. Only displaying the MLD in Fig.13 would determine when it occurred (when the MLD reached the bottom). For instance, it is clear that DWF was intermittent in early January and in March.*

- We added MLD to the figure, to already existing PDA Hovmoller plot.

*m10 p.12 L.3 Also (maybe mostly) the surface radiative forcing restratifies the water column.*

- This effect is present but it is of minor importance. We rewrote the sentence.

*m11 p.12, L.3-5 I don't agree on the comparison between G1 and G2: the water column looks much more homogeneous throughout the winter in G2 than G1. Once again, displaying the MLD would determine if it is the case or not.*

- We added MLD to the figure and rewrote the text accordingly.

*m12 p.13 L.14 Why do you expect wind-driven dynamics to induce an energy loss?*

- This area has been affected by strong bora wind episodes, which has the maximum there. Yet, we haven't estimated contributions of different processes, in that sense we removed the sentence as unjustified.

*m13 p.14 In the formula, rename U vector as Uout for instance to specify that it is only the outward velocity.*

- Ok, done.

*m14 p.14 L.13-25 There are a series of English and typing mistakes, please correct them.*

- The whole manuscript is corrected by professional language-correcting service, American Journal Experts (www.aje.com).

*m15 Use the same range of values between Fig.17 and 19.*

- Corrected.

*m16 p.17 paragraph 1 Also, the lateral boundary conditions for momentum impact highly the level of kinetic energy close to the coastline. Do you have free-slip conditions? If not, this would be a means of improving the modelled velocities in the NADEX area.*

- Free-slip conditions have been imposed as the lateral boundary conditions inside the domain (land). Coastline of the eastern side of the Adriatic is complex with many narrow channels and islands acting as lateral friction, similar like stairway effect. If using additional no-slip condition this effect would be even more pronounced. We fully agree that such choice could affect dynamics and currents in channel areas, so it is added to the text.

*m17 p.17 L.15 Don't talk about one-way coupling: it is just a forced atmospheric simulation.*

- Corrected.

*m18 A schematic scheme of the main phenomena evidenced in this field campaign would help the reader to have a general overview of the main results of the study.*

- We agree that it might be useful to have some sketching of observed processes, yet this will add extra figure to the paper and we feel that such a figure is better as a graphical abstract and not as a part of the paper.

**Manuscript with annotated changes (before AJE correction):**

[revised manuscript text omitted]

**6 Model results**

**6.1 Thermohaline, buoyancy and stratification changes**

Modelled temporal changes of thermohaline properties at location G1, positioned in the Velebit Channel at the core of the Senj bora jet, and at location G2, positioned in the outer Kvarnerić Channel, are displayed in Fig. 10. Mixed layer depth (MLD), computed by using de Boyer Montégut et al. (2004) methodology with density increment threshold set to 0.125 kg/m$^3$, is reaching the bottom all time at G2 till early April, indicating that water column was vertically homogenized over most of the basin from December until early April. Later on, surface thermocline developed, deepening to about 30-40 m in early July. Salinity series at the G1 location exhibit pronounced daily and weekly changes in the upper layer, as being influenced by the nearby freshwater discharge of the Senj hydropower plant. Yet, vertical homogeneity and MLD reaching the bottom have been present there during cooling events around 30 December 2014, 5 February and 5 March 2015 - this is due to its position on the track of the Senj bora jet. Between bora events, the ocean started to relax through horizontal advection and increasing a bit stratification due to radiative forcing.

The most of the coastal area has been vertically homogeneous prior to the early February and early March bora events (Fig. 11), with MLD reaching the bottom in the most of the region. The only exceptions were coastal waters close to river mouths, like eastern Velebit Channel and off Senj power plant (where G1 station is positioned), and areas characterized by strong thermohaline fronts, like Kvarner Bay. However, both boras were substantially strong and mixed even these regions to the bottom, allowing for the DWF to occur in the whole 
[revised manuscript text omitted]
. That also includes introduction of state-of-the-art measuring platform providing an insight to processes not documented in the Adriatic, like Arvor-C profiling float deployed in the Adriatic Sea for the very first time. The choice of the experimental area – the coastal northeastern Adriatic - emerged from its role in the winter of 2012, when it exhibited an unprecedented heat loss (up to 2000 W/m$^2$) and DWF rates that strongly contributed to the overall Adriatic dense water dynamics (Mihanović et al., 2013; Janeković et al., 2014).

We have overviewed observations and estimated rates of selected basic processes, using both an extensive set of measured data and a coupled atmosphere-ocean model. Our main intention was to answer two questions: (i) whether or not the DWF is common in the northeastern coastal Adriatic, and (ii) if so, what is DWF-related dynamics behaviour, in particular regarding water exchange with the open Adriatic. To answer these questions we reached the following conclusions: (i) CTD data show that the DWF does occur in the coastal northeastern Adriatic during normal winters in terms of wintertime heat losses, as was the case of 2015; (ii) observations at connecting channels show a marginal bottom outward transport of dense

waters to the open shelf and a balanced near-bottom inward-outward dense water exchange, while the outward transport of dense waters is mostly concentrated in intermediate and surface layers, (iii) modelling results show the domination of outward dense water transport, supporting the hypothesis that the NAdEx area is a source of dense waters for the open Adriatic, and (iv) exchange of waters between coastal and open Adriatic waters through a number of connecting passages has been quantified, with residence time varying from a week to a few weeks and being shorter during strong wind conditions.

These results are also in line with previous modelling results by Janeković et al. (2014), who also modelled outflow dense water transports in connecting passages during the winter of 2012. Yet, the density of these waters has been much higher and almost equalling the density of waters coming from the open Adriatic DWF site due to different preconditioning and DWF setup in the winter of 2012, so that the DWF outflow from the NAdEx area happened also in the bottom layers of the connecting passages. Precisely, 
[revised manuscript text omitted]

[Figure]

a) **Temperature bias and error (°C)**

b) **Salinity bias and error**

Winter bias
Spring bias
Winter std.
Spring std.

**Figure 8. Vertical profiles of the bias and root-mean-square-error during Leg 1 (winter) and Leg 2 (spring) cruises.**

[Figure]

a) **4-6 Dec 2014 Transect**

depth (m)

c)

depth (m)

CTD Station

¶

a) **Model - ARVOR-C**

depth (m)

19-Feb  21-Feb  23-Feb  25-Feb  27-Feb  01-Mar  03-Mar

c)

depth (m)

19-Feb  21-Feb  23-Feb  25-Feb  27-Feb  01-Mar  03-Mar

¶
**Figure 10. Model-to-observation difference in (a, b) temperature and (c, d) salinity estimated for (a, c) Arvor-C float and (b, d) glider measurements.¶**

[Figure]

[Figure]

5    **Figure 9. Model-to-observation Q-Q plots of (a) temperature and (b) salinity measured by CTD, Arvor-C profiling float and glider, and (c) current speed, (d) temperature and (e) salinity measured at the bottom of stations A1 to A9. Each dash-dot line represent the Q-Q slope of a particular dataset.**

[Figure]

[Figure]

eleted:

**Figure 10. Hovmöller plot of modelled temperature, salinity and PDA values at grid points (a) G1 and (b) G2. Vertical dashed lines indicate three severe bora episodes discussed in Section 3, while thick black line in PDA plot shows mixed layer depth.**

**Figure 11. Daily values of surface buoyancy fluxes (BF) and its components averaged over the nested model domain.**

[Figure]

[Figure]

**Figure 12. Mixed layer depth (MLD) computed for the nested model domain prior and after the major bora episodes: 3 and 6 February 2015, and 3 and 6 March 2015. Grey colour stand for MLD reaching the bottom.**

[Figure]

Figure 13. Modelled heat and salinity fluxes normal to transects T1 to T789 and averaged between 15 December 2014 and 15 May
2015. Positive fluxes are oriented northeastward over the T2, T3, T4 and T789 transects and northwestwards over the T1, T5 and
T6 transects. Green areas denote the bathymetry.

[Figure]

**Figure 14.** Time series of heat and salt transports normal to and integrated over transects T1 to T789 between 15 December 2014 and 15 May 2015. Transports across the outer (T1+T2+T3+T4+T789) and inner (T5+T6) domains are plotted too.

[Figure]

**Figure 15. As in Fig. 13, but for dense water mass fluxes. Green areas denote the bathymetry and model cells where no dense water is modelled. T1 is omitted from the figure, as almost no dense water transport occurred there.**

[Figure]

**Figure 16. As in Fig. 14, but for dense water mass transports.**

[Figure]

**Figure 17. Box-whiskers statistics of total (RT) and dense water (DWRT) residence times for the coastal northeastern Adriatic bordered by transects T1, T2, T3, T4, and T789.**

| Page 24: [1] Deleted | | Ivica Vilibic | | | | | | 18/09/2017 14:23:00 | |
|---|---|---|---|---|---|---|---|---|---|---|
| **Mass ($10^6$ kg s$^{-1}$)** | 4.6 | -1.7 | 1.0 | -11.3 | -1.7 | -0.5 | -58.2 | -65.0 | -2.1 |

| Page 24: [2] Deleted | Ivica Vilibic | 18/09/2017 14:23:00 |
|---|---|---|

**Table 2. Temperature, salinity mass and volume sum transports at the outer (T1+T2+T3+T4+T789) and inner (T5+T6) boundaries for peak outer transports.**

| | $T_{outer}$ Salinity $10^8$ kg s$^{-1}$ | $T_{outer}$ Temp. $10^{11}$ J s$^{-1}$ | $T_{outer}$ Mass $10^6$ kg s$^{-1}$ | $T_{outer}$ Volume $10^3$ m$^3$s$^{-1}$ | $T_{inner}$ Salinity $10^8$ kg s$^{-1}$ | $T_{inner}$ Temp. $10^{11}$ J s$^{-1}$ | $T_{inner}$ Mass $10^6$ kg s$^{-1}$ | $T_{inner}$ Volume $10^3$ m$^3$ s$^{-1}$ |
|---|---|---|---|---|---|---|---|---|
| 17/12/14 09:00:00 | -113.9 | -186.5 | -299.5 | -291.3 | -1.6 | -1.3 | -4.4 | -4.3 |
| 28/12/14 15:00:00 | -137.4 | -211.4 | -362.3 | -352.4 | -3.8 | -0.4 | -10.1 | -9.8 |
| 17/01/15 09:00:00 | -112.9 | -160.9 | -295.6 | -287.3 | 0.7 | -1.9 | 1.7 | 1.6 |
| 06/02/15 12:00:00 | -113.8 | -142.1 | -298.2 | -289.8 | 0.7 | 4.9 | 1.9 | 1.8 |
| 22/02/15 21:00:00 | -279.3 | -347.2 | -728.6 | -707.9 | -1.5 | -1.5 | -3.9 | -3.8 |
| 05/03/15 09:00:00 | -118.6 | -145.0 | -310.1 | -301.3 | -2.1 | -1.1 | -5.6 | -5.4 |

| Page 33: [3] Deleted | Ivica Vilibic | 20/09/2017 10:35:00 |
|---|---|---|

| Page 33: [3] Deleted | Ivica Vilibic | 20/09/2017 10:35:00 |
|---|---|---|

| Page 33: [3] Deleted | Ivica Vilibic | 20/09/2017 10:35:00 |
|---|---|---|

| Page 33: [3] Deleted | Ivica Vilibic | 20/09/2017 10:35:00 |
|---|---|---|

| Page 33: [3] Deleted | Ivica Vilibic | 20/09/2017 10:35:00 |
|---|---|---|

| Page 33: [3] Deleted | Ivica Vilibic | 20/09/2017 10:35:00 |
|---|---|---|

| Page 33: [4] Deleted | Ivica Vilibic | 20/09/2017 10:40:00 |
|---|---|---|

[revised manuscript text omitted]
³. A similar high density layer of a few metres that was found in mid-February 2012 on the western Adriatic shelf was associated with the initial near-bottom outflow of dense waters from the northern Adriatic (Vilibić and Mihanović, 2013). Analogously, our hypothesis is that a small fraction of dense waters generated on the northern Adriatic shelf, these waters had higher densities than the waters of Kvarner Bay (Janeković et al., 2014), and spread towards deeper Kvarner Bay as a weak bottom density current.

15 The assessment of the wintertime ADCP data (Fig. 7) revealed a substantial baroclinic component atop the barotropic circulation at all stations during and after two strong bora episodes (1 February – 1 April 2015), when the DWF and dense water flow were expected to occur. Weaker currents at station A9 and strong outflow in the surface layers at stations A7 and A8 indicate the presence of an anticyclonic curl at the entrance of Kvarner Bay. The pattern in currents also resembles the patterns of the local wind stress and wind curl, which are pronounced off the southern tip of Istria (Pullen et al., 2003; Grubišić,

20 2004). Inflow to Kvarner Bay may be seen in the bottom layer of station A9, part of this inflow might be ascribed to dense waters coming from the northern Adriatic shelf, thus confirming our hypothesis of the origin of the bottom density current seen in the Arvor-C data. Near the bottom of stations A7 and A8, the mean flow is weak, yet changeable over time, suggesting an interplay between dense waters coming from the coastal area with those coming from the northern Adriatic shelf in the Kvarner Bay area. At the same time, much stronger currents were observed in the surface and intermediate layers, indicating

25 the predominant outflow of waters from the NAdEx area towards the open Adriatic. Going to the southeast, at station A4, the measured currents were parallel to the coastline. However, as it was deployed too far from the connecting channel, this station was indeed not measuring the interchange between the coastal and open Adriatic waters but the Eastern Adriatic Current, which may be strong in that region (Orlić et al., 2006). The currents measured at station A2 exhibit a strong baroclinic pattern, pointing to an exchange of waters between the open and coastal Adriatic through a narrow channel: an outflow current is

30 present in the surface layer, and an inflow current is present near the bottom. However, these currents are strongly affected by local bathymetry, probably resembling the effects of both a very narrow connecting channel (approximately 600 m in width) and the Eastern Adriatic Current modulated by the cape of Veli Rat (approximately 2 km south of station A2). Finally, the current data measured at station A1 document the predominant inflow of the open Adriatic waters, mostly in the surface layer. The inflow is likely driven by the orientation of the channel and the incoming Eastern Adriatic Current. Interestingly, the

**Commented [Ed4]:** Please ensure that the intended meaning has been maintained in this edit.

**Commented [Ed5]:** Please ensure that the intended meaning has been maintained in this edit.

[revised manuscript text omitted]
) observations at connecting channels show a marginal bottom outward transport of dense waters to the open shelf and a balanced near-bottom inward-outward dense water exchange, while the outward transport of dense waters is mostly concentrated in the intermediate and surface layers, (iii) the modelling results show the domination of outward dense water transport, supporting the hypothesis that the NAdEx area is a source of dense waters for the open Adriatic, and (iv) the exchange of waters between the coastal and open Adriatic waters through a number

15    of connecting passages was quantified, and the residence time was found to vary from one week to a few weeks and was shorter during strong wind conditions.

       These results are also in line with the previous modelling results by Janeković et al. (2014), who also modelled the outflow of dense water transports in connecting passages during the winter of 2012. However, the densities of these waters were much higher and almost equal to the densities of the waters coming from the open Adriatic DWF site due to the different

20    preconditioning and DWF setups in the winter of 2012. Thus, the DWF outflow from the NAdEx area also occurred in the bottom layers of the connecting passages. There are several reasons for this result: (
[revised manuscript text omitted]

| Page 3: [1] Deleted | Editor | 28/09/2017 18:32:00 |
|---|---|---|

has been

| Page 3: [1] Deleted | Editor | 28/09/2017 18:32:00 |
|---|---|---|

has been

| Page 3: [1] Deleted | Editor | 28/09/2017 18:32:00 |
|---|---|---|

has been

| Page 3: [1] Deleted | Editor | 28/09/2017 18:32:00 |
|---|---|---|

has been

| Page 3: [1] Deleted | Editor | 28/09/2017 18:32:00 |
|---|---|---|

has been

| Page 3: [1] Deleted | Editor | 28/09/2017 18:32:00 |
|---|---|---|

has been

| Page 3: [1] Deleted | Editor | 28/09/2017 18:32:00 |
|---|---|---|

has been

| Page 3: [1] Deleted | Editor | 28/09/2017 18:32:00 |
|---|---|---|

has been

| Page 3: [1] Deleted | Editor | 28/09/2017 18:32:00 |
|---|---|---|

has been

| Page 3: [1] Deleted | Editor | 28/09/2017 18:32:00 |
|---|---|---|

has been

| Page 3: [1] Deleted | Editor | 28/09/2017 18:32:00 |
|---|---|---|

has been

| Page 3: [1] Deleted | Editor | 28/09/2017 18:32:00 |
|---|---|---|

has been

| Page 3: [1] Deleted | Editor | 28/09/2017 18:32:00 |
|---|---|---|

has been

| Page 3: [1] Deleted | Editor | 28/09/2017 18:32:00 |
|---|---|---|

has been

| Page 3: [1] Deleted | Editor | 28/09/2017 18:32:00 |
|---|---|---|

has been

| Page 3: [2] Deleted | Editor | 28/09/2017 18:36:00 |
|---|---|---|

Up till

| Page 3: [2] Deleted | Editor | 28/09/2017 18:36:00 |

Up till

| Page 3: [2] Deleted | Editor | 28/09/2017 18:36:00 |

Up till

| Page 3: [2] Deleted | Editor | 28/09/2017 18:36:00 |

Up till

| Page 3: [2] Deleted | Editor | 28/09/2017 18:36:00 |

Up till

| Page 3: [2] Deleted | Editor | 28/09/2017 18:36:00 |

Up till

| Page 3: [2] Deleted | Editor | 28/09/2017 18:36:00 |

Up till

| Page 3: [2] Deleted | Editor | 28/09/2017 18:36:00 |

Up till

| Page 3: [2] Deleted | Editor | 28/09/2017 18:36:00 |

Up till

| Page 3: [2] Deleted | Editor | 28/09/2017 18:36:00 |

Up till

| Page 3: [2] Deleted | Editor | 28/09/2017 18:36:00 |

Up till

| Page 3: [2] Deleted | Editor | 28/09/2017 18:36:00 |

Up till

| Page 3: [2] Deleted | Editor | 28/09/2017 18:36:00 |

Up till

| Page 3: [2] Deleted | Editor | 28/09/2017 18:36:00 |

Up till

| Page 3: [2] Deleted | Editor | 28/09/2017 18:36:00 |

Up till

| Page 3: [2] Deleted | Editor | 28/09/2017 18:36:00 |

Up till

| Page 3: [2] Deleted | Editor | 28/09/2017 18:36:00 |

Up till

| Page 3: [2] Deleted | Editor | 28/09/2017 18:36:00 |

Up till

| Page 3: [2] Deleted | Editor | 28/09/2017 18:36:00 |

Up till

| Page 3: [2] Deleted | Editor | 28/09/2017 18:36:00 |

Up till

| Page 3: [2] Deleted | Editor | 28/09/2017 18:36:00 |

Up till

| Page 3: [2] Deleted | Editor | 28/09/2017 18:36:00 |

Up till

| Page 3: [3] Deleted | Editor | 28/09/2017 18:40:00 |

gives

| Page 3: [3] Deleted | Editor | 28/09/2017 18:40:00 |

gives

| Page 3: [4] Deleted | Editor | 28/09/2017 18:42:00 |

a number of

| Page 3: [4] Deleted | Editor | 28/09/2017 18:42:00 |

a number of

| Page 3: [4] Deleted | Editor | 28/09/2017 18:42:00 |

a number of

| Page 3: [4] Deleted | Editor | 28/09/2017 18:42:00 |

a number of

| Page 4: [5] Deleted | Editor | 28/09/2017 18:42:00 |

connecting

| Page 4: [5] Deleted | Editor | 28/09/2017 18:42:00 |

connecting

| Page 4: [5] Deleted | Editor | 28/09/2017 18:42:00 |

connecting

| Page 4: [5] Deleted | Editor | 28/09/2017 18:42:00 |

connecting

| Page 4: [5] Deleted | Editor | 28/09/2017 18:42:00 |

connecting

| Page 4: [5] Deleted | Editor | 28/09/2017 18:42:00 |

connecting

| Page 4: [5] Deleted | Editor | 28/09/2017 18:42:00 |
|---|---|---|

connecting

| Page 4: [5] Deleted | Editor | 28/09/2017 18:42:00 |
|---|---|---|

connecting

| Page 4: [5] Deleted | Editor | 28/09/2017 18:42:00 |
|---|---|---|

connecting

| Page 4: [5] Deleted | Editor | 28/09/2017 18:42:00 |
|---|---|---|

connecting

| Page 4: [5] Deleted | Editor | 28/09/2017 18:42:00 |
|---|---|---|

connecting

| Page 4: [5] Deleted | Editor | 28/09/2017 18:42:00 |
|---|---|---|

connecting

| Page 4: [5] Deleted | Editor | 28/09/2017 18:42:00 |
|---|---|---|

connecting

| Page 4: [5] Deleted | Editor | 28/09/2017 18:42:00 |
|---|---|---|

connecting

| Page 4: [5] Deleted | Editor | 28/09/2017 18:42:00 |
|---|---|---|

connecting

| Page 4: [5] Deleted | Editor | 28/09/2017 18:42:00 |
|---|---|---|

connecting

| Page 4: [6] Deleted | Editor | 28/09/2017 18:46:00 |
|---|---|---|

experiment

| Page 4: [6] Deleted | Editor | 28/09/2017 18:46:00 |
|---|---|---|

experiment

| Page 4: [6] Deleted | Editor | 28/09/2017 18:46:00 |
|---|---|---|

experiment

| Page 4: [6] Deleted | Editor | 28/09/2017 18:46:00 |
|---|---|---|

experiment

| Page 4: [6] Deleted | Editor | 28/09/2017 18:46:00 |
|---|---|---|

experiment

| Page 4: [6] Deleted | Editor | 28/09/2017 18:46:00 |
|---|---|---|

experiment

| Page 4: [6] Deleted | Editor | 28/09/2017 18:46:00 |
|---|---|---|

experiment

| Page 4: [6] Deleted | Editor | 28/09/2017 18:46:00 |
|---|---|---|

experiment

| Page 4: [6] Deleted | Editor | 28/09/2017 18:46:00 |
|---|---|---|

experiment

| Page 4: [6] Deleted | Editor | 28/09/2017 18:46:00 |
|---|---|---|

experiment

| Page 4: [6] Deleted | Editor | 28/09/2017 18:46:00 |
|---|---|---|

experiment

| Page 4: [6] Deleted | Editor | 28/09/2017 18:46:00 |
|---|---|---|

experiment

| Page 4: [6] Deleted | Editor | 28/09/2017 18:46:00 |
|---|---|---|

experiment

| Page 4: [6] Deleted | Editor | 28/09/2017 18:46:00 |
|---|---|---|

experiment

| Page 4: [7] Deleted | Editor | 28/09/2017 18:50:00 |
|---|---|---|

and

| Page 4: [7] Deleted | Editor | 28/09/2017 18:50:00 |
|---|---|---|

and

| Page 4: [8] Deleted | Editor | 28/09/2017 18:52:00 |
|---|---|---|

,

| Page 4: [8] Deleted | Editor | 28/09/2017 18:52:00 |
|---|---|---|

,

| Page 4: [8] Deleted | Editor | 28/09/2017 18:52:00 |
|---|---|---|

,

| Page 4: [8] Deleted | Editor | 28/09/2017 18:52:00 |
|---|---|---|

,

| Page 4: [8] Deleted | Editor | 28/09/2017 18:52:00 |
|---|---|---|

,

| Page 4: [8] Deleted | Editor | 28/09/2017 18:52:00 |
|---|---|---|

,

| Page 4: [8] Deleted | Editor | 28/09/2017 18:52:00 |
|---|---|---|

,

| Page 4: [8] Deleted | Editor | 28/09/2017 18:52:00 |
|---|---|---|

,

| Page 4: [8] Deleted | Editor | 28/09/2017 18:52:00 |

,

| Page 4: [8] Deleted | Editor | 28/09/2017 18:52:00 |

,

| Page 4: [8] Deleted | Editor | 28/09/2017 18:52:00 |

,

| Page 4: [8] Deleted | Editor | 28/09/2017 18:52:00 |

,

| Page 8: [9] Deleted | Editor | 28/09/2017 19:28:00 |

L

| Page 8: [9] Deleted | Editor | 28/09/2017 19:28:00 |

L

| Page 8: [9] Deleted | Editor | 28/09/2017 19:28:00 |

L

| Page 8: [9] Deleted | Editor | 28/09/2017 19:28:00 |

L

| Page 8: [9] Deleted | Editor | 28/09/2017 19:28:00 |

L

| Page 8: [9] Deleted | Editor | 28/09/2017 19:28:00 |

L

| Page 8: [9] Deleted | Editor | 28/09/2017 19:28:00 |

L

| Page 8: [9] Deleted | Editor | 28/09/2017 19:28:00 |

L

| Page 8: [9] Deleted | Editor | 28/09/2017 19:28:00 |

L

| Page 8: [9] Deleted | Editor | 28/09/2017 19:28:00 |

L

| Page 8: [9] Deleted | Editor | 28/09/2017 19:28:00 |

L

| Page 8: [9] Deleted | Editor | 28/09/2017 19:28:00 |

L

| Page 8: [9] Deleted | Editor | 28/09/2017 19:28:00 |

L

| Page 8: [9] Deleted | Editor | 28/09/2017 19:28:00 |

L

| Page 8: [9] Deleted | Editor | 28/09/2017 19:28:00 |

L

| Page 8: [9] Deleted | Editor | 28/09/2017 19:28:00 |

L

| Page 8: [9] Deleted | Editor | 28/09/2017 19:28:00 |

L

| Page 8: [9] Deleted | Editor | 28/09/2017 19:28:00 |

L

| Page 8: [9] Deleted | Editor | 28/09/2017 19:28:00 |

L

| Page 8: [9] Deleted | Editor | 28/09/2017 19:28:00 |

L

| Page 8: [10] Deleted | Editor | 29/09/2017 09:48:00 |

a

| Page 8: [10] Deleted | Editor | 29/09/2017 09:48:00 |

a

| Page 8: [10] Deleted | Editor | 29/09/2017 09:48:00 |

a

| Page 8: [10] Deleted | Editor | 29/09/2017 09:48:00 |

a

| Page 8: [10] Deleted | Editor | 29/09/2017 09:48:00 |

a

| Page 8: [10] Deleted | Editor | 29/09/2017 09:48:00 |

a

| Page 8: [10] Deleted | Editor | 29/09/2017 09:48:00 |

a

| Page 8: [10] Deleted | Editor | 29/09/2017 09:48:00 |

a

| Page 8: [10] Deleted | Editor | 29/09/2017 09:48:00 |

a

| Page 8: [10] Deleted | Editor | 29/09/2017 09:48:00 |

a

| Page 8: [10] Deleted | Editor | 29/09/2017 09:48:00 |

a

| Page 8: [10] Deleted | Editor | 29/09/2017 09:48:00 |

a

| Page 8: [10] Deleted | Editor | 29/09/2017 09:48:00 |

a

| Page 8: [10] Deleted | Editor | 29/09/2017 09:48:00 |

a

| Page 8: [10] Deleted | Editor | 29/09/2017 09:48:00 |

a

| Page 8: [10] Deleted | Editor | 29/09/2017 09:48:00 |

a

| Page 8: [10] Deleted | Editor | 29/09/2017 09:48:00 |

a

| Page 8: [10] Deleted | Editor | 29/09/2017 09:48:00 |

a

| Page 8: [10] Deleted | Editor | 29/09/2017 09:48:00 |

a

| Page 8: [10] Deleted | Editor | 29/09/2017 09:48:00 |

a

| Page 8: [10] Deleted | Editor | 29/09/2017 09:48:00 |

a

| Page 8: [10] Deleted | Editor | 29/09/2017 09:48:00 |

a

| Page 8: [10] Deleted | Editor | 29/09/2017 09:48:00 |

a

| Page 8: [10] Deleted | Editor | 29/09/2017 09:48:00 |

a

| Page 8: [11] Deleted | Editor | 29/09/2017 09:53:00 |
| --- | --- | --- |

As for b

| Page 8: [11] Deleted | Editor | 29/09/2017 09:53:00 |
| --- | --- | --- |

As for b

| Page 8: [11] Deleted | Editor | 29/09/2017 09:53:00 |
| --- | --- | --- |

As for b

| Page 8: [11] Deleted | Editor | 29/09/2017 09:53:00 |
| --- | --- | --- |

As for b

| Page 8: [11] Deleted | Editor | 29/09/2017 09:53:00 |
| --- | --- | --- |

As for b

| Page 8: [11] Deleted | Editor | 29/09/2017 09:53:00 |
| --- | --- | --- |

As for b

| Page 8: [11] Deleted | Editor | 29/09/2017 09:53:00 |
| --- | --- | --- |

As for b

| Page 8: [11] Deleted | Editor | 29/09/2017 09:53:00 |
| --- | --- | --- |

As for b

| Page 8: [11] Deleted | Editor | 29/09/2017 09:53:00 |
| --- | --- | --- |

As for b

| Page 8: [11] Deleted | Editor | 29/09/2017 09:53:00 |
| --- | --- | --- |

As for b

| Page 8: [11] Deleted | Editor | 29/09/2017 09:53:00 |
| --- | --- | --- |

As for b

| Page 8: [11] Deleted | Editor | 29/09/2017 09:53:00 |
| --- | --- | --- |

As for b

| Page 8: [11] Deleted | Editor | 29/09/2017 09:53:00 |
| --- | --- | --- |

As for b

| Page 8: [11] Deleted | Editor | 29/09/2017 09:53:00 |
| --- | --- | --- |

As for b

| Page 9: [12] Deleted | Editor | 29/09/2017 09:57:00 |
| --- | --- | --- |

the

| Page 9: [12] Deleted | Editor | 29/09/2017 09:57:00 |
| --- | --- | --- |

the

| Page 9: [12] Deleted | Editor | 29/09/2017 09:57:00 |
| --- | --- | --- |

the

| Page 9: [12] Deleted | Editor | 29/09/2017 09:57:00 |

the

| Page 9: [12] Deleted | Editor | 29/09/2017 09:57:00 |

the

| Page 9: [12] Deleted | Editor | 29/09/2017 09:57:00 |

the

| Page 9: [12] Deleted | Editor | 29/09/2017 09:57:00 |

the

| Page 9: [12] Deleted | Editor | 29/09/2017 09:57:00 |

the

| Page 9: [12] Deleted | Editor | 29/09/2017 09:57:00 |

the

| Page 9: [12] Deleted | Editor | 29/09/2017 09:57:00 |

the

| Page 9: [12] Deleted | Editor | 29/09/2017 09:57:00 |

the

| Page 9: [12] Deleted | Editor | 29/09/2017 09:57:00 |

the

| Page 9: [13] Deleted | Editor | 29/09/2017 10:00:00 |

which

| Page 9: [13] Deleted | Editor | 29/09/2017 10:00:00 |

which

| Page 9: [13] Deleted | Editor | 29/09/2017 10:00:00 |

which

| Page 9: [13] Deleted | Editor | 29/09/2017 10:00:00 |

which

| Page 9: [13] Deleted | Editor | 29/09/2017 10:00:00 |

which

| Page 9: [13] Deleted | Editor | 29/09/2017 10:00:00 |

which

| Page 9: [13] Deleted | Editor | 29/09/2017 10:00:00 |

which

| Page 9: [14] Deleted | Editor | 29/09/2017 10:01:00 |

A

| Page 9: [14] Deleted | Editor | 29/09/2017 10:01:00 |

A

| Page 9: [14] Deleted | Editor | 29/09/2017 10:01:00 |

A

| Page 9: [15] Deleted | Editor | 29/09/2017 10:02:00 |

stations in the surface layer

| Page 9: [15] Deleted | Editor | 29/09/2017 10:02:00 |

stations in the surface layer

| Page 9: [15] Deleted | Editor | 29/09/2017 10:02:00 |

stations in the surface layer

| Page 9: [15] Deleted | Editor | 29/09/2017 10:02:00 |

stations in the surface layer

| Page 9: [15] Deleted | Editor | 29/09/2017 10:02:00 |

stations in the surface layer

| Page 9: [15] Deleted | Editor | 29/09/2017 10:02:00 |

stations in the surface layer

| Page 9: [15] Deleted | Editor | 29/09/2017 10:02:00 |

stations in the surface layer

| Page 9: [15] Deleted | Editor | 29/09/2017 10:02:00 |

stations in the surface layer

| Page 9: [15] Deleted | Editor | 29/09/2017 10:02:00 |

stations in the surface layer

| Page 9: [15] Deleted | Editor | 29/09/2017 10:02:00 |

stations in the surface layer

| Page 9: [15] Deleted | Editor | 29/09/2017 10:02:00 |

stations in the surface layer

| Page 9: [15] Deleted | Editor | 29/09/2017 10:02:00 |

stations in the surface layer

| Page 9: [15] Deleted | Editor | 29/09/2017 10:02:00 |

stations in the surface layer

| Page 9: [15] Deleted | Editor | 29/09/2017 10:02:00 |
|---|---|---|

stations in the surface layer

| Page 10: [16] Deleted | Editor | 29/09/2017 10:17:00 |
|---|---|---|

the

| Page 10: [16] Deleted | Editor | 29/09/2017 10:17:00 |
|---|---|---|

the

| Page 10: [16] Deleted | Editor | 29/09/2017 10:17:00 |
|---|---|---|

the

| Page 10: [17] Deleted | Editor | 29/09/2017 10:24:00 |
|---|---|---|

V

| Page 10: [17] Deleted | Editor | 29/09/2017 10:24:00 |
|---|---|---|

V

| Page 10: [17] Deleted | Editor | 29/09/2017 10:24:00 |
|---|---|---|

V

| Page 10: [17] Deleted | Editor | 29/09/2017 10:24:00 |
|---|---|---|

V

| Page 10: [17] Deleted | Editor | 29/09/2017 10:24:00 |
|---|---|---|

V

| Page 10: [17] Deleted | Editor | 29/09/2017 10:24:00 |
|---|---|---|

V

| Page 10: [17] Deleted | Editor | 29/09/2017 10:24:00 |
|---|---|---|

V

| Page 10: [18] Deleted | Editor | 29/09/2017 10:26:00 |
|---|---|---|

the L

| Page 10: [18] Deleted | Editor | 29/09/2017 10:26:00 |
|---|---|---|

the L

| Page 10: [18] Deleted | Editor | 29/09/2017 10:26:00 |
|---|---|---|

the L

| Page 10: [18] Deleted | Editor | 29/09/2017 10:26:00 |
|---|---|---|

the L

| Page 10: [18] Deleted | Editor | 29/09/2017 10:26:00 |
|---|---|---|

the L

| Page 10: [18] Deleted | Editor | 29/09/2017 10:26:00 |
|---|---|---|

the L

| Page 10: [18] Deleted | Editor | 29/09/2017 10:26:00 |
|---|---|---|

the L

| Page 10: [18] Deleted | Editor | 29/09/2017 10:26:00 |
|---|---|---|

the L

| Page 10: [18] Deleted | Editor | 29/09/2017 10:26:00 |
|---|---|---|

the L

| Page 10: [18] Deleted | Editor | 29/09/2017 10:26:00 |
|---|---|---|

the L

| Page 10: [18] Deleted | Editor | 29/09/2017 10:26:00 |
|---|---|---|

the L

| Page 10: [18] Deleted | Editor | 29/09/2017 10:26:00 |
|---|---|---|

the L

| Page 10: [18] Deleted | Editor | 29/09/2017 10:26:00 |
|---|---|---|

the L

| Page 10: [18] Deleted | Editor | 29/09/2017 10:26:00 |
|---|---|---|

the L

| Page 10: [18] Deleted | Editor | 29/09/2017 10:26:00 |
|---|---|---|

the L

| Page 10: [18] Deleted | Editor | 29/09/2017 10:26:00 |
|---|---|---|

the L

| Page 10: [18] Deleted | Editor | 29/09/2017 10:26:00 |
|---|---|---|

the L

| Page 10: [18] Deleted | Editor | 29/09/2017 10:26:00 |
|---|---|---|

the L

| Page 10: [18] Deleted | Editor | 29/09/2017 10:26:00 |
|---|---|---|

the L

| Page 10: [19] Deleted | Editor | 29/09/2017 10:31:00 |
|---|---|---|

M

| Page 10: [19] Deleted | Editor | 29/09/2017 10:31:00 |
|---|---|---|

M

| Page 10: [19] Deleted | Editor | 29/09/2017 10:31:00 |
|---|---|---|

M

| Page 10: [19] Deleted | Editor | 29/09/2017 10:31:00 |
|---|---|---|

M

| Page 10: [19] Deleted | Editor | 29/09/2017 10:31:00 |
|---|---|---|

M

| Page 10: [19] Deleted | Editor | 29/09/2017 10:31:00 |
|---|---|---|

M

| Page 10: [19] Deleted | Editor | 29/09/2017 10:31:00 |
|---|---|---|

M

| Page 10: [19] Deleted | Editor | 29/09/2017 10:31:00 |
|---|---|---|

M

| Page 10: [19] Deleted | Editor | 29/09/2017 10:31:00 |
|---|---|---|

M

| Page 10: [19] Deleted | Editor | 29/09/2017 10:31:00 |
|---|---|---|

M

| Page 10: [19] Deleted | Editor | 29/09/2017 10:31:00 |
|---|---|---|

M

| Page 10: [19] Deleted | Editor | 29/09/2017 10:31:00 |
|---|---|---|

M

| Page 11: [20] Deleted | Editor | 29/09/2017 10:34:00 |
|---|---|---|

M

| Page 11: [20] Deleted | Editor | 29/09/2017 10:34:00 |
|---|---|---|

M

| Page 11: [20] Deleted | Editor | 29/09/2017 10:34:00 |
|---|---|---|

M

| Page 11: [20] Deleted | Editor | 29/09/2017 10:34:00 |
|---|---|---|

M

| Page 11: [20] Deleted | Editor | 29/09/2017 10:34:00 |
|---|---|---|

M

| Page 11: [20] Deleted | Editor | 29/09/2017 10:34:00 |
|---|---|---|

M

| Page 11: [20] Deleted | Editor | 29/09/2017 10:34:00 |
|---|---|---|

M

| Page 11: [20] Deleted | Editor | 29/09/2017 10:34:00 |
|---|---|---|

M

| Page 11: [20] Deleted | Editor | 29/09/2017 10:34:00 |
|---|---|---|

M

| Page 11: [20] Deleted | Editor | 29/09/2017 10:34:00 |
|---|---|---|

M

| Page 11: [20] Deleted | Editor | 29/09/2017 10:34:00 |
|---|---|---|

M

| Page 11: [20] Deleted | Editor | 29/09/2017 10:34:00 |
|---|---|---|

M

| Page 11: [20] Deleted | Editor | 29/09/2017 10:34:00 |
|---|---|---|

M

| Page 11: [20] Deleted | Editor | 29/09/2017 10:34:00 |
|---|---|---|

M

| Page 11: [20] Deleted | Editor | 29/09/2017 10:34:00 |
|---|---|---|

M

| Page 11: [20] Deleted | Editor | 29/09/2017 10:34:00 |
|---|---|---|

M

| Page 11: [20] Deleted | Editor | 29/09/2017 10:34:00 |
|---|---|---|

M

| Page 11: [20] Deleted | Editor | 29/09/2017 10:34:00 |
|---|---|---|

M

| Page 11: [21] Deleted | Editor | 29/09/2017 10:37:00 |
|---|---|---|

C

| Page 11: [21] Deleted | Editor | 29/09/2017 10:37:00 |
|---|---|---|

C

| Page 11: [21] Deleted | Editor | 29/09/2017 10:37:00 |
|---|---|---|

C

| Page 11: [21] Deleted | Editor | 29/09/2017 10:37:00 |
|---|---|---|

C

| Page 11: [21] Deleted | Editor | 29/09/2017 10:37:00 |
|---|---|---|

C

| Page 11: [21] Deleted | Editor | 29/09/2017 10:37:00 |
|---|---|---|

C

| Page 11: [21] Deleted | Editor | 29/09/2017 10:37:00 |
|---|---|---|

C

| Page 11: [21] Deleted | Editor | 29/09/2017 10:37:00 |
|---|---|---|

C

| Page 11: [21] Deleted | Editor | 29/09/2017 10:37:00 |
|---|---|---|

C

| Page 11: [21] Deleted | Editor | 29/09/2017 10:37:00 |
|---|---|---|

C

| Page 11: [21] Deleted | Editor | 29/09/2017 10:37:00 |
|---|---|---|

C

| Page 11: [21] Deleted | Editor | 29/09/2017 10:37:00 |
|---|---|---|

C

| Page 11: [21] Deleted | Editor | 29/09/2017 10:37:00 |
|---|---|---|

C

| Page 11: [21] Deleted | Editor | 29/09/2017 10:37:00 |
|---|---|---|

C

| Page 11: [21] Deleted | Editor | 29/09/2017 10:37:00 |
|---|---|---|

C

| Page 11: [21] Deleted | Editor | 29/09/2017 10:37:00 |
|---|---|---|

C

| Page 11: [21] Deleted | Editor | 29/09/2017 10:37:00 |
|---|---|---|

C

| Page 11: [21] Deleted | Editor | 29/09/2017 10:37:00 |
|---|---|---|

C

| Page 11: [21] Deleted | Editor | 29/09/2017 10:37:00 |
|---|---|---|

C

| Page 11: [21] Deleted | Editor | 29/09/2017 10:37:00 |
|---|---|---|

C

| Page 11: [21] Deleted | Editor | 29/09/2017 10:37:00 |
|---|---|---|

C

| Page 11: [21] Deleted | Editor | 29/09/2017 10:37:00 |
|---|---|---|

C

| Page 11: [21] Deleted | Editor | 29/09/2017 10:37:00 |
|---|---|---|

C

| Page 11: [21] Deleted | Editor | 29/09/2017 10:37:00 |
|---|---|---|

C

| Page 11: [21] Deleted | Editor | 29/09/2017 10:37:00 |
|---|---|---|

C

| Page 11: [21] Deleted | Editor | 29/09/2017 10:37:00 |
|---|---|---|

C

| Page 11: [21] Deleted | Editor | 29/09/2017 10:37:00 |
|---|---|---|

C

| Page 11: [21] Deleted | Editor | 29/09/2017 10:37:00 |
|---|---|---|

C

| Page 11: [21] Deleted | Editor | 29/09/2017 10:37:00 |
|---|---|---|

C

| Page 11: [21] Deleted | Editor | 29/09/2017 10:37:00 |
|---|---|---|

C

| Page 11: [21] Deleted | Editor | 29/09/2017 10:37:00 |
|---|---|---|

C

| Page 11: [21] Deleted | Editor | 29/09/2017 10:37:00 |
|---|---|---|

C

| Page 11: [21] Deleted | Editor | 29/09/2017 10:37:00 |
|---|---|---|

C

| Page 11: [21] Deleted | Editor | 29/09/2017 10:37:00 |
|---|---|---|

C

| Page 11: [21] Deleted | Editor | 29/09/2017 10:37:00 |
|---|---|---|

C

| Page 11: [21] Deleted | Editor | 29/09/2017 10:37:00 |
|---|---|---|

C

| Page 11: [22] Deleted | Editor | 29/09/2017 10:58:00 |
|---|---|---|

being

| Page 11: [22] Deleted | Editor | 29/09/2017 10:58:00 |
being

| Page 11: [22] Deleted | Editor | 29/09/2017 10:58:00 |
being

| Page 11: [22] Deleted | Editor | 29/09/2017 10:58:00 |
being

| Page 11: [22] Deleted | Editor | 29/09/2017 10:58:00 |
being

| Page 11: [22] Deleted | Editor | 29/09/2017 10:58:00 |
being

| Page 11: [22] Deleted | Editor | 29/09/2017 10:58:00 |
being

| Page 11: [22] Deleted | Editor | 29/09/2017 10:58:00 |
being

| Page 11: [22] Deleted | Editor | 29/09/2017 10:58:00 |
being

| Page 11: [22] Deleted | Editor | 29/09/2017 10:58:00 |
being

| Page 11: [22] Deleted | Editor | 29/09/2017 10:58:00 |
being

| Page 11: [22] Deleted | Editor | 29/09/2017 10:58:00 |
being

| Page 11: [22] Deleted | Editor | 29/09/2017 10:58:00 |
being

| Page 11: [22] Deleted | Editor | 29/09/2017 10:58:00 |
being

| Page 11: [22] Deleted | Editor | 29/09/2017 10:58:00 |
being

| Page 11: [22] Deleted | Editor | 29/09/2017 10:58:00 |
being

| Page 11: [22] Deleted | Editor | 29/09/2017 10:58:00 |
|---|---|---|
| being | | |

| Page 12: [23] Deleted | Editor | 29/09/2017 11:02:00 |
|---|---|---|
| the | | |

| Page 12: [24] Deleted | Editor | 28/09/2017 18:07:00 |
|---|---|---|
| In conclusions, | | |

| Page 12: [24] Deleted | Editor | 28/09/2017 18:07:00 |
|---|---|---|

In conclusions,

| Page 12: [24] Deleted | Editor | 28/09/2017 18:07:00 |
|---|---|---|

In conclusions,

| Page 12: [24] Deleted | Editor | 28/09/2017 18:07:00 |
|---|---|---|

In conclusions,

| Page 12: [24] Deleted | Editor | 28/09/2017 18:07:00 |
|---|---|---|

In conclusions,

| Page 12: [24] Deleted | Editor | 28/09/2017 18:07:00 |
|---|---|---|

In conclusions,

| Page 12: [25] Deleted | Editor | 29/09/2017 11:06:00 |
|---|---|---|

M

| Page 12: [25] Deleted | Editor | 29/09/2017 11:06:00 |
|---|---|---|

M

| Page 12: [25] Deleted | Editor | 29/09/2017 11:06:00 |
|---|---|---|

M

| Page 12: [25] Deleted | Editor | 29/09/2017 11:06:00 |
|---|---|---|

M

| Page 12: [25] Deleted | Editor | 29/09/2017 11:06:00 |
|---|---|---|

M

| Page 12: [25] Deleted | Editor | 29/09/2017 11:06:00 |
|---|---|---|

M

| Page 12: [25] Deleted | Editor | 29/09/2017 11:06:00 |
|---|---|---|

M

| Page 12: [25] Deleted | Editor | 29/09/2017 11:06:00 |
|---|---|---|

M

| Page 12: [25] Deleted | Editor | 29/09/2017 11:06:00 |
|---|---|---|

M

| Page 12: [25] Deleted | Editor | 29/09/2017 11:06:00 |
|---|---|---|

M

| Page 12: [25] Deleted | Editor | 29/09/2017 11:06:00 |
|---|---|---|

M

| Page 12: [25] Deleted | Editor | 29/09/2017 11:06:00 |
|---|---|---|

M

| Page 12: [25] Deleted | Editor | 29/09/2017 11:06:00 |
|---|---|---|

M

| Page 12: [25] Deleted | Editor | 29/09/2017 11:06:00 |
|---|---|---|

M

| Page 12: [25] Deleted | Editor | 29/09/2017 11:06:00 |
|---|---|---|

M

| Page 12: [25] Deleted | Editor | 29/09/2017 11:06:00 |
|---|---|---|

M

| Page 12: [25] Deleted | Editor | 29/09/2017 11:06:00 |
|---|---|---|

M

| Page 12: [25] Deleted | Editor | 29/09/2017 11:06:00 |
|---|---|---|

M

| Page 12: [25] Deleted | Editor | 29/09/2017 11:06:00 |
|---|---|---|

M

| Page 12: [25] Deleted | Editor | 29/09/2017 11:06:00 |
|---|---|---|

M

| Page 12: [26] Deleted | Editor | 29/09/2017 11:10:00 |
|---|---|---|

The m

| Page 12: [26] Deleted | Editor | 29/09/2017 11:10:00 |
|---|---|---|

The m

| Page 12: [26] Deleted | Editor | 29/09/2017 11:10:00 |
|---|---|---|

The m

| Page 12: [26] Deleted | Editor | 29/09/2017 11:10:00 |
|---|---|---|

The m

| Page 12: [26] Deleted | Editor | 29/09/2017 11:10:00 |
|---|---|---|

The m

| Page 12: [26] Deleted | Editor | 29/09/2017 11:10:00 |
|---|---|---|

The m

| Page 12: [26] Deleted | Editor | 29/09/2017 11:10:00 |
|---|---|---|

The m

| Page 12: [26] Deleted | Editor | 29/09/2017 11:10:00 |
|---|---|---|

The m

| Page 12: [26] Deleted | Editor | 29/09/2017 11:10:00 |

The m

| Page 12: [26] Deleted | Editor | 29/09/2017 11:10:00 |

The m

| Page 12: [26] Deleted | Editor | 29/09/2017 11:10:00 |

The m

| Page 12: [26] Deleted | Editor | 29/09/2017 11:10:00 |

The m

| Page 12: [27] Deleted | Editor | 28/09/2017 18:07:00 |

e.g.

| Page 12: [27] Deleted | Editor | 28/09/2017 18:07:00 |

e.g.

| Page 12: [28] Deleted | Editor | 29/09/2017 11:12:00 |

,

| Page 12: [28] Deleted | Editor | 29/09/2017 11:12:00 |

,

| Page 12: [28] Deleted | Editor | 29/09/2017 11:12:00 |

,

| Page 13: [29] Deleted | Editor | 29/09/2017 11:13:00 |

what

| Page 13: [29] Deleted | Editor | 29/09/2017 11:13:00 |

what

| Page 13: [29] Deleted | Editor | 29/09/2017 11:13:00 |

what

| Page 13: [30] Deleted | Editor | 28/09/2017 18:07:00 |

8ºC

| Page 13: [30] Deleted | Editor | 28/09/2017 18:07:00 |

8ºC

| Page 13: [30] Deleted | Editor | 28/09/2017 18:07:00 |

8ºC

| Page 13: [30] Deleted | Editor | 28/09/2017 18:07:00 |

8ºC

| Page 13: [30] Deleted | Editor | 28/09/2017 18:07:00 |

8ºC

| Page 13: [30] Deleted | Editor | 28/09/2017 18:07:00 |

8ºC

| Page 13: [30] Deleted | Editor | 28/09/2017 18:07:00 |

8ºC

| Page 13: [31] Deleted | Editor | 29/09/2017 11:14:00 |

D

| Page 13: [31] Deleted | Editor | 29/09/2017 11:14:00 |

D

| Page 13: [31] Deleted | Editor | 29/09/2017 11:14:00 |

D

| Page 13: [31] Deleted | Editor | 29/09/2017 11:14:00 |

D

| Page 13: [31] Deleted | Editor | 29/09/2017 11:14:00 |

D

| Page 13: [32] Deleted | Editor | 29/09/2017 11:16:00 |

was

| Page 13: [32] Deleted | Editor | 29/09/2017 11:16:00 |

was

| Page 13: [32] Deleted | Editor | 29/09/2017 11:16:00 |

was

| Page 13: [32] Deleted | Editor | 29/09/2017 11:16:00 |

was

| Page 13: [32] Deleted | Editor | 29/09/2017 11:16:00 |

was

| Page 13: [32] Deleted | Editor | 29/09/2017 11:16:00 |

was

| Page 13: [32] Deleted | Editor | 29/09/2017 11:16:00 |

was

| Page 13: [32] Deleted | Editor | 29/09/2017 11:16:00 |

was

| Page 13: [32] Deleted | Editor | 29/09/2017 11:16:00 |
|---|---|---|

was

| Page 13: [32] Deleted | Editor | 29/09/2017 11:16:00 |
|---|---|---|

was

| Page 13: [32] Deleted | Editor | 29/09/2017 11:16:00 |
|---|---|---|

was

| Page 13: [32] Deleted | Editor | 29/09/2017 11:16:00 |
|---|---|---|

was

| Page 13: [33] Deleted | Editor | 29/09/2017 11:28:00 |
|---|---|---|

H

| Page 13: [33] Deleted | Editor | 29/09/2017 11:28:00 |
|---|---|---|

H

| Page 13: [33] Deleted | Editor | 29/09/2017 11:28:00 |
|---|---|---|

H

| Page 13: [33] Deleted | Editor | 29/09/2017 11:28:00 |
|---|---|---|

H

| Page 13: [33] Deleted | Editor | 29/09/2017 11:28:00 |
|---|---|---|

H

| Page 13: [33] Deleted | Editor | 29/09/2017 11:28:00 |
|---|---|---|

H

| Page 13: [33] Deleted | Editor | 29/09/2017 11:28:00 |
|---|---|---|

H

| Page 13: [33] Deleted | Editor | 29/09/2017 11:28:00 |
|---|---|---|

H

| Page 13: [33] Deleted | Editor | 29/09/2017 11:28:00 |
|---|---|---|

H

| Page 13: [33] Deleted | Editor | 29/09/2017 11:28:00 |
|---|---|---|

H

| Page 13: [33] Deleted | Editor | 29/09/2017 11:28:00 |
|---|---|---|

H

| Page 13: [33] Deleted | Editor | 29/09/2017 11:28:00 |
|---|---|---|

H

| Page 16: [34] Deleted | Editor | 29/09/2017 11:48:00 |
|---|---|---|

like

| Page 16: [34] Deleted | Editor | 29/09/2017 11:48:00 |
|---|---|---|

like

| Page 16: [34] Deleted | Editor | 29/09/2017 11:48:00 |
|---|---|---|

like

| Page 16: [34] Deleted | Editor | 29/09/2017 11:48:00 |
|---|---|---|

like

| Page 16: [34] Deleted | Editor | 29/09/2017 11:48:00 |
|---|---|---|

like

| Page 16: [34] Deleted | Editor | 29/09/2017 11:48:00 |
|---|---|---|

like

| Page 16: [34] Deleted | Editor | 29/09/2017 11:48:00 |
|---|---|---|

like

| Page 16: [35] Deleted | Editor | 29/09/2017 11:50:00 |
|---|---|---|

overviewed

| Page 16: [35] Deleted | Editor | 29/09/2017 11:50:00 |
|---|---|---|

overviewed

| Page 16: [35] Deleted | Editor | 29/09/2017 11:50:00 |
|---|---|---|

overviewed

| Page 16: [35] Deleted | Editor | 29/09/2017 11:50:00 |
|---|---|---|

overviewed

| Page 16: [35] Deleted | Editor | 29/09/2017 11:50:00 |
|---|---|---|

overviewed

| Page 16: [35] Deleted | Editor | 29/09/2017 11:50:00 |
|---|---|---|

overviewed

| Page 16: [35] Deleted | Editor | 29/09/2017 11:50:00 |
|---|---|---|

overviewed

| Page 16: [35] Deleted | Editor | 29/09/2017 11:50:00 |
|---|---|---|

overviewed

| Page 16: [35] Deleted | Editor | 29/09/2017 11:50:00 |
|---|---|---|

overviewed

| Page 16: [35] Deleted | Editor | 29/09/2017 11:50:00 |
|---|---|---|

overviewed

| Page 16: [36] Deleted | Editor | 28/09/2017 18:07:00 |
|---|---|---|

Yet,

| Page 16: [36] Deleted | Editor | 28/09/2017 18:07:00 |
|---|---|---|

Yet,

| Page 16: [36] Deleted | Editor | 28/09/2017 18:07:00 |
|---|---|---|

Yet,

| Page 16: [36] Deleted | Editor | 28/09/2017 18:07:00 |
|---|---|---|

Yet,

| Page 16: [36] Deleted | Editor | 28/09/2017 18:07:00 |
|---|---|---|

Yet,

| Page 16: [36] Deleted | Editor | 28/09/2017 18:07:00 |
|---|---|---|

Yet,

| Page 16: [36] Deleted | Editor | 28/09/2017 18:07:00 |
|---|---|---|

Yet,

| Page 16: [36] Deleted | Editor | 28/09/2017 18:07:00 |
|---|---|---|

Yet,

| Page 16: [36] Deleted | Editor | 28/09/2017 18:07:00 |
|---|---|---|

Yet,

| Page 16: [36] Deleted | Editor | 28/09/2017 18:07:00 |
|---|---|---|

Yet,

| Page 16: [36] Deleted | Editor | 28/09/2017 18:07:00 |
|---|---|---|

Yet,

| Page 16: [36] Deleted | Editor | 28/09/2017 18:07:00 |
|---|---|---|

Yet,

| Page 16: [36] Deleted | Editor | 28/09/2017 18:07:00 |
|---|---|---|

Yet,

| Page 16: [37] Deleted | Editor | 29/09/2017 11:57:00 |
|---|---|---|

documentation of

---

## Author Response (AR3)

**Response to the third round of reviewers' comments on the manuscript "Dense water formation in the coastal northeastern Adriatic Sea: the NAdEx 2015 experiment" by Vilibić and co-authors, submitted to Ocean Science (osd-2017-6-R2)**

We thank to both reviewers for their comments, efforts and patience. Thanks to the raised comments, quality of the manuscript has been significantly improved.

Reviewer #1 comments:

*This is the third review of the manuscript « Dense water formation in the coastal northeastern Adriatic Sea: the NAdEx 2015 experiment » by Vilibic et al. The authors have provided again a detailed response to the previous comments, in particular to all major concerns. I therefore recommend a minor review, addressing however a series of general and particular issues.*

- Thanks for doing such a throughout and detailed review!

*General remarks:*
*- In general, please think of a reader who doesn't know the geography of the region and who doesn't know by heart the location in time/space of all measurements. Try to help the reader to easily follow the results in time and space.*

- We added more geographical information in both figures and text (e.g. Figs. 5 and 6).

*- The results sections are still very descriptive, and I think that a summary at the end of each section of the main conclusions would help the reader to make a synthesis. One often ends a section with an accumulation of descriptive informations, but with no general and synthetic overview of the main results.*

- We added a short (2-3 sentences) summary in sections 4 and 6 (6.1 and 6.2), while Sections 3 and 5 already provide conclusions in their last paragraphs.

*p2 L33 hurricane magnitudes*
*p3 L9 advected, or being advected*
*p3 L11 was not considered*
*p4 L3 Senj. These inputs*
*p4 L5 many submarine*

- All corrected.

*Figure 3 and 5: the observation and model plots could be harmonized to ease the reading.*

- Done.

*p8 L20 which statistical significance test?*

- We applied t-test. The information is added in the manuscript.

*p8 L24 does the Figure display twice approximately the same track? If it is the case, give in abscissa the geographic location along the track (longitude and/or latitude)*

- Yes. We added small inset with glider pathway and time labels in Fig. 6.

*p9 L20 no inflow is visible at the bottom of station A9*

- The text is corrected.

*Figure 7: why is a significant fraction of the transport along-channel and not across-channel in stations A8 and A9? Are the current measurements located within each channel or outside them?*

- A7-A9 current measurements are located at the entrance of Kvarner Bay (see Fig. 1), which is quite wide (compared to other connecting passages), under strong influence of the open Adriatic circulation and therefore it has significant 2D dynamics. Also, wind-driven curl (of bora wind) is strong in the area, resulting in an anticyclonic circulation prevailing during bora forcing (see p9 L18-21). This is now clarified in the manuscript.

*p10 L9: the temperature overestimation close to the coast (stations 15 to 19) has not been commented. How does it relate to the salinity bias?*

- Temperature and salinity biases during leg 1 cruise are related to each other at stations 14-19, both being positive. That is much clearer now in the replotted Fig. 3. Two explanations are offered in the text now: (i) effect of submarine springs that discharge freshwater from the neighbouring freshwater lake, not introduced to the model, or (ii) advection of water masses from the southeast.

*p10 L14: (Fig. 8b).*
*p10 L16: (15-35m, Fig. 8a).*

- Both corrected.

*p10 L21-31:*

*- a geographic indication (longitude and/or latitude) is necessary in the abscissa for the reader to follow the text.*

- An inset is added in Fig. 6, containing the pathway of the float.

*- you mention an increase in negative bias although we see in both Figures an overall positive bias: is the model bias (model minus observations) displayed? In which case what increase of negative bias are you referring to?*

- Your observation is correct, it is "an increase in positive bias from the inner to the outer parts of the bay". The text is corrected.

*- the colorbar saturates in Figure 5b: please widen the range of values so that the reader can identify the magnitude of the bias.*

- Done.

*p11 L1: for the 3rd time, please explain how you compute the dashed-dotted lines: you mention in the legend that it is the « Q-Q slope of a particular dataset », but how do you determine it? For instance, why does the light blue line in Figure 9b have a much larger slope than the light blue dots? It does not seem to be a least square fit.*

- Dash-dot lines fit the second and third quartile of each Q-Q distribution. Explanation is now added to the Fig. 9 caption.

*Figure 10: the previous colorbar was fine, the problem was the absence of any MLD computation. I think the current colorbar reads worse and you should use the previous one.*

- The figure is replotted using previous colobar.

*p12 L13: the De Boyer Montegut, GRL 2004's density threshold of 0.125kg/m³ is not adapted to the Mediterranean Sea due to the low stratification in this basin, especially in DWF areas. This is why using this criterion, D'Ortenzio et al, GRL 2005 were unable to successfully identify the main Mediterranean DWF areas with this criterion (their Figure 1, months January to March). The usual threshold is DT=0.1°C or Drho=0.02kg/m³ (Houpert et al, PiO 2015, Somot et al, CD 2016), which allowed for instance Houpert et al, PiO 2015 to identify the main Mediterranean DWF areas (their Figure 7).*

- We recalculated MLD by using threshold DT=0.1°C. Figs. 10 and 11 are replotted following new MLD estimates.

*p12 L25 and last paragraph of 6.1: Figure 12 and Figure 11 seem to have been reversed. Please display first the first Figure that is commented in the text.*

- Corrected.

*p13 L12: they are surface buoyancy fluxes, not buoyancy changes. Also, in Figure 11 (not 12), it is daily integral values (in m²/s²), not daily values (in m²/s³), which are displayed.*

- Corrected.

*p14 L5 what do you mean by « constructions »?*

- Deleted.

*p15 L2: Uout (the outward velocity) is already U.n (the total velocity projected onto the outter normal unit vector), so that you can replace Uout.n simply by Uout in the formula.*

- We prefer to have the equation in a general form, as the outward velocity in a constriction does not necessary need to be normal to contour *C*.

*p15 L5: there is no need to assume no incoming water, but simply that the incoming flux is equal to the outcoming one, which is reasonable over long timescales because otherwise the sea level would vary a lot.*

- Omitted.

*p16, L.6: it is not a coupled atmosphere-ocean model because there is no feedback between both models. It is a nested oceanic model which is forced by an assimilated atmospheric model.*

- Corrected.

*p16, L.10: I have not understood your conclusion (ii).*

- Clarified.

*p16, L25: why do you cite a 2006 paper to comment on the excitation of the thermohaline circulation in the 2015 event? Orlic et al 2006 seem inappropriate here.*

- The reference is removed here.

*General comment to the Authors and the Editor:*

*With respect to the previous version, I must admit that now the discussion of data and model results is much clearer, and most of the confusion I found in the previous versions of the paper has been eliminated. Also English has much improved and I want to give credit to the authors for this big effort. However, and I am sorry that I might be repetitive….in my opinion the paper does not answer the questions it poses at the beginning. And this, to me, is now much clearer as the paper structure is much clearer. Most of the discussion is not even related with DWF. So I only have one advise, otherwise I cannot accept the paper: the title must be changed and the objectives. Other parts of the paper might undergo to some change accordingly. I know it is an incisive change, but this paper gives a nice and comprehensive description of the dynamics in the northeastern Adriatic area, and as such it deserves publication. But it is NOT a paper about DWF. The authors might write that they performed the experiment in an area that might be important for DWF, as shown by other studies, but not that this paper describes the DWF and how common it is in this area (with just one experiment, it is rather strange that you claim to be able to say how common DWF is).*

- Thanks for the comment, we changed the title, objectives of the manuscript, and some other relevant parts.

*Some more detailed comments are:*

*- Page 4, Line 18: you should rephrase here since it seems that RDI ADCPS were used also in A1-A3 and A5-A6. Please specify the depths of these stations as well and the depth of the SBE probes closed to the ADCPs.*

- Corrected. The depths are added to the text.

*- Page 4, Line 25-27: it appears that the cruises were not done in the period of DWF and that only the ARVOR might have "seen" something during the last episode.*

- Yes, unfortunately.

*- Page 5, Line 19: should be σ-coordinate.*

- Corrected.

*- Page 6, Line 10: Vilibic et al., 2016a or 2016b?*

- It is "a", corrected.

*- Page 6, Line 13: "warmer than average conditions"*

- Corrected.

*- Page 7, Line24. Page 8 line 26: this description has nothing to do with the objective of describing DWF in the area, it describes data taken in other periods of the year.*
- We agree.

*- Page 9, Line12-14: this hypothesis is unsupported by the observations.*
- We omitted two last sentences from this paragraph.

*- Page 9, Line 20-21: I cannot see this inflow in fig. 7 at A9….*
- The sentence is rephrased to follow the observations.

*- Page 9, line 15-16: showing a two months average of the circulation (fig. 7) is not that much significant in a coastal area where a lot of mesoscale features are expected and is therefore not useful to describe outflowing dense waters.*
- The sentence is rephrased.

*- Fig. 7 does not show the scale of velocities.*
- Added.

*- Caption of Figure 8 needs a more detailed description.*
- The caption is extended.

*- Page 10, line 20: the fact that the modelled PDA during leg 2 (which is in May) was higher that the observed PDA in December, is not at all demonstrating that the model was able to reproduce DWF in the Nadex area.*
- The second part of the sentence is omitted.

*- Page 12, line 14: you are saying that what you observe in one point, G2, is telling you something about "most of the basin"? (again stated at line 21….)*
- This part of the sentence is omitted from the manuscript.

*- Page 13, line 23: you don't say anything on how these fluxes have been computed…with the model? And if yes, why in these locations if no comparison with data-based flux computation is done?*

- Section 6 are all model computations. These locations, or better to say transects (transports are computed over whole transects and not just locations where measurements are executed), are chosen as bordering coastal and open Adriatic waters ($T_{outer}$) or bordering Velebit Channel from the rest of the coastal waters ($T_{inner}$). Thus the total flux between coastal and open waters may be assessed.

*- Page 16, line 1: "into the processes"....which ones?*
- The sentence is rewritten.

*- Page 16, line 6-7: your dataset is not able to answer to question (i).*
- This sentence as omitted as already introduced in Section 1.

*- Page 16, line 9: the CTD data were collected in December and in May, so they cannot show you that DWF did occur in the nadex area.*
- Corrected.

*- Page 16, line 12: the dense water is transported outward at the Surface layer?*
- Corrected.

*- Fig. 6: would be useful to indicate on the plots where is east and where west.*
- We added a small map to visualise geography of measurements (also available in Fig. 1).

*- Caption figure 9: what is the black line?*
- Black line is simply y=x line, denoting pairs of equal model and observation percentiles.

*- Figure 10: use the same colorbar as in fig- 3*
- Ok.

*- Caption fig. 17: should be "of total (TRT) and dense water (DWRT)"*
- Corrected.

[revised manuscript text omitted]

---

## Author Response (AR4)

**Response to the fourth round of reviewers' comments on the manuscript "Wintertime dynamics in the coastal northeastern Adriatic Sea: the NAdEx 2015 experiment" by Vilibić and co-authors, submitted to Ocean Science (osd-2017-6-R3)**

We thank to the reviewer for his/her final comments and polishing of the manuscript.

Reviewer #1 comments:

*p.2 L.16: I do not believe that only the spatial inhomogeneity of the Bora wind can explain the Adriatic double-gyre circulation. Hendershott and Rizzoli, DSR 1976 also put forward the key roles of:*

*- the coastal inflow of freshwater to enhance the horizontal density gradient that feeds the cyclonic circulation;*

*- the ocean bottom topography to force the double cyclonic circulation.*

- Hendershott and Rizzoli (1976) assume (1) spatially uniform surface heat loss, (2) diffusive river inflow, and (3) realistic bottom topography, and therefore they do not obtain the double-gyre circulation. As mentioned in several later papers (e.g. Beg Paklar et al., 2001), if it is allowed that (a) surface heat loss is not uniform (being related to the spatial variability of the bora wind) and (b) that river inflows are concentrated at some points, the circulation changes and two gyres could form. We accordingly modified the sentence and added key references (Supić et al., 1997; Beg Paklar et al., 2001).

*p.6 L.20-21: can you specify which percentiles you are referring to? Is it percentiles of the climatological mean distribution (30-year mean) or only of that particular year?*

- The percentiles are of months in 2015 compared to the baseline 30-year period. The information is added in the text.

*L.26-28: a cumulative heat loss of 0.80MJ/m²=800,000J/m² over those 2 months is equivalent to a mean heat loss of 800,000/(31+28)/86400=0.16W/m². Therefore I deduce that there is an error of unit that could have been avoided. Is it GJ rather than MJ? It would make it 160W/m², which is the right order of magnitude.*

- Yes, the losses are in GJ/m$^2$, thanks for the correction.

*p.8 L.18: continuous rather than constant*

- Corrected.

*p.9 L.3-5: the southeast-northwest differences in T,S are not clearly visible in the PDA series: I would rather say that they are density-compensated. What is mostly visible in PDA time series is the signature of the cooling trend.*

- The sentence is rephrased.

*p.10 L.5: what do you mean by constrictions?*

- Changed to 'connecting passages'.

*p.11 L.20: does*

- Corrected.

*p.12 L.1: misrepresented*

- Corrected.

*p.13 L.7: by the heat losses induced by the Bora wind*

- Corrected.

*p.13 L.30: for clarity, you could first comment on the overall negative flux, and then go to the details of individual transects T1, T3, T2 and T789.*

- The sentences are switched.

*p.14 L.2: summing, not averaging*

- Corrected.

*p.14 L.9: I guess 6 and 2 times are for resp. T and S?*

- No, for the inner and outer boundaries, it is clarified now.

*p.15 L.14: strictly speaking, you have shown that the coastal NE Adriatic loses dense water volume, not mass. Although I agree that because these dense waters are replaced by lighter incoming waters, it implies that the basin loses mass.*

- Ok, but this sentence refers to waters and not dense waters.

*p.17 L.16-19: just a remark to put into perspective the role of the Adriatic SST on Bora events. Stocchi and Davolio, Atmospheric Research 2017 show a very weak impact of air-sea water exchanges over the Adriatic Sea on the atmospheric water budget during Bora events.*

- Ok. Yet, this paper is describing impact of SST uncertainty during bora to precipitation over the land (Apennines) and not impact to the ocean dynamics, what is a bit out of the focus in the discussion.

[revised manuscript text omitted]